# Identification of the Generalized Condorcet Winner in Multi-dueling Bandits

**Björn Haddenhorst**
Department of Computer Science
Paderborn University
Paderborn, Germany
bjoernha@mail.upb.de

**Viktor Bengs**
Institute of Informatics
University of Munich (LMU)
Munich, Germany
viktor.bengs@lmu.de

**Eyke Hüllermeier**
Institute of Informatics
University of Munich (LMU)
Munich, Germany
eyke@ifi.lmu.de

## Abstract

The reliable identification of the "best" arm while keeping the sample complexity as low as possible is a common task in the field of multi-armed bandits. In the multi-dueling variant of multi-armed bandits, where feedback is provided in the form of a winning arm among a set of $k$ chosen ones, a reasonable notion of best arm is the generalized Condorcet winner (GCW). The latter is an arm that has the greatest probability of being the winner in each subset containing it. In this paper, we derive lower bounds on the sample complexity for the task of identifying the GCW under various assumptions. As a by-product, our lower bound results provide new insights for the special case of dueling bandits ($k = 2$). We propose the Dvoretzky–Kiefer–Wolfowitz tournament (DKWT) algorithm, which we prove to be nearly optimal. In a numerical study, we show that DKWT empirically outperforms current state-of-the-art algorithms, even in the special case of dueling bandits or under a Plackett-Luce assumption on the feedback mechanism.

## 1 Introduction

The standard multi-armed bandit (MAB) problem describes a sequential decision scenario, in which one of finitely many choice alternatives must be selected in each time step, resulting in the observation of a numerical reward of stochastic nature. One important and extensively studied variant of the MAB setting is the dueling bandits problem, where a duel consisting of two arms is chosen in each time step and one of the duelling arms is observed as the winner [4]. Recently, the multi-dueling bandits setting has been introduced [7, 40, 31] as a generalization with multiple practically relevant applications, such as algorithm configuration [13] or online retrieval evaluation [36]. Instead of pairs of arms, in this generalization a set consisting of $k \geq 2$ arms can be chosen in each time step. These arms compete against each other and determine a single winner, which is observed as feedback by the learner. The outcomes of the (multi-)duels in the (multi-)dueling bandit scenario are typically assumed to be of time-stationary stochastic nature in the sense that whenever arms $a_1, \ldots, a_k$ compete against each other, then $a_i$ wins with some underlying (unknown) ground-truth probability $\mathbf{P}(a_i | \{a_1, \ldots, a_k\})$.

One often targeted learning task in the context of multi-armed bandits and its variants is the problem of identifying the best among all arms. While for standard MABs, the canonical definition of the

35th Conference on Neural Information Processing Systems (NeurIPS 2021).

"best arm" is the arm with highest expected reward, the picture is less clear for its variants. In the realm of dueling bandits, any arm that is likely to win (i.e., with probability $> 1/2$) in each duel against another arm is called the *Condorcet winner* (CW). This notion dates back to the 18th century [8] and also appears in the social choice literature [16, 17], where the data is typically assumed to be available in the form of a list containing total rankings over all alternatives from different voters. In practice, the Condorcet winner does not necessarily exist due to the presence of preferential cycles in the probabilistic model in the sense that $a_i$ is likely to win against $a_j$, $a_j$ against $a_k$, and $a_k$ against $a_i$. For the theoretical analysis of the best-arm-identification problem, this issue is overcome in the literature either by the consideration of alternative optimality concepts such as *Borda winner* or *Copeland winner*, which are guaranteed to exist, or by simply assuming the existence of the CW.

In this paper, we focus on finding a generalized variant of the CW in the multi-dueling bandits setting under the assumption that it exists. There have been several suggestions for generalizations of the CW in social choice. For example, a weighted variant is introduced in [30], where the weights control the relevance given to the ranking positions of the alternatives, while in [25] the notion of a $k$-winner is defined as an alternative that (in some appropriate sense) outperforms all other arms among any $k$ alternatives. In contrast to our work, these papers focus on offline learning tasks and suppose full rankings over all alternatives to be given. In this paper, we adapt the notion of *generalized Condorcet winner* (GCW) as in [1], i.e., a GCW is an arm $a_i$ that outperforms each arm $a_j$ in every query set $S$ containing both $a_i$ and $a_j$, in the sense that $\mathbf{P}(a_i|S) \geq \mathbf{P}(a_j|S)$.

Regarding the dueling bandits setting as the multi-dueling setting where the allowed multi-duels $S$ are exactly those with $|S| = 2$, the GCW is indeed a generalization of the Condorcet winner. We analyze the sample complexity of (probabilistic) algorithms that are able to identify the GCW with high probability under the assumption of mere existence as well as more restrictive assumptions. We provide upper and lower bounds for this task, which depend on the desired confidence, the total number $m$ of alternatives, the size $k$ of allowed query sets as well as the underlying unknown preference probabilities $\mathbf{P}(a_i|S)$.

We start in Section 2 with a brief literature overview on the multi-dueling bandits scenario. Section 3 introduces the basic formalism and a precise definition of the considered GCW identification problem. It also gives a rough, simplified overview of the sample complexity bounds obtained in this paper. In Section 4, we discuss the special case $m = k$, in which the GCW identification problem essentially boils down to the task of finding the mode of a categorical distribution. We provide solutions to this problem and prove their sample complexity to be optimal up to logarithmic factors in the worst-case sense. Section 5 focuses on lower bounds for the general case $m \geq k$, and in Section 6, we discuss several upper bounds. In Section 7, we empirically compare the algorithms discussed before, prior to concluding in Section 8. For the sake of convenience, detailed proofs of all theoretical results presented in the paper are deferred to the supplemental material.

## 2 Related Work

Initially, the multi-dueling bandit problem was studied intensively in the case of pairs as actions of the learner, which is also better known as *dueling bandits* [42]. The extension to the scenario considered in this paper, where more general sets as pairs of arms are selectable as an action, has been the focus of recent work. Part of these works model the feedback process by essentially tracing it back to the dueling bandit case [7, 40, 31]. The majority of papers, however, assume latent utility values for the arms and model the feedback process using a random utility model (RUM) [3] based on these utility values. Thanks to the latent utility values, an ordering of the arms is obtained quite naturally, which in turn makes it easy to define an objective such as the optimal arm or the top-$k$ arms. Under these assumptions, the PB-MAB problem was investigated with respect to various performance metrics such as the regret [32, 5, 1] or the sampling complexity in an $(\epsilon, \delta)$-PAC setting [33, 34, 35].

In [1] a different approach is taken by generalizing the concept for the naturally optimal arm in the dueling bandit case, namely the Condorcet winner (CW), under the term generalized Condorcet winner (GCW). The optimal arm defined in this way coincides with the optimal arm if latent utility values for the arms and a RUM for the feedback process are assumed. While in [1] the problem for finding this GCW is investigated in a regret minimization scenario, we are interested in the minimum sampling complexity. In light of this, the work by [35] is the most related to ours, although the authors assume a PL model (a special case of a RUM).

If we restrict the learner's actions to pairs of arms in our more general setting, i.e., the dueling bandits case, the GCW and the CW coincide. This special case of our problem setting has been dealt with by [26], [20] and [29].

Finally, it remains to mention that there are a number of similar problem scenarios, namely the Stochastic click model (SCM) [43], the dynamic assortment problem (DAS) [9] and the best-of-k-bandits [39]. However, all these scenarios take into account other specific aspects in the modelling such as the order of the arms in the action subset (SCM), known revenues associated with the arms (DAS) or a so-called "no-choice option" (all three). Accordingly, these problem scenarios are fundamentally different from our learning scenario (see also Sec. 6.6 in [4] for a more detailed discussion). The same is true for combinatorial bandits [10], which also allow subsets of arms as actions, but differ fundamentally in the nature of feedback (quantitative vs. qualitative feedback).

# 3 The GCW Identification Problem

For adequately stating our results, we introduce in the following some basic terminology and notations used throghout this paper. For the sake of convenience, Table 1 summarizes the most frequently used notations.

## 3.1 The Notion of a GCW

If not explicitly stated otherwise, we suppose throughout the paper the total number of arms $m$, the query set size $k \in \{2, \ldots, m\}$, a desired confidence $1 - \gamma \in (0, 1)$ and a complexity parameter $h \in (0, 1)$ to be arbitrary but fixed. We write $[m] := \{1, \ldots, m\}$ and $[m]_k := \{S \subseteq [m] \,|\, |S| = k\}$.

**Parameter spaces of categorical distributions.** For any subset of size $k$, i.e., $S \in [m]_k$, define $\Delta_S := \{\mathbf{p} = (p_j)_{j \in S} \in [0,1]^{|S|} \,|\, \sum_{j \in S} p_j = 1\}$ as the set of all possible parameters for a categorical random variable $X \sim \mathrm{Cat}((p_j)_{j \in S})$, i.e., $\mathbb{P}(X = j) = p_j$ for any $j \in S$. For $\mathbf{p} \in \Delta_S$, we write $\mathrm{mode}(\mathbf{p}) := \arg\max_{j \in S} p_j$ and in case $|\mathrm{mode}(\mathbf{p})| = 1$ we denote by $\mathrm{mode}(\mathbf{p})$ — with a slight abuse of notation — also the unique element in $\mathrm{mode}(\mathbf{p})$. Let us define for $h \in (0, 1]$ the sets

$$\Delta_S^h := \left\{ \mathbf{p} \in \Delta_S \,|\, \exists i \in S \text{ s.t. } p_i \geq \max_{j \in S \setminus \{i\}} p_j + h \right\},$$

and with this $\Delta_S^0 := \bigcup_{h \in (0,1)} \Delta_S^h$. These sets are nested in the sense that $\Delta_S^h \subseteq \Delta_S^{h'} \Leftrightarrow h \geq h'$. If $\mathbf{p} \in \Delta_S$ is fixed, the value $h(\mathbf{p}) := \max\{h \in [0, 1] \,|\, \mathbf{p} \in \Delta_S^h\}$ is well-defined and we have $\mathbf{p} \in \Delta_S^h$ iff $h \leq h(\mathbf{p})$. Obviously, the equivalence $|\mathrm{mode}(\mathbf{p})| = 1 \Leftrightarrow \mathbf{p} \in \Delta_S^0$ holds for all $\mathbf{p} \in \Delta_S$.

**Probability models on $[m]_k$.** A family $\mathbf{P} = \{\mathbf{P}(\cdot \,|\, S)\}_{S \in [m]_k}$ of parameters $\mathbf{P}(\cdot \,|\, S) \in \Delta_S$, $S \in [m]_k$, is called a **probability model (short: PM) on** $[m]_k$. We write $PM_k^m$ for the set of all probability models on $[m]_k$ and define the following subsets of $PM_k^m$ :

$$PM_k^m(\Delta^0) := \{\mathbf{P} = \{\mathbf{P}(\cdot \,|\, S)\}_{S \in [m]_k} \,|\, \forall S \in [m]_k \,:\, \mathbf{P}(\cdot \,|\, S) \in \Delta_S^0\},$$
$$PM_k^m(\Delta^h) := \{\mathbf{P} = \{\mathbf{P}(\cdot \,|\, S)\}_{S \in [m]_k} \,|\, \forall S \in [m]_k \,:\, \mathbf{P}(\cdot \,|\, S) \in \Delta_S^h \},$$
$$PM_k^m(\mathrm{PL}) := \{\{\mathbf{P}(\cdot \,|\, S)\}_{S \in [m]_k} \,|\, \exists \boldsymbol{\theta} \in (0, \infty)^m \,\forall S \in [m]_k \,:\, \mathbf{P}(i \,|\, S) = \theta_i / (\sum_{j \in S} \theta_j)\}.$$

Note that $PM_k^m(\mathrm{PL})$ denotes the set of all probability models $\mathbf{P}$ consistent with a Plackett-Luce (PL) model [27, 23]. Let $h(\mathbf{P}) := \max_{h \in [0,1]} \{\mathbf{P} \in PM_k^m(\Delta^h)\} = \min_{S \in [m]_k} h(\mathbf{P}(\cdot \,|\, S))$, then it is easy to see that $\mathbf{P} \in PM_k^m(\Delta^h)$ iff $h \leq h(\mathbf{P})$.

An element $i \in [m]$ is called a **generalized Condorcet Winner** (short: GCW) of $\mathbf{P}$ if

$$\forall S \in [m]_k \text{ with } i \in S, \forall j \in S : \mathbf{P}(i \,|\, S) - \mathbf{P}(j \,|\, S) \geq 0$$

and we write $\mathrm{GCW}(\mathbf{P})$ for the set of all GCWs of $\mathbf{P}$. With this, we define the following subsets of $PM_k^m$ related to the concept of the GCW:

$$PM_k^m(\exists\mathrm{GCW}) := \{\mathbf{P} = \{\mathbf{P}(\cdot \,|\, S)\}_{S \in [m]_k} \,|\, \mathrm{GCW}(\mathbf{P}) \neq \emptyset\},$$
$$PM_k^m(\exists\mathrm{GCW}^*) := \{\mathbf{P} = \{\mathbf{P}(\cdot \,|\, S)\}_{S \in [m]_k} \,|\, |\mathrm{GCW}(\mathbf{P})| = 1\},$$
$$PM_k^m(\exists h\mathrm{GCW}) := \{\{\mathbf{P}(\cdot \,|\, S)\}_{S \in [m]_k} \,|\, \exists i : \forall S \in [m]_k, j \in S \setminus \{i\} : \mathbf{P}(i \,|\, S) - \mathbf{P}(j \,|\, S) \geq h\}.$$

| | |
|---|---|
| $m$ | the total number of arms |
| $k$ | the query set size |
| $\gamma$ | the desired error rate bound |
| $[m]$ | the set $\{1, \dots, m\}$ |
| $[m]_k$ | the set of all subsets of $[m]$ of size $k$ |
| $\mathbf{1}_A$ | indicator function, which is 1 if $A$ is a true statement and 0 otherwise; also denoted by $\mathbf{1}_{\{A\}}$ |
| $S$ | an element from $[m]_k$ |
| $PM_k^m$ | set of all parameters $\{\mathbf{P}(\cdot|S)\}_{S \in [m]_k} \subseteq [0,1]^{\binom{m}{k}}$ with $\sum_{j \in S} \mathbf{P}(j|S) = 1 \ \forall S \in [m]_k$ |
| $PM_k^m(\mathrm{X})$ | the set of all $\mathbf{P}$, which fulfill the condition(s) X |
| $PM_k^m(\mathrm{X} \wedge Y)$ | the set $PM_k^m(\mathrm{X}) \cap PM_k^m(\mathrm{Y})$ |
| $\mathbf{P}$ | an element from $PM_k^m$ |
| $\mathrm{GCW}(\mathbf{P})$ | set of all GCWs of $\mathbf{P}$; if $|\mathrm{GCW}(\mathbf{P})| = 1$ it denotes the only element in $\mathrm{GCW}(\mathbf{P})$ |
| | $\Delta^h, \Delta^0, \mathrm{PL}, \exists\mathrm{GCW}, \exists h\mathrm{GCW}$ and $\exists\mathrm{GCW}^*$, cf. Section 3.1 |
| $\Delta_S$ | set of all $\mathbf{w} = (w_i)_{i \in S} \in [0,1]^{|S|}$ with $\sum_{i \in [m]} w_i = 1$; here, $S$ is a finite set |
| $\Delta_S^h$ | set of all $\mathbf{w} \in \Delta_S$, for which $i \in S$ exists with $\forall j \in S \setminus \{i\} : w_i \geq w_j + h$ |
| $\Delta_k, \Delta_k^h$ | $\Delta_{[k]}$ resp. $\Delta_{[k]}^h$ |
| $\mathbf{p}$ | an element from $\Delta_k$ or an element from $\Delta_S$ for some $S \in [m]_k$ |
| $\mathrm{mode}(\mathbf{p})$ | $\arg\max_{i \in [k]} p_i$ for $\mathbf{p} = (p_1, \dots, p_k)$; the term $\mathrm{mode}(\mathbf{P}(\cdot|S))$ is defined accordingly |
| $h(\mathbf{p})$ | $\max\{h \in [0,1] \,|\, \mathbf{p} \in \Delta_k^h\}$ for $\mathbf{p} \in \Delta_k$ |
| $h(\mathbf{P})$ | $\max\{h \in [0,1] \,|\, \mathbf{P} \in PM_k^m(\Delta^h)\}$ |
| $\mathcal{A}$ | an algorithm |
| $\mathbf{D}(\mathcal{A})$ | the return value of $\mathcal{A}$ |
| $T^{\mathcal{A}}$ | the sample complexity of $\mathcal{A}$, i.e., the number of samples observed by $\mathcal{A}$ before termination |
| $\mathcal{A}(x_1, \dots, x_l)$ | An algorithm $\mathcal{A}$ called with the parameters $x_1, \dots, x_l$ |
| $\mathcal{P}_k^{m,\gamma}(\mathrm{X})$ | Problem of finding for any $\mathbf{P} \in PM_k^m(\mathrm{X})$ with error prob. $\leq \gamma$ the GCW, cf. Def. 3.1 |

Clearly, it holds that $PM_k^m(\exists\mathrm{GCW}^*) = \bigcup_{h>0} PM_k^m(\exists h\mathrm{GCW})$ and every probability model $\mathbf{P} \in PM_k^m(\exists\mathrm{GCW})$ has at least one GCW, while for a probability model $\mathbf{P} \in PM_k^m(\exists\mathrm{GCW}^*)$ the GCW is unique.

The figure on the right illustrates the relationships between the introduced subsets of $PM_k^m$. For the sake of convenience, we write simply (X) instead of $PM_k^m(\mathrm{X})$, where

$$\mathrm{X} \in \{\exists\mathrm{GCW}, \exists\mathrm{GCW}^*, \exists h\mathrm{GCW}, \Delta^0, \Delta^h, \mathrm{PL}\}.$$

This convention will be used several times in the course of the paper.

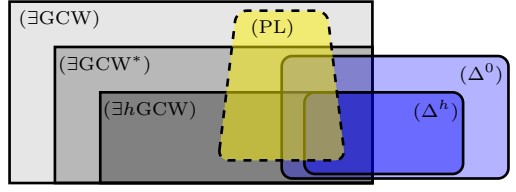

## 3.2 Problem Formulation

We are interested in algorithms $\mathcal{A}$ able to find the GCW of some $\mathbf{P} = \{\mathbf{P}(\cdot\,|\,S)\}_{S \in [m]_k} \in PM_k^m$, which is unknown and only observable via sampling from $\mathbf{P}$. More precisely, we suppose that at each time step $t \in \mathbb{N}$, such an algorithm $\mathcal{A}$ is allowed to choose one **query set** $S_t \in [m]_k$, for which it then observes a sample $S_t \ni X_t \sim \mathrm{Cat}(\mathbf{P}(\cdot\,|\,S_t))$. At some time, $\mathcal{A}$ may decide to make no more queries and output a prediction $\mathbf{D}(\mathcal{A}) \in [m]$ for the GCW. We write $T^{\mathcal{A}} \in \mathbb{N} \cup \{\infty\}$ for the sample complexity of $\mathcal{A}$, i.e., the total number of queries made by $\mathcal{A}$ before termination. Note that both $\mathbf{D}(\mathcal{A})$ and $T^{\mathcal{A}}$ are random variables w.r.t. the sigma-algebra generated by the stochastic feedback mechanism. We write $\mathbb{P}_{\mathbf{P}}$ for the probability measure corresponding to the stochastic feedback mechanism if the unknown ground-truth PM is given by $\mathbf{P}$.

**Definition 3.1** (The GCW Identification Problem). *Let* (X) *be any of the assumptions from above with* $PM_k^m(\mathrm{X}) \subseteq PM_k^m(\exists\mathrm{GCW})$, *and let* $\gamma \in (0,1)$ *be fixed. An algorithm* $\mathcal{A}$ ***solves the problem*** $\mathcal{P}_k^{m,\gamma}(\mathrm{X})$ *if* $\mathbb{P}_{\mathbf{P}}\big(\mathbf{D}(\mathcal{A}) \in \mathrm{GCW}(\mathbf{P})\big) \geq 1 - \gamma$ *holds for any* $\mathbf{P} \in PM_k^m(\mathrm{X})$.

## 3.3 Overview of Results

In this paper, we provide several upper and lower sample complexity bounds for solutions to the GCW identification problem $\mathcal{P}_k^{m,\gamma}(\mathrm{X})$ under different assumptions (X). We start in Section 4 with

the discussion of the special case $m = k$, in which $\mathcal{P}_k^{k,\gamma}(\mathrm{X})$ can simply be thought of as finding the mode of a categorical distribution on $[k]$. In Sections 5 and 6, we discuss lower and upper bounds for the general case $m \geq k$, respectively.

Table 2 summarizes the obtained worst-case sample complexity bounds[1] of solutions to $\mathcal{P}_k^{m,\gamma}(\mathrm{X})$, where the worst-case is meant w.r.t. $PM_k^m(\mathrm{X} \wedge \mathrm{Y})$, for different choices of X and Y, and the Bachmann-Landau notations $\Omega(\cdot)$ and $\mathcal{O}(\cdot)$ are to be understood w.r.t. $m, k, h^{-1}$ and $\gamma^{-1}$. In addition, we also provide instance-wise bounds in Theorems 5.2 and E.1.

Table 2: Sample complexity bounds of solutions to $\mathcal{P}_k^{m,\gamma}(\mathrm{X})$

| (X) | (Y) | Type | Asymptotic bounds | References |
|---|---|---|---|---|
| (PL) | $(\exists h\mathrm{GCW})$ | in exp. | $\Omega(\frac{m}{h^2 k}(\frac{1}{k} + h) \ln \frac{1}{\gamma})$ | Thm. 5.1 |
| $(\Delta^h \wedge \exists \mathrm{GCW})$ | $(\Delta^h)$ | in exp. | $\Omega(\frac{m}{h^2 k} \ln \frac{1}{\gamma})$ | Thm. 5.2 |
| $(\mathrm{PL} \wedge \exists \mathrm{GCW}^*)$ | $(\exists h\mathrm{GCW})$ | w.h.p. | $\mathcal{O}(\frac{m}{h^2 k}(\frac{1}{k} + h) \ln(\frac{k}{\gamma} \ln \frac{1}{h}))$ | Thm. 6.1 |
| $(\exists \mathrm{GCW} \wedge \Delta^0)$ | $(\Delta^h)$ | w.h.p. | $\mathcal{O}(\frac{m}{h^2 k} \ln(\frac{m}{k})(\ln \ln \frac{1}{h} + \ln \frac{1}{\gamma}))$ | Thm. 6.2 |
| $(\exists h\mathrm{GCW} \wedge \Delta^0)$ | $(\exists h\mathrm{GCW})$ | a.s. | $\mathcal{O}(\frac{m}{h^2 k} \ln(\frac{m}{k\gamma}))$ | Thm. E.2 |

Due to $PM_k^m(\Delta^h \wedge \exists \mathrm{GCW}) \subsetneq PM_k^m(\exists h\mathrm{GCW}) \subsetneq PM_k^m(\exists \mathrm{GCW})$, Thm. 5.2 implies in particular that any solution $\mathcal{A}$ to $\mathcal{P}_k^{m,\gamma}(\exists \mathrm{GCW})$ fulfills $\sup_{\mathbf{P} \in PM_k^m(\exists h\mathrm{GCW})} \mathbb{E}_{\mathbf{P}}[T^{\mathcal{A}}] \in \Omega(\frac{m}{kh^2} \ln(\gamma^{-1}))$. As Thm. 6.1 and Thm. 6.2 indicate that the bounds in Thm. 5.1 and Thm. 5.2 are asymptotically sharp up to logarithmic factors, the GCW identification problem seems to be easier under the PL assumption by a factor $1/k + h$.

## 4 The Single Bandit Case $m = k$

In this section, we address the problem $\mathcal{P}_k^{m,\gamma}(\mathrm{X})$ for the special case $k = m$. For sake of convenience, we abbreviate $\Delta_k := \Delta_{[k]}$ and similarly $\Delta_k^h := \Delta_{[k]}^h$ for any $h \in [0,1]$. Due to $[k]_k = \{[k]\}$, any probability model $\mathbf{P} \in PM_k^k$ is completely characterized by $\mathbf{P}(\cdot|[k])$ and the GCW of $\mathbf{P}$ is simply $\mathrm{mode}(\mathbf{P}(\cdot|[k]))$. Since the latter one always exists, we have $PM_k^k \subseteq PM_k^k(\exists \mathrm{GCW})$. Note that $\mathcal{P}_k^{k,\gamma}(\Delta^0) = \mathcal{P}_k^{k,\gamma}(\exists \mathrm{GCW}^*)$ as well as $\mathcal{P}_k^{k,\gamma}(\Delta^h) = \mathcal{P}_k^{k,\gamma}(\exists h\mathrm{GCW})$ are fulfilled trivially – i.e., we do not have to distinguish between the assumptions $\Delta^0$ and $\exists \mathrm{GCW}^*$ resp. $\Delta^h$ and $\exists h\mathrm{GCW}$ throughout this section. For the sake of convenience we will identify $\mathbf{P} = \{\mathbf{P}(\cdot|[k])\} \in PM_k^k$ with $\mathbf{p} := (p_1, \ldots, p_k) := (\mathbf{P}(1|[k]), \ldots, \mathbf{P}(k|[k])) \in \Delta_k$. Due to $h(\mathbf{P}) = h(\mathbf{p})$, the set $PM_k^k(\Delta^h)$ is identified with $\Delta_k^h$ this way for any $h \in [0,1]$, and thus an algorithm $\mathcal{A}$ solves $\mathcal{P}_k^{k,\gamma}(\Delta^h)$ for $h \in [0,1]$ iff it fulfills $\mathbb{P}_{\mathbf{p}}(\mathbf{D}(\mathcal{A}) = \mathrm{mode}(\mathbf{p})) \geq 1 - \gamma$ for any $\mathbf{p} \in \Delta_k^h$.

### 4.1 Lower Bounds

Based on Wald's identity, the optimality of the sequential probability ratio test and a result by [14] we are able to prove the following two results, each of which are proven in Section C. In the appendix, we state with Prop. C.1 a more explicit but technical version of Prop. 4.1.

**Proposition 4.1.** *For any $\gamma_0 \in (0, 1/2)$ and $h_0 \in (0, 1)$ there exists a constant $c(h_0, \gamma_0) > 0$ with the following property: Whenever $h \in (0, h_0)$, $\gamma \in (0, \gamma_0)$ and $\mathcal{A}$ is a solution to $\mathcal{P}_k^{k,\gamma}(\Delta^h)$, then*

$$\forall \mathbf{p} \in \Delta_k^h : \mathbb{E}_{\mathbf{p}}[T^{\mathcal{A}}] \geq 2c(h_0, \gamma_0)(h(\mathbf{p}))^{-2} \ln(\gamma^{-1})(1/k + h).$$

*In particular,* $\sup_{\mathbf{p} \in \Delta_k^h} \mathbb{E}_{\mathbf{p}}[T^{\mathcal{A}}] \geq 4c(h_0, \gamma_0)h^{-2} \ln(\gamma^{-1})$.

**Proposition 4.2.** *Let $\gamma \in (0, 1/2)$ be fixed and suppose $\mathcal{A}$ is a solution to $\mathcal{P}_k^{k,\gamma}(\Delta^0)$. Let $\mathbf{p} \in \Delta_k^0$ be arbitrary, $i := \mathrm{mode}(\mathbf{p})$ and $j := \arg\max_{j \in [k] \setminus \{i\}} p_j$. Then, the family $\{\mathbf{p}(h)\}_{h \in (0, p_i - p_j)} \subseteq \Delta_k^0$ defined via $(\mathbf{p}(h))_i := (p_i + p_j + h)/2$, $(\mathbf{p}(h))_j := (p_i + p_j - h)/2$ and $(\mathbf{p}(h))_l := p_l$ for $l \in [k] \setminus \{i, j\}$ fulfills $\mathbf{p}(h) \in \Delta_k^h$ as well as*

$$\limsup_{h \to 0} \mathbb{E}_{\mathbf{p}(h)}[T^{\mathcal{A}}]/(h^{-2} \ln \ln h^{-1}) \geq (1 - 2\gamma)(p_i + p_j) > 0.$$

---

[1]Here, "in exp." means *in expectation*, "w.h.p." means *with prob.* $\geq 1 - \gamma$ and "a.s." *with prob.* 1.

[37] have recently proven a result similar to Proposition 4.1. In contrast to theirs, our bound provides as additional information also the asymptotical behavior as $k \to \infty$. Moreover, our proof is based on the optimality of the Sequential Probability Ratio Test [41, 38] instead of a measure-changing argument [21].

## 4.2   Upper Bounds and Further Prerequisites

To construct a solution $\mathcal{A}$ to $\mathcal{P}_k^{k,\gamma}(\Delta^h)$, we have to decide in a sequential manner at each time $t$, whether we want to make a further query $S_t \in [k]_k$ resulting in a sample $X_t$ or to output an answer $D(\mathcal{A}) \in [k]$. As $[k]_k = \{[k]\}$, we can only choose $S_t = [k]$ in each time step $t$, upon which we observe as feedback $X_t \sim \mathrm{Cat}(\mathbf{p})$, i.e., $\mathbb{P}_\mathbf{p}(X_t = i) = p_i$ for any $i \in [k]$. Having observed $X_1, \ldots, X_t$, a straightforward idea for a prediction $\mathbf{D}(\mathcal{A})$ would be to use the mode of the empirical distribution $\hat{\mathbf{p}}^t := (\hat{p}_1^t, \ldots, \hat{p}_k^t)$ given by $\hat{p}_i^t := \frac{1}{t} \sum_{t' \le t} \mathbf{1}_{\{X_t = i\}}$. As the Dvoretzky-Kiefer-Wolfowitz (DKW) inequality assures us that

$$\mathbb{P}_\mathbf{p} \left( \left\| \hat{\mathbf{p}}^t - \mathbf{p} \right\|_\infty > \varepsilon \right) \le 4 e^{-t\varepsilon^2 / 2} \tag{1}$$

holds for any $\varepsilon > 0$ (Lem. D.1), we can infer that $\hat{\mathbf{p}}^t$ is close to $\mathbf{p}$ with high confidence for large values of $t$. Hence, if $t$ is large enough, predicting the mode of $\hat{\mathbf{p}}^t$ would be the correct prediction for $\mathrm{mode}(\mathbf{p})$ with high probability. In the following we show which choice of $t$ is sufficient to assure a confidence $\ge 1 - \gamma$.

Let us first consider the case $\mathbf{p} \in \Delta_k^h$. It can be shown that

$$(\exists i : \tilde{p}_i - \max_{j \neq i} \tilde{p}_j \ge \varepsilon \text{ and } p_i \neq \max_j p_j) \Rightarrow \|\tilde{\mathbf{p}} - \mathbf{p}\|_\infty \ge (h + \varepsilon)/2$$

holds for any $h \in [0, 1], \varepsilon \in (-h, 1], \mathbf{p} \in \Delta_k^h$ and $\tilde{\mathbf{p}} \in \Delta_k$ (Lemma D.2). This result is optimal in the sense that the term $(h + \varepsilon)/2$ therein cannot be improved (Remark D.3). Choosing $\varepsilon = 0$ and $\tilde{\mathbf{p}} = \hat{\mathbf{p}}^t$ shows us that $\|\hat{\mathbf{p}}^t - \mathbf{p}\|_\infty > h/2$ is necessary for $\mathrm{mode}(\hat{\mathbf{p}}^t) \neq \mathrm{mode}(\mathbf{p})$. Combining this with (1) based on the DKW inequality, we could simply query $S_t = [k]$ for $T = \lceil 8 \ln(4/\gamma) h^{-2} \rceil$ many times and return the mode of $\hat{\mathbf{p}}^T$ as the decision. This (non-sequential) strategy solves $\mathcal{P}_k^{k,\gamma}(\Delta^h)$ and terminates after exactly $\lceil 8 \ln(4/\gamma) h^{-2} \rceil$ time steps (Proposition D.4). Note that according to Prop. 4.1, this strategy is asymptotically optimal.

Next, we intend to solve the more challenging problem $\mathcal{P}_k^{k,\gamma}(\Delta^0)$. Note that any solution to $\mathcal{P}_k^{k,\gamma}(\Delta^0)$ is also a solution to $\mathcal{P}_k^{k,\gamma}(\Delta^h)$ for any $h > 0$, whence Prop. 4.1 shows that $\mathcal{P}_k^{k,\gamma}(\Delta^0)$ cannot be solved by any non-sequential algorithm, i.e., one which decides a priori the number of samples it observes. To construct a solution, we make use of Alg. 1, which also tackles the problem of finding the mode of $\mathbf{p}$ in a non-sequential manner but is allowed to return UNSURE as an indicator that it is not confident enough for its prediction. In other words, the algorithm is allowed to abstain from making a decision. Since

$$\forall i : \tilde{p}_i \le \max_{j \neq i} \tilde{p}_j + h \quad \Rightarrow \quad \|\mathbf{p} - \tilde{\mathbf{p}}\|_\infty \ge h.$$

holds for any $h > 0, \mathbf{p} \in \Delta_k^{3h}$ and $\tilde{\mathbf{p}} \in \Delta_k$ (Lem. D.5), Alg. 1 can be shown to return with probability at least $1 - \gamma$ the correct mode in case $\mathbf{p} \in \Delta_k^{3h}$ is fulfilled. The constraint $\mathbf{p} \in \Delta_k^{3h}$ in the statement above is sharp in the sense that we show in Lem. D.6 for any $h \in (0, 1/8)$ that

$$\inf \left\{ s > 0 \, \middle| \, \forall \mathbf{p} \in \Delta_k^{sh} \, \forall \tilde{\mathbf{p}} \in \Delta_k : (\forall i : \tilde{p}_i \le \max_{j \neq i} \tilde{p}_j + h \Rightarrow \|\mathbf{p} - \tilde{\mathbf{p}}\|_\infty \ge h) \right\} = 3.$$

---

**Algorithm 1** DKW mode-identification with abstention

---

**Input:** $\gamma \in (0, 1), h \in (0, 1)$, access to iid samples $X_t \sim \mathrm{Cat}(\mathbf{p})$

1: $T \leftarrow \lceil 8 \ln(4/\gamma) / h^2 \rceil$
2: Observe samples $X_1, \ldots, X_T$
3: Calculate $\hat{\mathbf{p}}^T = (\hat{p}_1^T, \ldots, \hat{p}_k^T)$ as $\hat{p}_i^T := \frac{1}{T} \sum_{t=1}^T \mathbf{1}_{\{X_t = i\}}, i \in [k]$
4: Choose $i^* \in \mathrm{mode}(\hat{\mathbf{p}}^T)$
5: **if** $\hat{p}_{i^*}^T > \max_{j \neq i^*} \hat{p}_j^T + h$ **then return** $i^*$
6: **else return** UNSURE

---

**Lemma 4.3.** $\mathcal{A} :=$ *Alg. 1 initialized with parameters* $\gamma, h \in (0,1)$ *fulfills* $T^{\mathcal{A}} = \lceil 8 \ln(4/\gamma)/h^2 \rceil$,

$$\forall \mathbf{p} \in \Delta_k \quad : \mathbb{P}_{\mathbf{p}}(\mathbf{D}(\mathcal{A}) \in [k] \text{ and } p_{\mathbf{D}(\mathcal{A})} < \max_{j \in [k]} p_j) \leq \gamma, \tag{2}$$

$$\forall \mathbf{p} \in \Delta_k^0 \quad : \mathbb{P}_{\mathbf{p}}(\mathbf{D}(\mathcal{A}) \in \{\text{mode}(\mathbf{p}), \text{UNSURE}\}) \geq 1 - \gamma, \tag{3}$$

$$\forall \mathbf{p} \in \Delta_k^{3h} : \mathbb{P}_{\mathbf{p}}(\mathbf{D}(\mathcal{A}) = \text{mode}(\mathbf{p})) \geq 1 - \gamma. \tag{4}$$

Lemma 4.3 (proven in Section D) reveals that Alg. 1 has a low failure rate (2) by appropriate choice of $\gamma$, while in turn by an appropriate choice of $h$, namely $h \leq \frac{1}{3}h(\mathbf{p})$, the correct decision will be returned (4) with high probability. However, there are two problems arising: Alg. 1 can also abstain from making a decision (3) and more importantly, the value of $h(\mathbf{p})$ is unknown. As a remedy, we could run Alg. 1 successively with appropriately decreasing choices for $\gamma$ and $h$ until a (real) decision is returned. This approach is followed by Alg. 2 and the following proposition shows that it is indeed a solution to $\mathcal{P}_k^{k,\gamma}(\Delta^0)$; its proof is an adaptation of Lem. 11 in [28] and given in Section D.

---

**Algorithm 2** DKW mode-identification – Solution to $\mathcal{P}_k^{k,\gamma}(\Delta^0)$

---

**Input:** $\gamma \in (0,1)$, sample access to $\text{Cat}(\mathbf{p})$
**Initialization:** $\widetilde{\mathcal{A}} :=$ Alg. 1, $s \leftarrow 1, \forall r \in \mathbb{N} : \gamma_r := \frac{6\gamma}{\pi^2 r^2}, h_r := 2^{-r-1}$
 1: feedback $\leftarrow$ UNSURE
 2: **while** feedback is UNSURE **do**
 3:     feedback $\leftarrow \widetilde{\mathcal{A}}(\gamma_s, h_s, \text{sample access to } \text{Cat}(\mathbf{p}))$
 4:     $s \leftarrow s + 1$
 5: **return** feedback

---

**Proposition 4.4.** $\mathcal{A} :=$ *Alg. 2 initialized with the parameter* $\gamma \in (0,1)$ *solves* $\mathcal{P}_k^{k,\gamma}(\Delta^0)$ *s.t.*

$$\mathbb{P}_{\mathbf{p}}(T^{\mathcal{A}} < \infty) = 1 \quad and \quad \mathbb{P}_{\mathbf{p}}\left(\mathbf{D}(\mathcal{A}) = \text{mode}(\mathbf{p}) \text{ and } T^{\mathcal{A}} \leq t_0(\gamma, h(\mathbf{p}))\right) \geq 1 - \gamma$$

*for any* $\mathbf{p} \in \Delta_k^0$, *where* $t_0(\gamma, h)$ *is mon. decr. w.r.t.* $h$ *with* $t_0(\gamma, h) \in \mathcal{O}\left(h^{-2}\left(\ln \ln h^{-1} + \ln \gamma^{-1}\right)\right)$.

The sample complexity of $\mathcal{A}$ in Proposition 4.4 improves upon the existing alternative solution for $\mathcal{P}_k^{k,\gamma}(\Delta^0)$ in Theorem 2 in [37] with respect to two essential aspects: First, its sample complexity bound is constant instead of increasing in $k$ and second, the dependence on the hardness parameter $h(\mathbf{p})$ is $h(\mathbf{p})^{-2} \ln \ln h(\mathbf{p})^{-1}$ instead of $h(\mathbf{p})^{-2} \ln h(\mathbf{p})^{-1}$.

## 5 Lower Bounds on the General GCW Identification Problem

In this section we provide lower sample complexity bounds for solutions to the GCW identification problem for arbitrary $2 \leq k \leq m$. The following theorem is based on a result by [35], which we state as Thm. B.1 in the appendix.

**Theorem 5.1.** *Any solution* $\mathcal{A}$ *to* $\mathcal{P}_k^{m,\gamma}(\text{PL})$ *fulfills*

$$\sup_{\mathbf{P} \in PM_k^m(\text{PL} \wedge \exists h\text{GCW})} \mathbb{E}_{\mathbf{P}}\left[T^{\mathcal{A}}\right] \in \Omega\left(m(1/k+h)\ln(1/\gamma)/(kh^2)\right). \tag{5}$$

One of the key ingredients for proving Thm. B.1 and thus for Thm. 5.1 is a change-of-measure argument by [21]. By means of the latter technique, we are also able to show the following instance-based as well as worst-case lower bounds for any solution to $\mathcal{P}_k^{m,\gamma}(\Delta^h \wedge \exists \text{GCW})$, which is proven in Section F.

**Theorem 5.2.** *Suppose* $\mathcal{A}$ *solves* $\mathcal{P}_k^{m,\gamma}(\Delta^h \wedge \exists \text{GCW})$ *and let* $\mathbf{P} \in PM_k^m(\Delta^h \wedge \exists \text{GCW})$ *be arbitrary with* $\min_{S \in [m]_k} \min_{j \in S} \mathbf{P}(j|S) > 0$. *For* $S \in [m]_k$ *write* $m_S := \text{mode}(\mathbf{P}(\cdot|S))$ *and for any* $l \in S \setminus \{m_S\}$ *define* $\mathbf{P}^{[l]}(\cdot|S) \in \Delta_S$ *via*

$$\mathbf{P}^{[l]}(l|S) := \mathbf{P}(m_S|S), \quad \mathbf{P}^{[l]}(m_S|S) := \mathbf{P}(l|S), \quad \forall j \in S \setminus \{l, m_S\} : \mathbf{P}^{[l]}(j|S) := \mathbf{P}(j|S).$$

*Then,*

$$\mathbb{E}_{\mathbf{P}}\left[T^{\mathcal{A}}\right] \geq \ln((2.4\gamma)^{-1})/(k-1) \sum_{l \in [m] \setminus \{\text{GCW}(\mathbf{P})\}} \min_{S \in [m]_k : l \in S \setminus \{m_S\}} 1/\text{KL}(\mathbf{P}(\cdot|S), \mathbf{P}^{[l]}(\cdot|S)),$$

*where* $\text{KL}(\mathbf{P}(\cdot|S), \mathbf{P}^{[l]}(\cdot|S))$ *denotes the Kullback-Leibler divergence between two categorical distributions* $X \sim \text{Cat}(\mathbf{P}(\cdot|S))$ *and* $Y \sim \text{Cat}(\mathbf{P}^{[l]}(\cdot|S))$. *Moreover, we have*

$$\sup_{\mathbf{P} \in PM_k^m(\Delta^h \wedge \exists \text{GCW})} \mathbb{E}_{\mathbf{P}}\left[T^{\mathcal{A}}\right] \geq m(1-h^2)\ln((2.4\gamma)^{-1})/(4kh^2).$$

In the special case of dueling bandits ($k = 2$), the instance-dependent lower bound is a novel result,[2] it actually leads to a slightly larger worst-case lower bound than the worst-case bound in Theorem 5.2 for the dueling bandit case (Cor. F.4). For $k \geq 3$, the worst-case bound in Theorem 5.2 is not a consequence of the instance-wise version (Rem. F.3), instead it requires a more involved proof than the latter. For $m = k$, the instance-wise lower bound underlying Prop. 4.1 is apparently larger than that of Thm. 5.2 (Rem. F.5). The reason is that the proof for the instance-wise bound in Theorem 5.2 is tailored to the problem class $PM_k^m(\Delta^h \wedge \exists \mathrm{GCW})$ and consequently has to deal with combinatorial issues arising in case $k < m$.

## 6   Upper Bounds on the General GCW Identification Problem

In [35] the PAC-WRAPPER algorithm is introduced, which is an algorithm able to identify the GCW under the Plackett-Luce assumption with (up to logarithmic terms) optimal instance-wise sample complexity, see Section B. By translating the sample complexity result of PAC-WRAPPER into our setting, we obtain the following result (see Section B for its proof), which is also by Thm. 5.1 suggested to be optimal up to logarithmic factors.

**Theorem 6.1.** *There exists a solution $\mathcal{A}$ to $\mathcal{P}_k^{m,\gamma}(\mathrm{PL} \wedge \exists \mathrm{GCW}^*)$ s.t.*

$$\inf\nolimits_{\mathbf{P} \in PM_k^m(\mathrm{PL} \wedge \exists h\mathrm{GCW})} \mathbb{P}_{\mathbf{P}}\left(\mathbf{D}(\mathcal{A}) \in \mathrm{GCW}(\mathbf{P}) \text{ and } T^{\mathcal{A}} \leq t'(m,h,k,\gamma)\right) \geq 1 - \gamma$$

*holds with $t'(h,m,k,\gamma) \in \mathcal{O}\left({}^{m(1/k+h)\ln(k/\gamma\ln(1/h))}/_{(kh^2)}\right)$.*

---

**Algorithm 3** DVORETZKY–KIEFER–WOLFOWITZ TOURNAMENT – Solution to $\mathcal{P}_k^{m,\gamma}(\exists \mathrm{GCW} \wedge \Delta^0)$

---

**Input:** $k, m \in \mathbb{N}, \gamma \in (0,1)$, sample access to $\mathbf{P} = \{\mathbf{P}(\cdot|S)\}_{S \in [m]_k}$
**Initialization:** $\widetilde{\mathcal{A}} := \mathrm{Alg.}\ 2$, choose $S_1 \in [m]_k$ arbitrary, $F_1 \leftarrow [m], \gamma' \leftarrow \frac{\gamma}{\lceil m/(k-1) \rceil}, s \leftarrow 1$

    ▷ $S_s$ : candidates in round $s$,   $F_s$ : remaining elements in round $s$,   $i_s$ : output of $\widetilde{\mathcal{A}}$ in round $s$
1: **while** $s \leq \lceil \frac{m}{k-1} \rceil - 1$ **do**
2:    $i_s \leftarrow \widetilde{\mathcal{A}}(\gamma', \text{sample access to } \mathbf{P}(\cdot|S_s))$
3:    $F_{s+1} \leftarrow F_s \setminus S_s$
4:    Write $F_{s+1} = \{j_1, \ldots, j_{|F_{s+1}|}\}$.
5:    **if** $|F_{s+1}| < k$ **then**
6:        Fix distinct $j_{|F_{s+1}|+1}, \ldots, j_{k-1} \in [m] \setminus F_{s+1}$.
7:    $S_{s+1} \leftarrow \{i_s, j_1, \ldots, j_{k-1}\}$
8:    $s \leftarrow s + 1$
9: $i_s \leftarrow \widetilde{\mathcal{A}}(\gamma', \text{sample access to } \mathbf{P}(\cdot|S_s))$
10: **return** $i_s$

---

Next, we consider the problem class $\mathcal{P}_k^{m,\gamma}(\exists \mathrm{GCW} \wedge \Delta^0)$, for which we propose the DVORETZKY–KIEFER–WOLFOWITZ TOURNAMENT (DKWT) algorithm (see Alg. 3). DKWT is a simple round-based procedure eliminating in each round those arms from a candidate set of possible GCWs that have been discarded by Alg. 2 with high confidence as being the GCW. In the following theorem we derive theoretical guarantees for DKWT, while a more sophisticated sample complexity bound is provided in Thm. E.1.

**Theorem 6.2.** $\mathcal{A} := \mathrm{DKWT}$ *initialized with the parameter $\gamma \in (0,1)$ solves $\mathcal{P}_k^{m,\gamma}(\exists \mathrm{GCW} \wedge \Delta^0)$ s.t.*

$$\mathbb{P}_{\mathbf{P}}\left(\mathbf{D}(\mathcal{A}) \in \mathrm{GCW}(\mathbf{P}) \text{ and } T^{\mathcal{A}} \leq T'(h(\mathbf{P}), m, k, \gamma)\right) \geq 1 - \gamma$$

*holds for all $\mathbf{P} \in PM_k^m(\exists \mathrm{GCW} \wedge \Delta^0)$ with $T'(h,m,k,\gamma) \in \mathcal{O}\left(\frac{m}{kh^2} \ln\left(\frac{m}{k}\right)\left(\ln\ln\frac{1}{h} + \ln\frac{1}{\gamma}\right)\right)$.*

The result stated in Table 2 for $(\mathrm{X}) = (\exists \mathrm{GCW} \wedge \Delta^0)$ and $(\mathrm{Y}) = (\Delta^h)$ follows from this by noting that $h(\mathbf{P}) \geq h$ holds for any $\mathbf{P} \in PM_k^m(\exists h\mathrm{GCW} \wedge \Delta^h)$. Regarding Prop. 4.2, the additional factor

---

[2]So far, existing lower sample complexity bounds for solutions to $\mathcal{P}_2^{m,\gamma}(\Delta^h \wedge \exists \mathrm{GCW})$ are either restricted to worst-case scenarios [6] or to the special case where $\mathbf{P}$ belongs to a Thurstone model [29] or a Plackett-Luce model [35].

Table 3: Comparison of DKWT with PAC-WRAPPER (PW) on $\boldsymbol{\theta} = (1, 0.8, 0.6, 0.4, 0.2)$

| | $T^{\mathcal{A}}$ | | Accuracy | |
|---|---|---|---|---|
| $k$ | DKWT | PW | DKWT | PW |
| 3 | **44293** (3695.6) | 1631668498 (1453661392.0) | 1.0 | 1.0 |
| 4 | **32427** (2516.2) | 263543687 (127401593.7) | 1.0 | 1.0 |

$\ln \ln h^{-1}$ in the upper bounds from Thm. 6.1 and Thm. 6.2 appears indispensable. Since $PM_k^m(\text{PL} \wedge \exists \text{GCW}^*) \not\subseteq PM_k^m(\exists \text{GCW} \wedge \Delta^0)$ and $PM_k^m(\text{PL} \wedge \exists \text{GCW}^*) \not\subseteq PM_k^m(\exists \text{GCW} \wedge \Delta^0)$ hold, a solution to $\mathcal{P}_k^{m,\gamma}(\text{PL} \wedge \exists \text{GCW}^*)$ is in general not comparable with a solution to $\mathcal{P}_k^{m,\gamma}(\exists \text{GCW} \wedge \Delta^0)$, i.e., neither Thm. 6.1 nor Thm. 6.2 implies the other one.

Replacing $\exists \text{GCW} \wedge \Delta^0$ with the more restrictive assumption $\exists h\text{GCW} \wedge \Delta^0$ (as an assumption on $\mathbf{P}$) makes the GCW identification task much easier. This is similar to the case of $\mathcal{P}_k^{k,\gamma}(\Delta^h)$ and $\mathcal{P}_k^{k,\gamma}(\Delta^0)$ discussed in Section 4.2. For $\mathcal{P}_k^{m,\gamma}(\exists h\text{GCW} \wedge \Delta^0)$ we can modify Alg. 3 in order to incorporate the knowledge of $h$ as follows: choose in round $s$ a query set $S_s \subseteq F_s$ (filled up with $|F_s| - k$ further elements from $[m] \setminus F_s$ if $|F_s| < k$) and execute Alg. 1 with parameters $\frac{h}{3}$, $\frac{\gamma}{\lceil m/(k-1) \rceil}$ and sample access to $\mathbf{P}(\cdot | S_s)$. In case Alg. 1 returns as decision an element $i \in S_s$, we let $F_{s+1} = F_s \setminus (S_s \setminus \{i\})$, and otherwise $F_{s+1} = F_s$. Then we proceed with the next round $s + 1$. We repeat this procedure until $|F_s| = 1$, and return the unique element in $F_s$ as the prediction for the GCW. In Sec. E we provide detailed a pseudocode for this algorithm (Alg. 5) and show that it indeed solves $\mathcal{P}_k^{m,\gamma}(\exists h\text{GCW} \wedge \Delta^0)$ with the guarantee that it terminates almost surely for any $\mathbf{P} \in PM_k^m(\exists h\text{GCW} \wedge \Delta^0)$ before some time $t'(m, k, h, \gamma) \in \mathcal{O}\left( \frac{m \ln(m/(k\gamma))}{kh^2} \right)$ (see Thm. E.2). A look at Thm. 5.2 reveals that this solution to $\mathcal{P}_k^m(\exists h\text{GCW} \wedge \Delta^0)$ is asymptotically optimal up to logarithmic factors in a worst-case sense w.r.t. $PM_k^m(\exists h\text{GCW} \wedge \Delta^0)$.

# 7 Empirical Evaluation

In the following, we present experimental results on the performance of our GCW identification solution.[3] We restrict ourselves in the main paper to DKWT, which is our solution of the most general problem $\mathcal{P}_k^{m,\gamma}(\exists h\text{GCW} \wedge \Delta^0)$. Throughout all experiments, if not specified differently in the pseudocode, every choice of an element within a specific set made by DKWT is performed uniformly at random. All experiments were conducted on a machine with an Intel® Core™ i7-4700MQ Processor, executing all experiments (including those in the supplemental material) with only one CPU core in use took less than 72 hours.

At first, we compare DKWT with PAC-WRAPPER (PW), which is the solution to $\mathcal{P}_k^{m,\gamma}(\text{PL})$ in [35] underlying Thm. 6.1 and so far the only solution in the literature to the best of our knowledge for identifying the GCW in multi-dueling bandits with an error probability at most $\gamma$. Table 3 shows the results of both algorithms when started on an instance $\mathbf{P} \in PM_k^5(\text{PL})$ with underlying PL-parameter $\boldsymbol{\theta} = (1, 0.8, 0.6, 0.4, 0.2)$ and $\gamma = 0.05$, for different values of $k$. The observed termination time $T^{\mathcal{A}}$, the corresponding standard error (in brackets) and the accuracy are averaged over 10 repetitions.

Both algorithms achieve the desired accuracy $\geq 95\%$ in every case, but DKWT requires far less samples than PW to find the GCW. Further experiments in the appendix (cf. Section G) demonstrate the superiority of DKWT over PW also for other values of $m$, $\boldsymbol{\theta}$ and $\gamma$, including the problem instances considered in [35]. Note that the observed extremely large sample complexity of PW appears to be consistent with the experimental results in [35] and is supposedly caused by multiple runs of a costly procedure PAC-BEST-ITEM, which does not exploit the DKW inequality but is rather based on applications of Chernoff's bound.

In the case $k = 2$, the GCW identification problem coincides with the Condorcet winner (CW) identification problem in dueling bandits. Thus, we can compare DKWT to state-of-the art solutions for finding the CW if it exists: SELECT [26], SEEBS[4] [29] and EXPLORE-THEN-VERIFY (EtV) [20]. Formally, SELECT requires $h \in (0, 1)$ as a parameter as it solves $\mathcal{P}_2^{m,\gamma}(\exists \text{GCW} \wedge \Delta^h)$,

---

[3] Our implementation is provided at `https://github.com/bjoernhad/GCWidentification`.

[4] We include SEEBS even though it technically requires $\mathbf{P}$ to fulfill *strong stochastic transitivity* and the *stochastic triangle inequality*, cf. Sec. G.2.

Table 4: Comparison of DKWT, SEEBS and EXPLORE-THEN-VERIFY (EtV)

| | | $T^{\mathcal{A}}$ | | |
|---|---|---|---|---|
| $m$ | $h$ | DKWT | SEEBS | EtV |
| 5 | 0.20 | **6010** (293.2) | 7305 (432.1) | 8601 (589.2) |
| 5 | 0.15 | **8874** (460.0) | 13393 (904.5) | 11899 (986.9) |
| 5 | 0.10 | **15769** (1457.1) | 19802 (1543.2) | 260171 (210678.1) |
| 5 | 0.05 | **31454** (4127.4) | 36855 (3533.2) | 156534 (115903.1) |
| 10 | 0.20 | **14334** (492.8) | 16956 (617.9) | 26115 (969.2) |
| 10 | 0.15 | **18563** (734.5) | 27527 (1126.7) | 32548 (2514.6) |
| 10 | 0.10 | **33040** (1625.1) | 47330 (2138.2) | 68858 (11304.5) |
| 10 | 0.05 | **78660** (6517.2) | 83877 (5842.6) | 220098 (92484.9) |

while DKWT, SEEBS and EtV solve the more challenging problem $\mathcal{P}_2^{m,\gamma}(\exists\text{GCW} \wedge \Delta^0)$. As a consequence, we compare here only the latter three algorithms on probability models **P** sampled uniformly at random from $PM_k^m(\exists\text{GCW} \wedge \Delta^h)$ for various values of $h$ *without* providing these algorithms with the explicit value of $h$. We also compare SELECT with the three considered algorithms in Section G. Without great surprise, it turns out that SELECT has a much smaller sample complexity due to its advantage of knowing the explicit value of $h$.

Table 4 reports the observed sample complexities, together with the standard errors in brackets, obtained for $\gamma = 0.05$ and different choices of $m$ and $h$, the numbers are averaged over 100 repetitions. Every algorithm achieves an accuracy of 1 in each case. DKWT clearly outperforms SEEBS and EtV in any case, which is consistent with similar results for larger values of $m$ in the appendix. Overall, these results show that DKWT is also well suited for the dueling bandit case.

We complement our empirical study in Section G.3 with a comparison of DKWT with Alg. 5 showing that the latter outperforms the former in the case where $h(\mathbf{P})$ is small and $\mathbf{P} \in PM_k^{m,\gamma}(\exists h'\text{GCW} \wedge \Delta^0)$ for some a priori known $h' > h(\mathbf{P})$.

## 8   Conclusion

We investigated the sample complexity required for identifying the generalized Condorcet winner (GCW) in multi-dueling bandits within a fixed confidence setting. We provided lower bound results, which as a special case yield a novel instance-wise lower sample complexity bound for identifying the Condorcet winner in the realm of dueling bandits. We introduced DVORETZKY-KIEFER-WOLFOWITZ TOURNAMENT (DKWT), an algorithmic solution to the GCW identification task with asymptotically nearly optimal worst-case sample complexity. In our experiments, DKWT outperformed competing state-of-the art algorithms, even in the special case of dueling bandits. Last but not least, we pointed out that and to which extent incorporating a Plackett-Luce assumption on the feedback mechanism makes the GCW identification problem asymptotically easier w.r.t. the worst-case required sample complexity.

There are several directions in which this work could be extended. First, one could investigate the GCW identification problem in the so-called *probably approximately correct* (PAC) setting and search not for the GCW but instead for an arm that outperforms any other arm only with some margin $\varepsilon > 0$. Secondly, one may generalize this problem to the identification of the GCW *without* assuming its existence, where one has to check on-the-fly whether a GCW exists, i.e., a testification (*test* and *identify*) problem as in [19] in the dueling bandit case for the Condorcet Winner. Moreover, one may also extend our problem to the case where query sets *up to size k* are allowed at each time step. This variant has already been discussed in a regret minimization scenario [32, 1] or a PAC setting [33].

## Acknowledgments and Disclosure of Funding

The authors gratefully acknowledge financial support by the German Research Foundation (DFG - project number 317046553).

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
