# A  Relationships between the Probability Models

**Lemma A.1.** *For any $k, m \in \mathbb{N}$ and $h \in (0, 1)$ we have the implications*

$$PM_k^m(\exists h\mathrm{GCW}) \subsetneq PM_k^m(\exists \mathrm{GCW}^*) \subsetneq PM_k^m(\exists \mathrm{GCW}),$$
$$PM_k^m(\Delta^h) \subsetneq PM_k^m(\Delta^0),$$
$$PM_k^m(\mathrm{PL}) \subsetneq PM_k^m(\exists \mathrm{GCW}),$$
$$PM_k^m(\Delta^h) \cap PM_k^m(\exists \mathrm{GCW}) \subsetneq PM_k^m(\exists h\mathrm{GCW}),$$
$$PM_k^m(\Delta^0) \cap PM_k^m(\exists \mathrm{GCW}) \subsetneq PM_k^m(\exists \mathrm{GCW}^*).$$

*Proof.* This is a direct consequence of the definitions. □

# B  GCW Identification under the Plackett-Luce Assumption

In this section, we prove the lower and upper bounds of solutions to the GCW identification problem under the Plackett-Luce assumption stated in Theorems 5.1 and 6.1. For $\boldsymbol{\theta} \in (0, \infty)^m$ we denote by $\mathbf{P}(\boldsymbol{\theta}) \in PM_k^m(\mathrm{PL})$ the corresponding PM, which is consistent with the Plackett-Luce model with parameter $\boldsymbol{\theta}$ on $S_m$, i.e., $\mathbf{P}(\boldsymbol{\theta}) = \{\mathbf{P}(\boldsymbol{\theta})(\cdot|S)\}_{S \in [m]_k}$ is defined via

$$\mathbf{P}(\boldsymbol{\theta})(i|S) := \frac{\theta_i}{\sum_{a \in S} \theta_a} \quad \text{for any } S \in [m]_k \text{ and } i \in S.$$

As $\mathbf{P}(x\boldsymbol{\theta}) = \mathbf{P}(\boldsymbol{\theta})$ holds for any $x > 0$ and $\boldsymbol{\theta} \in (0, \infty)^m$, we may restrict ourselves w.l.o.g. to those $\mathbf{P}(\boldsymbol{\theta})$ with $\max_{i \in [m]} \theta_i = 1$.

In [35], the following lower resp. upper sample complexity bounds for solutions to $\mathcal{P}_k^{m,\gamma}(\mathrm{PL})$ resp. $\mathcal{P}_k^{m,\gamma}(\mathrm{PL} \wedge \exists \mathrm{GCW}^*)$ depending on the ground-truth Plackett-Luce parameter have been proven.

**Theorem B.1.** *Any solution $\mathcal{A}$ to $\mathcal{P}_k^{m,\gamma}(\mathrm{PL})$ fulfills*

$$\mathbb{E}_{\mathbf{P}(\boldsymbol{\theta})}\left[T^{\mathcal{A}}\right] \in \Omega\left(\max\left(\sum_{j=2}^{m} \frac{\theta_j}{(1-\theta_j)^2} \ln \frac{1}{\gamma}, \frac{m}{k} \ln \frac{1}{\gamma}\right)\right)$$

*for any $\boldsymbol{\theta} \in (0,1]^m$ with $1 = \theta_1 > \max_{j \geq 2} \theta_j$.*

*Proof.* Confer Theorem 7 in [35]. □

**Theorem B.2.** *There is a solution $\mathcal{A}$ to $\mathcal{P}_k^{m,\gamma}(\mathrm{PL} \wedge \exists \mathrm{GCW}^*)$, which fulfills for any $\boldsymbol{\theta} \in (0,1]^m$ with $1 = \theta_1 > \max_{j \geq 2} \theta_j$ the estimate*

$$\mathbb{P}_{\mathbf{P}(\boldsymbol{\theta})}\left(\mathbf{D}(\mathcal{A}) \in \mathrm{GCW}(\mathbf{P}) \text{ and } T^{\mathcal{A}} \leq t'(\boldsymbol{\theta}, k, \gamma)\right) \geq 1 - \gamma$$

*with*

$$t'(\boldsymbol{\theta}, k, \gamma) \in \mathcal{O}\left(\frac{\Theta_{[k]}}{k} \sum_{j=2}^{m} \frac{1}{(1-\theta_j)^2} \ln\left(\frac{k}{\gamma} \ln\left(\frac{1}{1-\theta_j}\right)\right)\right)$$

*and $\Theta_{[k]} := \max_{S \in [m]_k} \sum_{a \in S} \theta_a$.*

*Proof.* Confer Theorem 3 in [35] and note that $\min_{j \geq 2}(1 - \theta_j)^{-2} \geq 1$ holds for any $\boldsymbol{\theta} \in (0,1]^m$ with $1 = \theta_1 > \max_{j \geq 2} \theta_j$. □

To translate the preceding results into our setting, we need a better understanding of the set $PM_k^m(\mathrm{PL} \wedge \exists h\mathrm{GCW})$. This is achieved by means of the following observation. For the sake of completeness, we also provide a characterization of $PM_k^m(\mathrm{PL} \wedge \Delta^h)$.

**Lemma B.3.** *For $\boldsymbol{\theta} \in (0, \infty)^m$ with $\theta_1 \geq \cdots \geq \theta_m$ we have*

$$\mathbf{P}(\boldsymbol{\theta}) \in PM_k^m(\exists h\mathrm{GCW}) \Leftrightarrow \forall j \in \{2, \ldots, k\} : h(\theta_1 + \cdots + \theta_k) + \theta_j - \theta_1 \leq 0$$
$$\Leftrightarrow h(\theta_1 + \cdots + \theta_k) + \theta_2 - \theta_1 \leq 0$$

*and*

$$\mathbf{P}(\boldsymbol{\theta}) \in PM_k^m(\Delta^h) \Leftrightarrow \forall i \in [m-k] : h(\theta_i + \cdots + \theta_{i+k-1}) + \theta_{i+1} - \theta_i \leq 0.$$

*Proof.* This follows directly from the definitions. $\qquad\square$

From this, we obtain the following result, which is not explicitly needed anywhere but rather stated for the sake of completeness.

**Corollary B.4.** *For any $h \in (0,1)$ and $m, k \in \mathbb{N}$ with $k \leq m$ we have $PM_k^m(\mathrm{PL} \wedge \exists h\mathrm{GCW}) \supseteq PM_k^m(\mathrm{PL} \wedge \Delta^h) \neq \emptyset$.*

*Proof.* Note that $PM_k^m(\mathrm{PL} \wedge \exists h\mathrm{GCW}) \supseteq PM_k^m(\mathrm{PL} \wedge \Delta^h)$ is a direct consequence from the definitions. To see $PM_k^m(\mathrm{PL} \wedge \Delta^h) \neq \emptyset$ we fix $x > 1$ with $h + \frac{h}{x} \leq 1$ and define $\boldsymbol{\theta} \in (0,1]^m$ via $\theta_j := \frac{h^j}{(kx)^j}$ for any $j \in [m]$. Then,

$$
\begin{aligned}
&h(\theta_i + \cdots + \theta_{i+k-1}) + \theta_{i+1} - \theta_i \\
&= \frac{h^{i+1}}{(kx)^i} + \left( \frac{h^{i+2}}{(kx)^{i+1}} + \cdots + \frac{h^{i+k}}{(kx)^{i+k-1}} + \frac{h^{i+1}}{(kx)^{i+1}} \right) - \frac{h^i}{(kx)^i} \\
&\leq \frac{h^{i+1}}{(kx)^i} + \frac{kh^{i+1}}{(kx)^{i+1}} - \frac{h^i}{(kx)^i} = \frac{h^i}{(kx)^i}\left( h + \frac{h}{x} - 1 \right) \leq 0
\end{aligned}
$$

holds for any $i \in [m-k]$ and thus $\mathbf{P}(\boldsymbol{\theta}) \in PM_k^m(\mathrm{PL} \wedge \Delta^h)$ follows from Lemma B.3. $\qquad\square$

*Proof of Theorem 5.1.* Define $\boldsymbol{\theta} \in (0,1]^m$ via $\theta_1 := 1$ and $\theta_j := \frac{1-h}{h(k-1)+1}$ for $2 \leq j \leq m$. Then,

$$
\begin{aligned}
h\sum_{j=1}^k \theta_j + \theta_2 - \theta_1 &= h\left( 1 + \frac{(k-1)(1-h)}{h(k-1)+1} \right) + \frac{1 - h - h(k-1) - 1}{h(k-1)+1} \\
&= \frac{h(h(k-1) + 1 + (k-1)(1-h)) - hk}{h(k-1)+1} = 0
\end{aligned}
$$

shows with regard to Lemma B.3 that $\mathbf{P}(\boldsymbol{\theta}) \in PM_k^m(\exists h\mathrm{GCW})$ is fulfilled. Moreover, for $j \in \{2, \ldots, m\}$ we have $1 - \theta_j = \frac{hk}{h(k-1)+1}$ and thus

$$
\frac{\theta_j}{(1-\theta_j)^2} = \frac{(h(k-1)+1)(1-h)}{h^2 k^2} = \frac{hk(1-h) + (1-h)^2}{h^2 k^2},
$$

which is in $\Theta(\frac{1}{hk} + \frac{1}{h^2 k^2}) = \Theta\left( \frac{1}{kh^2}\left( \frac{1}{k} + h \right) \right)$, since $1 - h \in \Theta(1)$ as $h \searrow 0$. In particular,

$$
\sum_{j=2}^m \frac{\theta_j}{(1-\theta_j)^2} \in \Theta\left( \frac{m}{kh^2}\left( \frac{1}{k} + h \right) \right)
$$

and thus the statement follows from Theorem B.1. $\qquad\square$

*Proof of Theorem 6.1.* Suppose $\gamma \in (0,1)$, $h \in (0,1)$ and $m, k \in \mathbb{N}_{\geq 2}$ with $k \leq m$ to be arbitrary but fixed for the moment and let $\mathcal{A}$ be the solution to $\mathcal{P}_k^{m,\gamma}(\mathrm{PL} \wedge \exists \mathrm{GCW}^*)$ from Theorem B.2. For $l \in \{2, \ldots, k\}$ define $g_l : [0,1]^m \to \mathbb{R}$ via $g_l(\boldsymbol{\theta}) := h(1 + \theta_2 + \cdots + \theta_k) + \theta_l - 1$ and denote by $\mathfrak{B}$ the set

$$
\{\boldsymbol{\theta} \in (0,1]^m \mid 1 = \theta_1 > \theta_2 \geq \cdots \geq \theta_m \text{ and } \forall l \in \{2, \ldots, k\} : g_l(\boldsymbol{\theta}) \leq 0\}.
$$

According to Lemma B.3, any $\mathbf{P} \in PM_k^m(\mathrm{PL})$ with $\mathrm{GCW}(\mathbf{P}) = 1$ fulfills $\mathbf{P} \in PM_k^m(\exists h\mathrm{GCW})$ iff $\mathbf{P} = \mathbf{P}(\boldsymbol{\theta})$ for some $\boldsymbol{\theta} \in \mathfrak{B}$. Consequently, it is with regard to Theorem B.2 sufficient to show that

$$
\frac{\Theta_{[k]}}{k} \sum_{j=2}^m \frac{1}{(1-\theta_j)^2} \ln\left( \frac{k}{\gamma} \ln\left( \frac{1}{1-\theta_j} \right) \right) \leq \frac{6m}{kh^2}\left( \frac{1}{k} + h \right) \ln\left( \frac{k}{\gamma} \ln(h^{-1}) \right) \tag{6}
$$

holds for any $\boldsymbol{\theta} \in \mathfrak{B}$. We prove this in several steps.

**Claim 1:** For any $\boldsymbol{\theta} \in \mathfrak{B}$ we have

$$
\sum_{j=2}^k \frac{1 + \theta_2 + \cdots + \theta_k}{(1-\theta_j)^2} \leq \frac{3(1+hk)}{h^2}. \tag{7}
$$

**Proof of Claim 1:** Let $\mathfrak{B}'$ be the set of all $\boldsymbol{\theta} = (1, \theta_2, \ldots, \theta_k)$ with $1 \geq \theta_2 \geq \cdots \geq \theta_k \geq 0$ and $g_l(\boldsymbol{\theta}) \leq 0$ for all $l \in \{2, \ldots, k\}$. As $(1, \theta_2, \ldots, \theta_k) \in \mathfrak{B}'$ holds for any $(1, \theta_2, \ldots, \theta_m) \in \mathfrak{B}$, it is sufficient to show that (7) holds for any $\boldsymbol{\theta} = (1, \theta_2, \ldots, \theta_k) \in \mathfrak{B}'$.

**Claim 1a:** For any $\boldsymbol{\theta} \in \mathfrak{B}'$ and $l \in \{2, \ldots, k\}$ we have $\theta_l \leq 1 - h$.
**Proof:** For $\boldsymbol{\theta} = (1, \theta_2, \ldots, \theta_k) \in \mathfrak{B}'$ and $l \in \{2, \ldots, k\}$ we have

$$0 \geq g_l(\boldsymbol{\theta}) = h(1 + \theta_2 + \cdots + \theta_k) + \theta_l - 1 \geq h + \theta_l - 1,$$

and thus $\theta_l \leq 1 - h$. ♣

According to Claim 1a, $\mathfrak{B}'$ is a compact subset of $\{1\} \times [0, 1 - h]^{k-1}$. Consequently, the continuous function $f : \mathfrak{B}' \to \mathbb{R}$, $f(\boldsymbol{\theta}) := \sum_{j=2}^{k} \frac{1 + \theta_2 + \cdots + \theta_k}{(1 - \theta_j)^2}$ is well-defined and takes its maximum on $\mathfrak{B}'$ in a point $\boldsymbol{\theta}^* \in \mathfrak{B}'$.

**Claim 1b:** There is some $j \in \{2, \ldots, k\}$ s.t. $g_2(\boldsymbol{\theta}^*) = \cdots = g_j(\boldsymbol{\theta}^*) = 0$ and $\theta_{j+2}^* = \cdots = \theta_k^* = 0$.
**Proof:** To show indirectly the existence of some $j \in \{2, \ldots, k\}$ with $g_j(\boldsymbol{\theta}^*) = 0$ assume on the contrary that $g_l(\boldsymbol{\theta}^*) < 0$ for any $l \in \{2, \ldots, k\}$. Then, if $\varepsilon > 0$ is small enough, $\boldsymbol{\theta}_\varepsilon := (1, \theta_2^* + \varepsilon, \theta_3^*, \ldots, \theta_k^*)$ is an element of $\mathfrak{B}'$. Since

$$\frac{\partial f}{\partial \theta_2}(\boldsymbol{\theta}) = \frac{2\theta_2(1 + \theta_2 + \cdots + \theta_k)}{(1 - \theta_2)^3} + \sum_{l=2}^{k} \frac{1}{(1 - \theta_l)^2} > 0$$

holds for any $\boldsymbol{\theta}$ in the interior of $\mathfrak{B}'$, we would obtain $f(\boldsymbol{\theta}_\varepsilon) > f(\boldsymbol{\theta}^*)$ in contradiction to the optimality of $\boldsymbol{\theta}^*$. Hence, there has to be a $j \in \{2, \ldots, k\}$ with $g_j(\boldsymbol{\theta}^*) = 0$. In case $j \geq 3$, we may infer from $g_{j-1}(\boldsymbol{\theta}^*) - g_j(\boldsymbol{\theta}^*) = \theta_{j-1}^* - \theta_j^* \geq 0$ inductively $0 = g_{j-1}(\boldsymbol{\theta}^*) = \cdots = g_2(\boldsymbol{\theta}^*)$.
It remains to prove $\theta_{j+2}^* = \cdots = \theta_k^* = 0$. Assume this was not the case, i.e., $j \leq k - 2$ and $j' := \max\{l \in \{2, \ldots, k\} \mid \theta_l^* > 0\} \geq j + 2$. By definition of $j$ we have $g_j(\boldsymbol{\theta}^*) < 0$. Consequently,

$$\boldsymbol{\theta}'_\varepsilon := (1, \theta_2^*, \ldots, \theta_j^*, \theta_{j+1}^* + \varepsilon, \theta_{j+2}^*, \ldots, \theta_{j'}^* - \varepsilon, 0, \ldots, 0)$$

is for small values of $\varepsilon \geq 0$ an element of $\mathfrak{B}'$. Using $\sum_{l=2}^{k}(\boldsymbol{\theta}'_\varepsilon)_k = \sum_{l=2}^{k} \theta_l^*$ we see that

$$\frac{\mathrm{d}}{\mathrm{d}\varepsilon} f(\boldsymbol{\theta}'_\varepsilon) = \frac{2}{(1 - \theta_{j+1}^* - \varepsilon)^3} - \frac{2}{(1 - \theta_{j'}^* + \varepsilon)^3},$$

which is due to $\theta_{j+1}^* \geq \theta_{j'}^*$ positive for small values of $\varepsilon > 0$. In particular, $f(\boldsymbol{\theta}'_\varepsilon) > f(\boldsymbol{\theta}'_0) = f(\boldsymbol{\theta}^*)$ holds for small $\varepsilon > 0$, which contradicts the optimality of $\boldsymbol{\theta}^*$. This completes the proof of Claim 1b. ♣

According to Claim 1b we may fix some $j \in \{2, \ldots, k\}$ with $g_2(\boldsymbol{\theta}^*) = \cdots = g_j(\boldsymbol{\theta}^*) = 0$ and $\theta_{j+2}^* = \cdots = \theta_k^* = 0$. Since $g_l(\boldsymbol{\theta}^*) - g_{l'}(\boldsymbol{\theta}^*) = \theta_l^* - \theta_{l'}^* = 0$ holds for any $l, l' \in \{2, \ldots, k\}$, we have $\theta_2^* = \cdots = \theta_j^*$. From $0 \geq g_2(\boldsymbol{\theta}^*) \geq h(1 + (j-1)\theta_2^*) + \theta_2^* - 1$ we infer

$$\theta_2^* = \cdots = \theta_j^* \leq \frac{1 - h}{1 + (j-1)h} = 1 - \frac{hj}{1 + h(j-1)}.$$

Together with $\theta_j^* \geq \theta_{j+1}^* \geq 0 = \theta_{j+2}^* = \cdots = \theta_k^*$ we obtain

$$
\begin{aligned}
\frac{1 + \theta_2^* + \cdots + \theta_k^*}{(1 - \theta_2^*)^2} &\leq \frac{1 + j\theta_2^*}{(1 - \theta_2^*)^2} \leq \frac{(1 + h(j-1))^2}{h^2 j^2}\left(1 + \frac{j(1-h)}{1 + h(j-1)}\right) \\
&= \frac{(1 + h(j-1))(1 - h + j)}{h^2 j^2} \leq 2\left(\frac{1}{h^2 j} + \frac{h(j-1)}{h^2 j}\right) \\
&\leq \frac{2}{h^2}\left(\frac{1}{j} + h\right),
\end{aligned}
$$

where we have used that $1 - h + j \leq 2j$ holds trivially. Combining this with the fact that $g_2(\boldsymbol{\theta}^*) \leq 0$ implies $(1 + \theta_2^* + \cdots + \theta_k^*) \leq \frac{1-\theta_2^*}{h} \leq \frac{1}{h}$ yields

$$
\begin{aligned}
f(\boldsymbol{\theta}^*) &= \sum_{l=2}^{k} \frac{1 + \theta_2^* + \cdots + \theta_k^*}{(1 - \theta_l^*)^2} \\
&\leq (1 + \theta_2^* + \cdots + \theta_k^*) \left( \sum_{l=2}^{j+1} \frac{1}{(1 - \theta_2^*)^2} + \sum_{l=j+2}^{k} 1 \right) \\
&\leq \frac{2j}{h^2} \left( \frac{1}{j} + h \right) + \frac{k - j - 1}{h} \leq \frac{3(1 + hk)}{h^2}.
\end{aligned}
$$

Since $\boldsymbol{\theta}^*$ was a maximum point of $f$ in $\mathfrak{B}'$, Claim 1 follows. $\blacksquare$

**Claim 2:** For any $\boldsymbol{\theta} \in \mathfrak{B}$ we have $\sum_{j=2}^{m} \frac{1}{(1-\theta_j)^2} \leq \frac{m-1}{k-1} \sum_{j=2}^{k} \frac{1}{(1-\theta_j)^2}$.

**Proof of Claim 2:** Using $1 \geq \theta_2 \geq \cdots \geq \theta_m$, this follows directly from comparing the $(m-1)(k-1)$ summands in $(k-1) \sum_{j=2}^{m} \frac{1}{(1-\theta_j)^2} = \sum_{j=2}^{m} \frac{1}{(1-\theta_j)^2} + \cdots + \sum_{j=2}^{m} \frac{1}{(1-\theta_j)^2}$ with those in $(m-1) \sum_{j=2}^{k} \frac{1}{(1-\theta_j)^2}$. $\blacksquare$

**Claim 3:** Inequality (6) holds for any $\boldsymbol{\theta} \in \mathfrak{B}$.

**Proof of Claim 3:** Let $\boldsymbol{\theta} \in \mathfrak{B}$ be fixed and note that $\Theta_{[k]} = 1 + \theta_2 + \cdots + \theta_k$ holds. From $1 \geq \theta_2 \geq \cdots \geq \theta_m \geq 0$ we get $\Theta_{[k]} \in [1, k]$. Together with $\frac{1-\theta_2}{\Theta_{[k]}} = \frac{h\Theta_{[k]} - g_2(\boldsymbol{\theta})}{\Theta_{[k]}} \geq h$ this shows $1 - \theta_j \geq 1 - \theta_2 \geq h$ and in particular $\ln(1/(1 - \theta_j)) \leq \ln(h^{-1})$ for each $j \in \{2, \ldots, m\}$. In combination with Claims 1 and 2 this allows us to conclude

$$
\begin{aligned}
&\frac{\Theta_{[k]}}{k} \sum_{j=2}^{m} \frac{1}{(1 - \theta_j)^2} \ln \left( \frac{k}{\gamma} \ln \left( \frac{1}{1 - \theta_j} \right) \right) \\
&\leq \frac{1}{k} \ln \left( \frac{k}{\gamma} \ln(h^{-1}) \right) \sum_{j=2}^{m} \frac{1 + \theta_2 + \cdots + \theta_k}{(1 - \theta_j)^2} \\
&\leq \frac{m-1}{k(k-1)} \ln \left( \frac{k}{\gamma} \ln(h^{-1}) \right) \sum_{j=2}^{k} \frac{1 + \theta_2 + \cdots + \theta_k}{(1 - \theta_j)^2} \\
&\leq \frac{3(m-1)(1 + hk)}{k(k-1)h^2} \ln \left( \frac{k}{\gamma} \ln(h^{-1}) \right) \\
&\leq \frac{6m}{kh^2} \left( \frac{1}{k} + h \right) \ln \left( \frac{k}{\gamma} \ln(h^{-1}) \right),
\end{aligned}
$$

where we have used that $\frac{m-1}{k-1} \leq \frac{2m}{k}$ holds due to $k \geq 2$. This completes the proof of Claim 3 and of the theorem.

$\square$

## C  Proofs for Section 4.1

**Proposition C.1** (Detailed version of Proposition 4.1). *Let $0 < \gamma < \gamma_0 < 1/2$ and $0 < h < h_0 < 1$ be fixed. Suppose $\mathcal{A}$ solves $\mathcal{P}_k^{k,\gamma}(\Delta^h)$, let $\mathbf{p} \in \Delta_k^h$ be arbitrary and write $i := \mathrm{mode}(\mathbf{p})$. Then,*

$$
\mathbb{E}_{\mathbf{p}} \left[ T^{\mathcal{A}} \right] \geq \frac{f \left( \frac{p_i - p_j}{2(p_i + p_j)}, \gamma \right)}{p_i + p_j}
$$

*holds for all $j \in [k] \setminus \{i\}$ with $f(z, \gamma) := \frac{1 - 2\gamma}{2z} \left\lceil \frac{\ln((1-\gamma)/\gamma)}{\ln((1/2+z)/(1/2-z))} \right\rceil$, which fulfills $\forall z \in (0, h_0/2)$ : $f(z, \gamma) \geq c(h_0, \gamma_0) z^{-2} \ln(\gamma^{-1})$ for some appropriate constant $c(h_0, \gamma_0) > 0$ that does not depend on $\gamma$ or $h$. In particular, we obtain the worst-case bound*

$$
\sup_{\mathbf{p} \in \Delta_k^h} \mathbb{E}_{\mathbf{p}}[T^{\mathcal{A}}] \geq 4c(h_0, \gamma_0) h^{-2} \ln(\gamma^{-1}) \tag{8}
$$

*and the instance-wise bound*

$$\forall \mathbf{p} \in \Delta_k^h : \mathbb{E}_{\mathbf{p}}[T^{\mathcal{A}}] \geq 2c(h_0, \gamma_0)(h(\mathbf{p}))^{-2} \ln(\gamma^{-1}) \left( \frac{1}{k} + h \right). \tag{9}$$

We prepare the proof of Proposition C.1 with sample complexity lower bounds of solutions to $\mathcal{P}_2^{2,\gamma}(\Delta^h)$. For the sake of convenience, we write $p$ for $(p, 1-p) \in \Delta_2$. Note that solving $\mathcal{P}_2^{2,\gamma}(\Delta^h)$ resp. $\mathcal{P}_2^{2,\gamma}(\Delta^0)$ reduces to deciding with error probability $\leq \gamma$

$$\mathbf{H}_0 : p > 1/2 \quad \text{vs.} \quad \mathbf{H}_1 : p < 1/2 \tag{10}$$

based on iid samples $X_1, X_2, \cdots \sim \mathrm{Ber}(p)$ for any $p \in [0, 1]$ with $|p - 1/2| \geq h$ resp. $|p - 1/2| > 0$.

**Lemma C.2.** *Let $0 < \gamma < \gamma_0 < 1/2$ and $0 < h < h_0 < 1/2$ and suppose $\mathcal{A}$ is able to decide* (10) *with confidence $\geq 1 - \gamma$ for any $p \in \{1/2 \pm h\}$, i.e.,*

$$\mathbb{P}_{1/2+h}(\mathbf{D}(\mathcal{A}) = 0) \geq 1 - \gamma \quad \text{and} \quad \mathbb{P}_{1/2-h}(\mathbf{D}(\mathcal{A}) = 1) \geq 1 - \gamma.$$

*There exists a constant $c(h_0, \gamma_0) > 0$, which does not depend on $\gamma$ or $h$ s.t.*

$$\mathbb{E}_{1/2 \pm h}[T^{\mathcal{A}}] = \frac{1 - 2\gamma}{2h} \left\lceil \frac{\ln((1 - \gamma)/\gamma)}{\ln((1/2+h)/(1/2-h))} \right\rceil \geq c(h_0, \gamma_0)h^{-2}\ln(\gamma^{-1}).$$

*Proof.* Let $\mathcal{A}'$ be the corresponding *Sequential Probability Ratio Test* (cf. [41]) for (10), i.e. it samples $X_1, X_2, \ldots$ until the first time $n$, where $\frac{1}{n}\sum_{k=1}^n X_k \notin [1/2 \pm C_{h,\gamma}(n)]$ with $C_{h,\gamma}(n) :=$ $\frac{1}{2n}\left\lceil \frac{\ln((1-\gamma)/\gamma)}{\ln((1/2+h)/(1/2-h))} \right\rceil$ and decides for 0 in case $\frac{1}{n}\sum_{k=1}^n X_k > 1/2 + C_{h,\gamma}(n)$ and for 1 in case $\frac{1}{n}\sum_{k=1}^n X_k < 1/2 - C_{h,\gamma}(n)$. On p.10–15 in [38] it is shown that $\mathcal{A}'$ fulfills

$$\mathbb{P}_{1/2+h}(\mathbf{D}(\mathcal{A}') = 0) \geq 1 - \gamma \quad \text{and} \quad \mathbb{P}_{1/2-h}(\mathbf{D}(\mathcal{A}') = 1) \geq 1 - \gamma,$$

as well as

$$\mathbb{E}_{1/2 \pm h}[T^{\mathcal{A}'}] = \frac{1 - 2\gamma}{2h} \left\lceil \frac{\ln((1 - \gamma)/\gamma)}{\ln((1/2+h)/(1/2-h))} \right\rceil =: g(h, \gamma).$$

According to pages 19–22 in [38] or [15, Theorem 2, p. 365] or the original proof from [41], $\mathcal{A}'$ is a test $\mathcal{A}''$ with error $\leq \gamma$ (on any instance $p \in \{1/2 \pm h\}$) for (10), for which $\mathbb{E}_{1/2 \pm h}[T^{\mathcal{A}''}]$ is minimal. In particular, we have

$$\mathbb{E}_{1/2 \pm h}[T^{\mathcal{A}}] \geq \mathbb{E}_{1/2 \pm h}[T^{\mathcal{A}'}] \geq g(h, \gamma).$$

Since $w : (0, 1) \to \mathbb{R}, \gamma \mapsto \frac{\ln((1-\gamma)/\gamma) \cdot (1-2\gamma)}{\ln(1/\gamma)}$ fulfills $w(1/2) = 0$ and

$$w'(\gamma) = \frac{(1 - 2\gamma)\ln(\gamma^{-1}) - (\gamma - 1)\ln(\gamma^{-1} - 1)(2\gamma + 2\gamma\ln(\gamma^{-1}) - 1))}{(\gamma - 1)\gamma\ln^2(\gamma^{-1})} < 0$$

for every $\gamma \in (0, 1/2)$, there exists some $c'(\gamma_0) > 0$ with $\ln((1 - \gamma)/\gamma)(1 - 2\gamma) \geq c'(\gamma_0)\ln(1/\gamma)$ for each $\gamma \in (0, \gamma_0)$. Moreover, as $\ln(1 + x) < x$ for $x > -1$, we obtain for $h \in (0, h_0)$ the inequality

$$\ln\left(\frac{1/2 + h}{1/2 - h}\right) = \ln\left(1 + \frac{4h}{1 - 2h}\right) < \frac{4h}{1 - 2h} < \frac{4h}{1 - 2h_0}.$$

Combining these estimates, we get $g(h, \gamma) \geq c(h_0, \gamma_0)h^{-2}\ln(\gamma^{-1})$ with $c(h_0, \gamma_0) := \frac{c'(\gamma_0)(1-2h_0)}{8}$. $\square$

Before proving Proposition C.1, we state two further auxiliary lemmata. The first one is a simplified version of *Walds identity* (cf. e.g. Thm. 17.7 in [2]), which we shortly prove for the sake of convenience. The second lemma is only required for the instance-wise bound in Proposition C.1.

**Lemma C.3.** *Let $k \in \mathbb{N}$ and $(p_1, \ldots, p_k) \in \Delta_k$ be fixed. Suppose $\{X_t\}_{t \in \mathbb{N}}$ to be an iid family of random variables $X_t \sim \mathrm{Cat}(p_1, \ldots, p_k)$ on some joint probability space $(\Omega, \mathcal{F}, \mathbb{P})$ and $\{\mathcal{F}_t\}_{t \in \mathbb{N}} \subseteq \mathcal{F}$ to be a filtration, such that $\{X_t\}_t$ is $\{\mathcal{F}_t\}_t$-adapted and $\forall t : X_t \perp\!\!\!\perp \mathcal{F}_{t-1}$, e.g. $\mathcal{F}_t = \sigma(X_1, \ldots, X_t)$. If $\tau$ is an $\{\mathcal{F}_t\}_t$-stopping time, then the random variables*

$$T_i(\tau) := \sum_{t \leq \tau} \mathbf{1}_{\{X_t = i\}}, \quad i \in [k],$$

*fulfill* $\mathbb{E}[T_i(\tau)] = p_i\mathbb{E}[\tau]$ *for each* $i \in [k]$. *In particular, we obtain*

$$\mathbb{E}[\tau] = \frac{\sum_{i\in I}\mathbb{E}[T_i(\tau)]}{\sum_{i\in I}p_i}$$

*for any* $I \subseteq [k]$ *with* $\sum_{i\in I}p_i > 0$.

*Proof.* Since $\{t \leq \tau\} = \{t > \tau\}^c = \{\tau \leq t-1\}^c \in \mathcal{F}_{t-1}$ holds for any $t \in \mathbb{N}$ and $X_t \perp\!\!\!\perp \mathcal{F}_{t-1}$, we obtain

$$\mathbb{E}\left[\mathbf{1}_{\{X_t=i\}}\mathbf{1}_{\{t\leq\tau\}}\right] = \mathbb{E}\left[\mathbb{E}\left[\mathbf{1}_{\{X_t=i\}}\mathbf{1}_{\{t\leq\tau\}}\big|\mathcal{F}_{t-1}\right]\right]$$
$$= \mathbb{E}[\mathbf{1}_{\{t\leq\tau\}}\mathbb{E}[\mathbf{1}_{\{X_t=i\}}|\mathcal{F}_{t-1}]] = p_i\mathbb{E}[\mathbf{1}_{\{t\leq\tau\}}].$$

Via an application of the monotone convergence theorem we infer

$$\mathbb{E}\left[T_i(\tau)\right] = \lim_{T\to\infty}\mathbb{E}[T_i(\tau \wedge T)]$$
$$= \lim_{T\to\infty}\sum_{t\leq T}\mathbb{E}\left[\mathbf{1}_{\{X_t=i\}}\mathbf{1}_{\{t\leq\tau\}}\right]$$
$$= p_i\lim_{T\to\infty}\sum_{t\leq T}\mathbb{E}\left[\mathbf{1}_{\{t\leq\tau\}}\right]$$
$$= p_i\lim_{T\to\infty}\mathbb{E}[\tau \wedge T] = p_i\mathbb{E}[\tau].$$

and thus in particular $\sum_{i\in I}\mathbb{E}[T_i(\tau)] = \mathbb{E}[\tau]\sum_{i\in I}p_i$. $\qquad\square$

**Lemma C.4.** *Suppose* $\mathbf{p} \in \Delta_k^h \setminus \Delta_k^{\tilde{h}}$ *for some* $0 < h < \tilde{h} < 1$ *and let* $i := \mathrm{mode}(\mathbf{p})$ *and* $j \in \arg\max_{l\in[k]\setminus\{i\}}p_l$. *Then, we have* $p_i + p_j \geq \frac{2+(k-2)h}{k}$ *and* $p_i - p_j < \tilde{h}$.

*Proof.* From $\mathbf{p} \in \Delta_k^h$ and $\mathrm{mode}(\mathbf{p}) = i$ we infer that $p_l \leq p_i - h$ holds for each $l \in [k] \setminus \{i\}$. Thus,

$$1 = \sum_{l\in[k]}p_l \leq p_i + \sum_{l\neq i}(p_i - h) = kp_i - (k-1)h$$

shows us that $p_i = \frac{1+(k-1)h}{k} + \varepsilon$ for some $\varepsilon \geq 0$. Due to $\sum_{l\neq i}p_l = 1 - p_i$ and $p_j = \max_{l\in[k]\setminus\{i\}}p_l$, we have

$$p_j \geq \frac{1-p_i}{k-1} = \frac{1 - \frac{1+(k-1)h}{k} - \varepsilon}{k-1} = \frac{1+h}{k} - \frac{\varepsilon}{k-1}.$$

Consequently,

$$p_i + p_j \geq \frac{1+(k-1)h}{k} + \varepsilon + \frac{1+h}{k} - \frac{\varepsilon}{k-1}$$
$$= \frac{2+(k-2)h}{k} + \frac{(k-2)\varepsilon}{k-1}$$
$$\geq \frac{2+(k-2)h}{k}.$$

Moreover, $\mathbf{p} \notin \Delta_k^{\tilde{h}}$ assures the existence of some $j' \in [m] \setminus \{i\}$ with $p_i < p_{j'} + \tilde{h}$. Since the choice of $j$ guarantees $p_{j'} + \tilde{h} \leq p_j + \tilde{h}$, this implies $p_i - p_j < \tilde{h}$. $\qquad\square$

*Proof of Proposition C.1.* We may suppose w.l.o.g. $i = 1$ and fix $j = 2$. Let us define $a := \frac{p_1}{p_1+p_2}$ and suppose we have a coin $C \sim \mathrm{Ber}(p)$ with $p \in \{a, 1-a\}$. By simulating $\mathcal{A}$, we will construct an algorithm $\mathcal{A}'$ for testing

$$\mathbf{H}'_0 : p = a \qquad \mathbf{H}'_1 : p = 1 - a$$

in the following way: Whenever $\mathcal{A}$ makes a query at time $t$, we generate an independent sample $U_t \sim \mathcal{U}([0,1])$. Then, we return the feedback $X_t = i' \in \{3, \ldots, k\}$ iff $U_t \in (\sum_{j'\leq i'-1}p_{j'}, \sum_{j'\leq i'}p_{j'}]$ and in case $U_t \in [0, p_1 + p_2]$ we generate an independent sample $C_t \sim \mathrm{Ber}(p)$ from our coin $C$ and return

$$X_t = \begin{cases} 1, & \text{if } C_t = 1, \\ 2, & \text{if } C_t = 0. \end{cases}$$

As soon as $\mathcal{A}$ terminates, we terminate and return $\mathbf{D}(\mathcal{A}') = 0$ if $\mathbf{D}(\mathcal{A}) = 1$ and $\mathbf{D}(\mathcal{A}') = 1$ otherwise. By our construction, we have $\mathbb{P}_p(X_t = i) = p_i$ for each $i \in \{3, \dots, k\}$

$$\mathbb{P}_a(X_t = 1) = (p_1 + p_2)\mathbb{P}(C_t = 1) = p_1, \quad \mathbb{P}_a(X_t = 2) = (p_1 + p_2)\mathbb{P}(C_t = 0) = p_2$$

and similarly $\mathbb{P}_{1-a}(X_t = 1) = p_2$ and $\mathbb{P}_{1-a}(X_t = 2) = p_1$. Thus if $p = a$, $\mathcal{A}$ behaves as started on $\mathbf{p}$ and if $p = 1 - a$, $\mathcal{A}$ behaves as started on $\mathbf{p}' := (p_2, p_1, p_3, \dots, p_k) \in \Delta_k^h$. Since $\mathcal{A}$ solves $\mathcal{P}_k^{k,\gamma}(\Delta^h)$, we obtain

$$\mathbb{P}_a(\mathbf{D}(\mathcal{A}') = 0) = \mathbb{P}_{\mathbf{p}}\left(\mathbf{D}(\mathcal{A}) = 1\right) \geq 1 - \gamma$$

and (due to $2 = \arg\max_{j' \in [k]} p'_j$)

$$\mathbb{P}_{1-a}(\mathbf{D}(\mathcal{A}') = 1) = \mathbb{P}_{\mathbf{p}'}(\mathbf{D}(\mathcal{A}) \neq 1) \geq \mathbb{P}_{\mathbf{p}'}(\mathbf{D}(\mathcal{A}) = 2) \geq 1 - \gamma,$$

i.e., $\mathcal{A}'$ is able to decide $\mathbf{H}'_0$ versus $\mathbf{H}'_1$ with error probability $\leq \gamma$. From Lemma C.2 we infer that it has to throw the coin $C$ (in both cases $p \in \{a, 1 - a\}$) in expectation at least $f(a - 1/2, \gamma)$ times for this. Regarding that $C$ is thrown in our construction iff we return as feedback an element from $\{1, 2\}$, we get that

$$\mathbb{E}_{\mathbf{p}}[T_1(T^{\mathcal{A}}) + T_2(T^{\mathcal{A}})] \geq f(a - 1/2, \gamma) \quad \text{where } T_i(T^{\mathcal{A}}) := \sum_{t \leq T^{\mathcal{A}}} \mathbf{1}_{\{X_t = i\}}.$$

An application of Lemma C.3 yields

$$\mathbb{E}_{\mathbf{p}}[T^{\mathcal{A}}] \geq \frac{f(a - 1/2, \gamma)}{p_1 + p_2} = \frac{f\left(\frac{p_1 - p_2}{2(p_1 + p_2)}, \gamma\right)}{p_1 + p_2},$$

which completes the proof of the first statement.

The worst-case bound (8) then follows from the just proven bound via

$$\sup_{\mathbf{p} \in \Delta_k^h} \mathbb{E}_{\mathbf{p}}[T^{\mathcal{A}}] \geq \mathbb{E}_{\left(\frac{1+h}{2}, \frac{1-h}{2}, 0, \dots, 0\right)}[T^{\mathcal{A}}] \geq f(h/2, \gamma) \geq 4c(h_0, \gamma_0)h^{-2}\ln(\gamma^{-1})$$

for some $c(h_0, \gamma_0) > 0$, that is assured to exist by Lemma C.2. To prove (9) suppose at first $\tilde{h} \in (h, 1)$ and $\mathbf{p} \in \Delta_k^h \setminus \Delta_k^{\tilde{h}}$ to be fixed and write $i := \text{mode}(\mathbf{p})$. Lemma C.4 reveals that there exists some $j \in [k] \setminus \{i\}$ with $p_i + p_j \geq \frac{2 + (k-2)h}{k}$ and $p_i - p_j < \tilde{h}$. Consequently, the above proven bound and the estimate $f(h, \gamma) \geq c(h_0, \gamma_0)h^{-2}\ln(\gamma^{-1})$ yield

$$\begin{aligned}
\mathbb{E}_{\mathbf{p}}[T^{\mathcal{A}}] &\geq \frac{f\left(\frac{p_i - p_j}{2(p_i + p_j)}, \gamma\right)}{p_i + p_j} \geq 4c(h_0, \gamma_0)\frac{p_i + p_j}{(p_i - p_j)^2}\ln(\gamma^{-1}) \\
&\geq 4c(h_0, \gamma_0)\tilde{h}^{-2}\ln(\gamma^{-1})\frac{2 + (k-2)h}{k} \\
&\geq 2c(h_0, \gamma_0)\tilde{h}^{-2}\ln(\gamma^{-1})\left(\frac{1}{k} + h\right).
\end{aligned}$$

Since $\mathbf{p} \in \Delta_k^{h(\mathbf{p})} \setminus \left(\bigcup_{\tilde{h} > h(\mathbf{p})} \Delta_k^{\tilde{h}}\right) = \bigcap_{\tilde{h} > h(\mathbf{p})}(\Delta_k^{h(\mathbf{p})} \setminus \Delta_k^{\tilde{h}})$ for any $\mathbf{p} \in \Delta_k^h$, (9) can be inferred from this by taking the limit $\tilde{h} \searrow h(\mathbf{p})$. $\qquad\square$

From Lemma C.2 we can infer that any solution $\mathcal{A}$ to $\mathcal{P}_2^{2,\gamma}(\Delta^0)$ fulfills $\lim_{h \to 0} \mathbb{E}_{1/2 \pm h}[T^{\mathcal{A}}] \in \Omega(h^{-2})$ as $h \to 0$. The following lemma improves upon this bound and is the key ingredient for the proof of Proposition 4.2.

**Lemma C.5.** *Let $\gamma \in (0, 1/2)$ be fixed and suppose $\mathcal{A}$ to be an algorithm, which terminates a.s. for any $p \neq 1/2$ and is able to decide (10) for any $p \neq 1/2$ with confidence $\geq 1 - \gamma$, i.e.,*

$$\forall p > 1/2 : \mathbb{P}_p(\mathbf{D}(\mathcal{A}) = 0) \geq 1 - \gamma \quad \text{and} \quad \forall p < 1/2 : \mathbb{P}_p(\mathbf{D}(\mathcal{A}) = 1) \geq 1 - \gamma.$$

*Then,*

$$\limsup_{h \to 0} \frac{\mathbb{E}_{1/2 \pm h}\left[T^{\mathcal{A}}\right]}{h^{-2}\ln\ln h^{-1}} \geq \frac{1}{2}\mathbb{P}_{1/2}(T^{\mathcal{A}} = \infty) \geq \frac{1}{2}(1 - 2\gamma) > 0.$$

*Proof.* This is stated in Theorem 1 in [14]. To verify this, note that $|\ln|\ln|h|||^{-1} = (\ln\ln h^{-1})^{-1}$ holds for $h < \frac{1}{e}$ and also confer the remark directly after Theorem 1 therein. $\square$

*Proof of Proposition 4.2.* We suppose w.l.o.g. $(i,j) = (1,2)$ throughout the proof. For $h \in (0, p_1 - p_2)$ we have $(\mathbf{p}(h))_1 > (\mathbf{p}(h))_2 > (\mathbf{p}(h))_l$ for every $l \in \{3, \ldots, k\}$ and together with $|(\mathbf{p}(h))_1 - (\mathbf{p}(h))_2| = h$ this shows $\mathbf{p}(h) \in \Delta_k^h$. Suppose we have a coin $C \sim \mathrm{Ber}(p)$ for $p \neq 1/2$. By simulating $\mathcal{A}$ as in the proof of Proposition 4.1 we obtain an algorithm $\mathcal{A}'$ for testing $\mathbf{H}_0 : p > 1/2$ versus $\mathbf{H}_1 : p < 1/2$, which has (due to the theoretical guarantees of $\mathcal{A}$) an error probability $\leq \gamma$ for every $p \neq 1/2$. Consequently, Lemma C.5 guarantees the existence of a sequence $\{h'_l\}_{l \in \mathbb{N}} \subseteq (0, e^{-4})$ with

$$\forall l \in \mathbb{N} : \frac{\mathbb{E}_{1/2 \pm h'_l}[T^{\mathcal{A}'}]}{h'^{-2}_l \ln\ln h'^{-1}_l} \geq \frac{1 - 2\gamma}{2} - \varepsilon > 0$$

for some arbitrarily small but fixed $\varepsilon \in (0, \frac{1-2\gamma}{2})$. If we choose $h_l := 2(p_1 + p_2)h'_l$, then the corresponding bias of the coin $C$ in the reduction (cf. the proof of Proposition 4.1) is exactly

$$\frac{(\mathbf{p}(h_l))_1}{(\mathbf{p}(h_l))_1 + (\mathbf{p}(h_l))_2} = \frac{\frac{p_1+p_2}{2} + \frac{h_l}{2}}{p_1 + p_2} = \frac{1}{2} + \frac{h_l}{2(p_1 + p_2)} = \frac{1}{2} + h'_l$$

Hence, if $\mathcal{A}'$ is started on $1/2 + h'_l$, its internal method $\mathcal{A}$ works as if started on $\mathbf{p}(h_l)$. From $h_l \leq e^{-4}$ we obtain $4 = (1/2)^{-2} \leq \ln(h_l^{-1})$ and thus $-2\ln(1/2) \leq \ln\ln(h_l^{-1})$, i.e., $\ln(1/2) \geq -1/2 \ln\ln(h_l^{-1}) \geq -1/2 \ln(h_l^{-1})$. Consequently,

$$\ln\ln h'^{-1}_l = \ln\ln\left(\frac{h_l^{-1}}{2(p_1 + p_2)}\right) \geq \ln\left(\ln(1/2) + \ln(h_l^{-1})\right)) \geq \ln\left(\frac{1}{2}\ln(h_l^{-1})\right)$$

$$= \ln(1/2) + \ln\ln(h_l^{-1}) \geq \frac{1}{2}\ln\ln(h_l^{-1})$$

holds, and we obtain similarly as in the proof of Proposition 4.1

$$\mathbb{E}_{\mathbf{p}(h_l)}[T_1(T^{\mathcal{A}}) + T_2(T^{\mathcal{A}})] \geq \mathbb{E}_{1/2+h'_l}[T^{\mathcal{A}'}] \geq \left(\frac{1}{2}(1 - 2\gamma) - \varepsilon\right) h'^{-2}_l \ln\ln h'^{-1}_l$$

$$\geq 2(p_1 + p_2)^2 \left(\frac{1}{2}(1 - 2\gamma) - \varepsilon\right) h_l^{-2} \ln\ln h_l^{-1}$$

Regarding that his holds for arbitrarily small $\varepsilon > 0$, Lemma C.3 shows[5] that

$$\frac{\mathbb{E}_{\mathbf{p}(h_l)}[T^{\mathcal{A}}]}{h_l^{-2} \ln\ln h_l^{-1}} \geq (1 - 2\gamma)(p_1 + p_2)$$

holds for every $l \in \mathbb{N}$, which completes the proof. $\square$

# D  Proofs of Section 4.2

Our upper bounds for both the cases $m = k$ and $m \geq k$ rely on the Kiefer-Dvoretzky-Wolfowitz inequality, which we state in the following for convenience only for categorical random variables.

**Lemma D.1.** *Suppose $X_1, X_2, \ldots$ to be iid random variables $X_n \sim \mathrm{Cat}(\mathbf{p})$ for some $\mathbf{p} \in \Delta_k$. For $t \in \mathbb{N}$ let $\hat{\mathbf{p}}^t$ be the corresponding empirical distribution after the $t$ observations $X_1, \ldots, X_t$, i.e., $\hat{p}_i^t = \frac{1}{t} \sum_{s=1}^{t} \mathbf{1}_{\{X_s = i\}}$ for all $i \in [k]$. Then, we have for any $\varepsilon > 0$ and $t \in \mathbb{N}$ the estimate*

$$\mathbb{P}\left(\left|\left|\hat{\mathbf{p}}^t - \mathbf{p}\right|\right|_\infty > \varepsilon\right) \leq 4e^{-t\varepsilon^2/2}.$$

*Proof.* Confer [12, 24] as well as Theorem 11.6 in [22]. Moreover, note that the cumulative distribution functions $F$ resp. $\hat{F}^t$ of $X_1 \sim \mathrm{Cat}(\mathbf{p})$ resp. $\hat{\mathbf{p}}^t$ fulfill $p_j = F(j) - F(j-1)$ and $\hat{p}_j^t = \hat{F}^t(j) - \hat{F}^t(j-1)$ and thus

$$|\hat{p}_j^t - p_j| \leq |\hat{F}^t(j) - F(j)| + |\hat{F}^t(j-1) - F(j-1)|.$$

for each $j \in [k]$. $\square$

---

[5] Note here that $(\mathbf{p}(h))_1 + (\mathbf{p}(h))_2 = p_1 + p_2$.

**Lemma D.2.** *For $h \in [0,1], \varepsilon \in (-h, 1], \mathbf{p} \in \Delta_k^h$ and $\tilde{\mathbf{p}} \in \Delta_k$ we have*

$$(\exists i : \tilde{p}_i - \max_{j \neq i} \tilde{p}_j \geq \varepsilon \text{ and } p_i \neq \max_j p_j) \quad \Rightarrow \quad ||\tilde{\mathbf{p}} - \mathbf{p}||_\infty \geq (h + \varepsilon)/2.$$

*Proof.* Suppose there is some $i \in [k]$ s.t. $\tilde{p}_i - \max_{j \neq i} \tilde{p}_j \geq \varepsilon$ and $p_i \neq \max_j p_j$ hold. Then, there exists some $j \in [k] \setminus \{i\}$ with

$$p_j \geq p_i + h \quad \text{and} \quad \tilde{p}_i \geq \tilde{p}_j + \varepsilon$$

and we conclude

$$2\,||\tilde{\mathbf{p}} - \mathbf{p}||_\infty \geq |p_j - \tilde{p}_j| + |\tilde{p}_i - p_i| \geq (p_j - p_i) + (\tilde{p}_i - \tilde{p}_j) \geq h + \varepsilon.$$

$\square$

**Remark D.3.** *The bounds from Lemma D.2 are sharp: Consider e.g. $\mathbf{p} \in \Delta_k^h$ and $\tilde{\mathbf{p}} \in \Delta_k$ defined via*

$$p_i = \begin{cases} 1/2 - h/2, & \text{if } i = 1, \\ 1/2 + h/2, & \text{if } i = 2, \\ 0, & \text{otherwise} \end{cases} \quad \text{and} \quad \tilde{p}_i = \begin{cases} 1/2 + \varepsilon/2, & \text{if } i = 1, \\ 1/2 - \varepsilon/2, & \text{if } i = 2, \\ 0, & \text{otherwise.} \end{cases}$$

*Then, we have $\tilde{p}_1 - \max_{j \neq 1} \tilde{p}_j = \varepsilon$ and $p_1 \neq \max_{j \in [k]} p_j$ and at the same time $||\mathbf{p} - \tilde{\mathbf{p}}||_\infty = \frac{h + \varepsilon}{2}$.*

For sake of convenience, we give a pseudo-code for the straightforward strategy described in Section 4.2 for solving $\mathcal{P}_k^{k,\gamma}(\Delta^h)$ .

---

**Algorithm 4** DKW mode identification – (non-sequential) solution to $\mathcal{P}_k^{k,\gamma}(\Delta^h)$

---

**Input:** $\gamma \in (0,1)$, $h \in (0,1)$, $k \in \mathbb{N}$, access to iid samples $X_t \sim \text{Cat}(\mathbf{p})$
1: Let $T \leftarrow \lceil 8 \ln(4/\gamma) h^{-2} \rceil$
2: Observe $X_1, \ldots, X_T \sim \text{Cat}(\mathbf{p})$
3: **return** $\text{mode}(\hat{\mathbf{p}}^T) = \arg\max_{i \in [k]} \sum_{t=1}^T \mathbf{1}_{\{X_t = i\}}$

---

As a direct consequence of Lemma D.1 and Lemma D.2 we obtain the following result.

**Proposition D.4.** *For any $k \in \mathbb{N}$, $h \in (0,1)$ and $\gamma \in (0,1)$, Algorithm 4 called with parameters $\gamma, h, k$ solves $\mathcal{P}_k^{k,\gamma}(\Delta^h)$ and terminates after exactly $\lceil 8 \ln(4/\gamma) h^{-2} \rceil$ time steps.*

**Lemma D.5.** *Let $h > 0$, $\mathbf{p} \in \Delta_k^{3h}$ and $\tilde{\mathbf{p}} \in \Delta_k$ be fixed. Then,*

$$\forall i : \tilde{p}_i \leq \max_{j \neq i} \tilde{p}_j + h \quad \Rightarrow \quad ||\mathbf{p} - \tilde{\mathbf{p}}||_\infty \geq h.$$

*Proof.* To prove the contraposition, we suppose $||\mathbf{p} - \tilde{\mathbf{p}}||_\infty < h$ to be fulfilled. Let $i := \text{mode}(\mathbf{p}) \in [k]$ and fix some arbitrary $j \in [k] \setminus \{i\}$. Since $\mathbf{p} \in \Delta_k^{3h}$ assures $p_i \geq p_j + 3h$, we obtain

$$\tilde{p}_i - \tilde{p}_j = p_i + (\tilde{p}_i - p_i) + (p_j - \tilde{p}_j) - p_j \geq p_i - p_j - 2\,||\mathbf{p} - \tilde{\mathbf{p}}||_\infty$$
$$> p_i - p_j - 2h \geq h.$$

As $j$ was arbitrary, we conclude that $\tilde{p}_i > \max_{j \neq i} \tilde{p}_j + h$, which completes the proof. $\square$

**Lemma D.6.** *For any $h \in (0, 1/8)$, $\varepsilon \in (0, 1/3)$ and $k \in \mathbb{N}_{\geq 3}$ there exist $\mathbf{p} \in \Delta_k^{(3-\varepsilon)h}$ and $\tilde{\mathbf{p}} \in \Delta_k$ such that*

$$\forall i \in [k] : \tilde{p}_i \leq \max_{j \neq i} \tilde{p}_j + h \quad \text{and} \quad ||\mathbf{p} - \tilde{\mathbf{p}}||_\infty < h.$$

*Proof.* Suppose $h \in (0, 1/8)$, $\varepsilon \in (0, 1/3)$ and $k \in \mathbb{N}_{\geq 3}$ to be fixed. Now, define $\mathbf{p} \in \Delta_k$ and $\tilde{\mathbf{p}} \in \Delta_k$ via

$$p_j := \begin{cases} \frac{1}{2} + h, & \text{if } j = 1, \\ \frac{1}{2} - (2 - \varepsilon)h, & \text{if } j = 2, \\ \frac{(1-\varepsilon)h}{k-2}, & \text{if } j \geq 3, \end{cases}$$

and

$$\tilde{p}_j := \begin{cases} p_1 - (1 - \frac{\varepsilon}{4})h = \frac{1}{2} + \frac{\varepsilon h}{4}, & \text{if } j = 1, \\ p_2 + (1 - \frac{\varepsilon}{4})h = \frac{1}{2} + (\frac{3\varepsilon}{4} - 1)h, & \text{if } j = 2, \\ \frac{(1-\varepsilon)h}{k-2}, & \text{if } j \geq 3. \end{cases}$$

From $h < 1/8$ we infer $1/2 - (2 - \varepsilon)h > 1/2 - 2h > 1/4$ and thus

$$\forall j \geq 3 : \frac{(k-2)p_j}{p_2} = \frac{(1-\varepsilon)h}{1/2 - (2-\varepsilon)h} < 4(1-\varepsilon)h < 4h < 1/2 < k - 2.$$

This shows $p_1 - (3 - \varepsilon)h = p_2 > \max_{j \geq 3} p_j$ and consequently $\mathbf{p} \in \Delta_k^{(3-\varepsilon)h}$. Since $\tilde{p}_j = p_j$ is fulfilled for each $j \geq 3$, we have $\tilde{p}_1 > \tilde{p}_2 > p_2 > \max_{j \geq 3} \tilde{p}_j$, and together with

$$\tilde{p}_1 - \tilde{p}_2 = \frac{\varepsilon h}{4} - \frac{3\varepsilon h}{4} + h = \left(1 - \frac{\varepsilon}{2}\right) h < h$$

we see that $\tilde{p}_i \leq \max_{j \neq i} \tilde{p}_j + h$ holds for each $i \in [m]$. Finally $||\mathbf{p} - \tilde{\mathbf{p}}||_\infty < h$ follows from $|p_1 - \tilde{p}_1| = (1 - \frac{\varepsilon}{4})h = |p_2 - \tilde{p}_2|$ as well as $p_j = \tilde{p}_j$ for all $j \geq 3$. $\qquad\square$

*Proof of Lemma 4.3.* Let $\mathbf{p} \in \Delta_k$ be fixed, and note that Algorithm 1 terminates after exactly $\lceil 8\ln(4/\gamma)h^{-2} \rceil$ time steps. Lemma D.2 and Lemma D.1 let us directly infer

$$\mathbb{P}_{\mathbf{p}}\left(\mathbf{D}(\mathcal{A}) \in [k] \text{ and } p_{\mathbf{D}(\mathcal{A})} < \max_{j \in [k]} p_j\right)$$
$$= \mathbb{P}\left(\exists i \in [k] : \hat{p}_i^t - \max_{j \neq i} \hat{p}_j^t > h \text{ and } p_i \neq \max_{j \in [k]} p_j\right)$$
$$\leq \mathbb{P}\left(\left|\left|\hat{\mathbf{p}}^t - \mathbf{p}\right|\right|_\infty > h/2\right) \leq \gamma. \tag{11}$$

Next, suppose $\mathbf{p} \in \Delta_k^0$ and let $i' := \mathrm{mode}(\mathbf{p}) \in [k]$. Again, Lemma D.2 yields

$$\{\mathbf{D}(\mathcal{A}) \in [k] \setminus \{i'\}\} = \left\{\exists i \neq i' : \hat{p}_i^t - \max_{j \neq i} \hat{p}_j^t > h \text{ and } p_{i'} > \max_{j \neq i'} p_j\right\}$$
$$\subseteq \left\{\left|\left|\hat{\mathbf{p}}^t - \mathbf{p}\right|\right|_\infty > h/2\right\}, \tag{12}$$

and thus

$$\mathbb{P}_{\mathbf{p}}(\mathbf{D}(\mathcal{A}) \notin \{i', \text{UNSURE}\}) \leq \mathbb{P}_{\mathbf{p}}\left(\left|\left|\hat{\mathbf{p}}^t - \mathbf{p}\right|\right|_\infty > h/2\right) \leq \gamma$$

follows from Lemma D.1 and the choice of $t$. Now, let us suppose $\mathbf{p} \in \Delta_k^{3h}$. A look at Lemma D.5 reveals

$$\{\mathbf{D}(\mathcal{A}) = \text{UNSURE}\} = \left\{\forall i \in [k] : \hat{p}_i^t \leq \max_{j \neq i} \hat{p}_j^t + h\right\} \subseteq \left\{\left|\left|\hat{\mathbf{p}}^t - \mathbf{p}\right|\right|_\infty > h\right\},$$

and combining this with (12) yields

$$\mathbb{P}_{\mathbf{p}}(\mathbf{D}(\mathcal{A}) \neq \mathrm{mode}(\mathbf{p})) = \mathbb{P}_{\mathbf{p}}\left(\mathbf{D}(\mathcal{A}) \in [k] \setminus \{i'\} \text{ or } \mathbf{D}(\mathcal{A}) = \text{UNSURE}\right)$$
$$\leq \mathbb{P}_{\mathbf{p}}\left(\left|\left|\hat{\mathbf{p}}^t - \mathbf{p}\right|\right|_\infty > h/2\right) \leq \gamma,$$

where the last estimate is again due to Lemma D.1. $\qquad\square$

We proceed with the proof of Proposition 4.4.

*Proof of Proposition 4.4.* Let $\mathbf{p} \in \Delta_k^0$ be fixed and abbreviate $h := h(\mathbf{p})$. Moreover, denote by $\mathbf{D}(\mathcal{A}_s)$ the output of the instance of Algorithm 1 with parameters $\gamma_s, h_s$ that is called in iteration $s$ of the while loop of $\mathcal{A}$ (Algorithm 2). Let us define for each $s \in \mathbb{N}$ the set

$$\mathcal{E}_1^s := \{h_s > h/3 \text{ and } \mathbf{D}(\mathcal{A}_s) \in \{\text{UNSURE}, \mathrm{mode}(\mathbf{p})\}\},$$
$$\mathcal{E}_2^s := \{h_s \leq h/3 \text{ and } \mathbf{D}(\mathcal{A}_s) = \mathrm{mode}(\mathbf{p})\}$$

and

$$\mathcal{E} := \bigcup_{s \in \mathbb{N}} (\mathcal{E}_1^s \cup \mathcal{E}_2^s)^c.$$

From the equivalence $h' \leq \frac{1}{3}h(\mathbf{p}) \Leftrightarrow \mathbf{p} \in \Delta_k^{3h'}$ and Lemma 4.3 we infer

$$\mathbb{P}_{\mathbf{p}}\left(\left(\mathcal{E}_1^s \cup \mathcal{E}_2^s\right)^c\right) = \begin{cases} \mathbb{P}_{\mathbf{p}}\left(\left(\mathcal{E}_1^s\right)^c\right), & \text{if } h_s > h/3 \\ \mathbb{P}_{\mathbf{p}}\left(\left(\mathcal{E}_2^s\right)^c\right), & \text{if } h_s \leq h/3 \end{cases} \leq \gamma_s$$

and therefore

$$\mathbb{P}_{\mathbf{p}}(\mathcal{E}) \leq \sum_{s \in \mathbb{N}} \gamma_s = \sum_{s \in \mathbb{N}} \frac{6\gamma}{\pi^2 s^2} = \gamma. \tag{13}$$

Now, let $s_0 := s_0(h) \in \mathbb{N}$ be such that $h_{s_0} \leq h/3 < h_{s_0-1}$ and note that

$$\mathcal{E}^c \subseteq \mathcal{E}_2^{s_0} \subseteq \{\mathbf{D}(\mathcal{A}_{s_0}) \neq \text{UNSURE}\}$$
$$\subseteq \{\mathcal{A} \text{ terminates at latest after the } s_0\text{-th iteration of the while loop}\}. \tag{14}$$

In particular, $\mathcal{A}$ terminates almost surely on $\mathcal{E}^c$. Regarding the construction[6] of $\mathcal{A}$ we also have

$$\mathcal{E}^c = \bigcap_{s \in \mathbb{N}}\left(\mathcal{E}_1^s \cup \mathcal{E}_2^s\right) \subseteq \bigcap_{s \in \mathbb{N}} \{\mathbf{D}(\mathcal{A}_s) \in \{\text{UNSURE}, \text{mode}(\mathbf{p})\}\}$$
$$\subseteq \{\mathbf{D}(\mathcal{A}) = \text{mode}(\mathbf{p})\}. \tag{15}$$

Since $\mathcal{A}$ makes in its $s$-th iteration of the while loop (according to Algorithm 1) exactly $\lceil 8\ln(4/\gamma_s)h_s^{-2}\rceil$ queries, combining (13), (14) and (15) yields

$$\mathbb{P}_{\mathbf{p}}\left(\mathbf{D}(\mathcal{A}) = \text{mode}(\mathbf{p}) \text{ and } T^{\mathcal{A}} \leq t_0(h, \gamma)\right) \geq \mathbb{P}_{\mathbf{p}}\left(\mathcal{E}^c\right) \geq 1 - \gamma,$$

with $t_0(h, \gamma) := \sum_{s \leq s_0(h)} \lceil 8\ln(4/\gamma_s)h_s^{-2}\rceil$. As the choice of $s_0 = s_0(h)$ guarantees $\frac{h}{3} < h_{s_0-1} = 2^{-s_0}$ and thus $s_0 < \log_2(3h^{-1})$, we obtain with regard to the choices of $h_s = 2^{-s-1}$ and $\gamma_s = \frac{6\gamma}{\pi^2 s^2}$ that

$$t_0(h, \gamma) \leq 2^7 \sum_{s=1}^{s_0(h)} 2^{2s-1} \ln\left(\frac{2\pi^2 s^2}{3\gamma}\right) \in \mathcal{O}\left(\sum_{s=1}^{s_0(h)} 2^{2s-1} \ln\left(\frac{s_0(h)}{\gamma}\right)\right)$$
$$\subseteq \mathcal{O}\left(4^{s_0(h)} \ln\left(\frac{s_0(h)}{\gamma}\right)\right)$$
$$\subseteq \mathcal{O}\left(4^{\log_2(3/h)} \ln\left(\log_2(3h^{-1})\gamma^{-1}\right)\right)$$
$$\subseteq \mathcal{O}\left(h^{-2}\left(\ln\ln h^{-1} + \ln\gamma^{-1}\right)\right)$$

as $\min\{h, \gamma\} \to 0$. It remains to show that $T^{\mathcal{A}}$ is almost surely finite w.r.t. $\mathbb{P}_{\mathbf{p}}$. For an arbitrary integer $s \geq \log_2(3/h)$ we have $h_s \leq h/3$ and thus

$$\mathbb{P}_{\mathbf{p}}\left(T^{\mathcal{A}} = \infty\right) \leq \mathbb{P}_{\mathbf{p}}\left(\forall s' \in \mathbb{N} \text{ with } h_{s'} \leq h/3 : \mathbf{D}(\mathcal{A}_{s'}) = \text{UNSURE}\right)$$
$$\leq \mathbb{P}_{\mathbf{p}}\left(\mathbf{D}(\mathcal{A}_s) = \text{UNSURE}\right) \leq \mathbb{P}_{\mathbf{p}}((\mathcal{E}_2^s)^c) \leq \gamma_s,$$

which directly implies $\mathbb{P}_{\mathbf{p}}(T^{\mathcal{A}} = \infty) \leq \lim_{s \to \infty} \gamma_s = 0.$ □

## E  Remaining Proofs for Section 6

We prove the following more detailed version of Theorem 6.2.

**Theorem E.1.** *Let $\mathcal{A}$ be Algorithm 3 called with the parameters $k, m \in \mathbb{N}$ with $k \leq m$ and $\gamma \in (0, 1)$. Then, $\mathcal{A}$ solves $\mathcal{P}_k^{m,\gamma}(\exists \text{GCW} \wedge \Delta^0)$ and fulfills for any $\mathbf{P} = \{\mathbf{P}(\cdot|S)\}_{S \in [m]_k} \in PM_k^m(\exists \text{GCW} \wedge \Delta^0)$*

$$\mathbb{P}_{\mathbf{P}}\left(\mathbf{D}(\mathcal{A}) = \text{GCW}(\mathbf{P}) \text{ and } T^{\mathcal{A}} \leq t'(\mathbf{P}, m, k, \gamma)\right) \geq 1 - \gamma,$$

*where $t'(\mathbf{P}, m, k, \gamma)$ is given as*

$$\max\left\{\sum_{s \leq s'} t_0(h(\mathbf{P}(\cdot|B_s)), \gamma') : B_1, B_2, \ldots, B_{s'} \in [m]_k \text{ s.t. } \bigcup_{s \leq s'} B_s = [m]\right\} \tag{16}$$

*with $s' := \lceil\frac{m}{k-1}\rceil$, $\gamma' := \frac{\gamma}{s'}$ and $t_0(h, \gamma)$ defined as in Proposition 4.4, i.e., $t_0(h, \gamma) = \sum_{s \leq s_0(h)} \lceil 8\ln(4/\gamma_s)h_s^{-2}\rceil$ with $s_0(h) = \lceil\log_2(3/h)\rceil - 1$.*

[6]Note here that $\mathbf{D}(\mathcal{A}) \in [m]$ holds, i.e., $\mathcal{A}$ cannot terminate with UNSURE as output.

*Proof.* Suppose $\mathbf{P} = \{\mathbf{P}(\cdot|S)\}_{S \in [m]_k} \in PM_k^m(\exists \mathrm{GCW} \wedge \Delta^0)$ to be fixed and abbreviate $i := \mathrm{GCW}(\mathbf{P})$. Recall the internal values $s$, $S_s$ and $F_s$ of Algorithm 3. If $\mathcal{A}$ terminates, then the value of $s$ is $s' := \lceil \frac{m}{k-1} \rceil$. Let us write $\widetilde{\mathcal{A}}_s$ for the instance of Algorithm 2, which is called with parameters $m, \gamma'$ and sample access to $\mathbf{P}(\cdot|S_s)$ in Step 2 (or 9), i.e., we have $i_s = \mathbf{D}(\widetilde{\mathcal{A}}_s) \in S_s$ for each $s \leq s'$. For $s \geq 2$, $S_s$ and $F_s$ depend on the outcome of $\widetilde{\mathcal{A}}_{s-1}$ and are thus random variables.

**Claim 1:** On the event $\{T^{\mathcal{A}} < \infty\}$ we have

(i) $F_{s'} = \emptyset$ and $\bigcup_{s \leq s'} S_s = [m]$, i.e., $\sum_{s \leq s'} t_0(\gamma', h(\mathbf{P}(\cdot|S_s))) \leq t'(\mathbf{P}, m, k, \gamma)$ holds a.s.,

(ii) $\{\mathbf{D}(\mathcal{A}) \neq i\} \subseteq \bigcup_{s \leq s'} \{\mathbf{D}(\widetilde{\mathcal{A}}_s) \neq \mathrm{mode}(\mathbf{P}(\cdot|S_s))\}$.

**Proof of Claim 1:** Suppose $T^{\mathcal{A}} < \infty$. Clearly, $|F_s|$ is monotonically decreasing in $s$. Whenever $|F_s| \geq k$, then $|S_s \cap F_s| \geq k - 1$ and thus $|F_{s+1}| \leq |F_s| - (k-1)$ are fulfilled. Hence, $|F_s| \leq m - s(k-1)$ holds for any $s \leq s' - 1$. In particular, we have $|F_{s'-1}| \leq k - 1$, which implies $F_{s'} = \emptyset$.
From $[m] = F_0 \supseteq F_1 \supseteq \cdots \supseteq F_{s'} = \emptyset$ and $\forall s \leq s' : F_{s+1} = F_s \setminus S_s$ we infer $\bigcup_{s \leq s'} S_s = [m]$, which proves (i). Regarding that the implications

$$j \in S_s \setminus S_{s'} \;\Rightarrow\; \exists l \in \{0, \ldots, s' - s\} : j \in S_{s+l-1} \setminus S_{s+l}$$

and

$$j \in S_s \setminus S_{s+1} \;\Rightarrow\; j \neq i_s$$

are trivially fulfilled for all $j \in [m]$ and $s \in \{0, \ldots, s' - 1\}$, we obtain

$$\{i \notin S_{s'}\} \subseteq \{\exists s < s' : i \in S_s \text{ and } i \notin S_{s+1}\}$$
$$\subseteq \{\exists s < s' : i \in S_s \text{ and } i_s \neq i\}.$$

Due to $\{i \in S_s \text{ and } i_s \neq i\} \subseteq \{\mathbf{D}(\widetilde{\mathcal{A}}_s) \neq \mathrm{mode}(\mathbf{P}(\cdot|S_s))\}$, this implies

$$\{\mathbf{D}(\mathcal{A}) \neq i\} = \{i \in S_{s'} \text{ and } i \neq i_{s'}\} \cup \{i \notin S_{s'}\}$$
$$\subseteq \bigcup_{s \leq s'} \{i \in S_s \text{ and } i \neq i_s\}$$
$$\subseteq \bigcup_{s \leq s'} \{\mathbf{D}(\widetilde{\mathcal{A}}_s) \neq \mathrm{mode}(\mathbf{P}(\cdot|S_s))\}.$$

∎

**Claim 2:** We have the estimate

$$\mathbb{P}_{\mathbf{P}}\left( \exists s \leq s' : \mathbf{D}(\widetilde{\mathcal{A}}_s) \neq \mathrm{mode}(\mathbf{P}(\cdot|S_s)) \text{ or } T^{\widetilde{\mathcal{A}}_s} > t_0(\gamma', h(\mathbf{P}(\cdot|S_s))) \right) \leq \gamma.$$

**Proof of Claim 2:** For $s \leq s'$ let

$$E_s := \left\{ \mathbf{D}(\widetilde{\mathcal{A}}_s) \neq \mathrm{mode}(\mathbf{P}(\cdot|S_s)) \text{ or } T^{\widetilde{\mathcal{A}}_s} > t_0(\gamma', h(\mathbf{P}(\cdot|S_s))) \right\}$$

denote the set, where $\mathcal{A}$ fails at round $s$ in the sense that $\widetilde{\mathcal{A}}_s$ either makes an error in finding $\mathrm{mode}(\mathbf{P}(\cdot|S_s))$ or queries "too many" samples for this. For $B \in [m]_k$ and $s \leq s' - 1$ with $\mathbb{P}_{\mathbf{P}}(\{S_s = B\} \cap \bigcap_{\tilde{s} \leq s-1} E_{\tilde{s}}^c) > 0$ we have with regard to Proposition 4.4

$$\mathbb{P}_{\mathbf{P}}\left( E_s \,\Big|\, \{S_s = B\} \cap \bigcap_{\tilde{s} \leq s-1} E_{\tilde{s}}^c \right)$$
$$= \mathbb{P}_{\mathbf{P}(\cdot|B)}\left( \mathbf{D}(\widetilde{\mathcal{A}}_s) \neq \mathrm{mode}(\mathbf{P}(\cdot|B)) \text{ or } T^{\widetilde{\mathcal{A}}_s} > t_0(\gamma', h(\mathbf{P}(\cdot|B))) \right) \leq \gamma',$$

where we have used that both $\bigcap_{\tilde{s}\le s-1} E_{\tilde{s}}^c$ and the choice $\{S_s = B\}$ are independent of the samples observed by $\widetilde{\mathcal{A}}_s$. We conclude

$$\mathbb{P}_{\mathbf{P}}\left(\bigcup_{s\le s'} E_s\right) = \mathbb{P}_{\mathbf{P}}\left(\bigcup_{s\le s'} E_s \setminus \left(\bigcup_{\tilde{s}\le s-1} E_{\tilde{s}}\right)\right)$$

$$\le \sum_{s\le s'}\sum_{B\in[m]_k} \mathbb{P}_{\mathbf{P}}\left(E_s \cap \{S_s = B\} \cap \bigcap_{\tilde{s}\le s-1} E_{\tilde{s}}^c\right)$$

$$= \sum_{s\le s'}\left[\sum_B \mathbb{P}_{\mathbf{P}}\left(E_s \Big| \{S_s = B\} \cap \bigcap_{\tilde{s}\le s-1} E_{\tilde{s}}^c\right) \mathbb{P}_{\mathbf{P}}\left(\{S_s = B\} \cap \bigcap_{\tilde{s}\le s-1} E_{\tilde{s}}^c\right)\right]$$

$$\le \sum_{s\le s'} \gamma' \le \gamma,$$

where we have written $\sum_B$ for the sum over all $B \in [m]_k$ with $\mathbb{P}_{\mathbf{P}}\left(\{S_s = B\} \cap \bigcap_{\tilde{s}\le s-1} E_{\tilde{s}}^c\right) > 0$. $\blacksquare$

Now, let us define for $s \le s'$ the events

$$\mathcal{R}_s := \left\{T^{\widetilde{\mathcal{A}}_s} \le t_0(\gamma', h(\mathbf{P}(\cdot|S_s)))\right\}$$

and $\mathcal{R} := \bigcap_{s\le s'} \mathcal{R}_s$. Due to $T^{\mathcal{A}} = \sum_{s\le s'} T^{\widetilde{\mathcal{A}}_s}$ we have

$$\mathcal{R} \subseteq \left\{T^{\mathcal{A}} \le \sum_{s\le s'} t_0(\gamma', h(\mathbf{P}(\cdot|S_s)))\right\} \subseteq \left\{T^{\mathcal{A}} < \infty\right\}.$$

The equality $\mathcal{R}^c = \bigcup_{s\le s'} \mathcal{R}_s^c$ together with Part (ii) of Claim 1 and Claim 2 let us infer

$$\mathbb{P}_{\mathbf{P}}\left(\{\mathbf{D}(\mathcal{A}) \ne i\} \cup \mathcal{R}^c\right) = \mathbb{P}_{\mathbf{P}}\left((\{\mathbf{D}(\mathcal{A}) \ne i\} \cap \mathcal{R}) \cup \mathcal{R}^c\right)$$

$$\le \mathbb{P}_{\mathbf{P}}\left(\bigcup_{s\le s'}\left\{\mathbf{D}(\widetilde{\mathcal{A}}_s) \ne \mathrm{mode}(\mathbf{P}(\cdot|S_s))\right\} \cup \mathcal{R}_s^c\right)$$

$$= \mathbb{P}_{\mathbf{P}}\left(\exists s \le s' : \mathbf{D}(\widetilde{\mathcal{A}}_s) \ne \mathrm{mode}(\mathbf{P}(\cdot|S_s)) \text{ or } T^{\widetilde{\mathcal{A}}_s} > t_0(\gamma', h(\mathbf{P}(\cdot|S_s)))\right)$$

$$\le \gamma$$

and we can thus conclude with the help of Part (i) of Claim 1 that

$$\mathbb{P}_{\mathbf{P}}\left(\mathbf{D}(\mathcal{A}) = i \text{ and } T^{\mathcal{A}} \le t'(\mathbf{P}, m, k, \gamma)\right)$$

$$\ge \mathbb{P}_{\mathbf{P}}\left(\mathbf{D}(\mathcal{A}) = i \text{ and } T^{\mathcal{A}} \le \sum_{s\le s'} t_0(\gamma', h(\mathbf{P}(\cdot|S_s)))\right)$$

$$\ge \mathbb{P}_{\mathbf{P}}\left(\{\mathbf{D}(\mathcal{A}) = i\} \cap \mathcal{R}\right)$$

$$\ge 1 - \gamma.$$

$\square$

*Proof of Theorem 6.2.* According to Theorem E.1, $\mathcal{A}$ solves $\mathcal{P}_k^{m,\gamma}(\exists\mathrm{GCW} \wedge \Delta^0)$. Let $\mathbf{P} = \{\mathbf{P}(\cdot|S)\}_{S\in[m]_k} \in PM_k^m(\exists\mathrm{GCW} \wedge \Delta^0)$ be arbitrary. Theorem E.1 ensures that

$$\mathbb{P}_{\mathbf{P}}\left(\mathbf{D}(\mathcal{A}) \in \mathrm{GCW}(\mathbf{P}) \text{ and } T^{\mathcal{A}} \le t'(\mathbf{P}, m, k, \gamma)\right) \ge 1 - \gamma$$

holds with $t'(\mathbf{P}, m, k, \gamma)$ as in (16). By definition of $h(\mathbf{P})$ we have $h(\mathbf{P}(\cdot|S)) \ge h(\mathbf{P})$ for any $S \in [m]_k$, whence monotonicity of $t_0(h, \gamma)$ from Proposition 4.4 w.r.t. $h$ shows us that $t_0(h(\mathbf{P}(\cdot|S)), \gamma) \ge t_0(h(\mathbf{P}), \gamma)$ for any $S \in [m]_k$. Thus, a look at (16) reveals that

$$t'(\mathbf{P}, m, k, \gamma) \le T'(h(\mathbf{P}), m, k, \gamma)$$

with $T'(h, m, k, \gamma) := \left\lceil\frac{m}{k-1}\right\rceil t_0\left(h, \frac{\gamma}{\lceil m/(k-1)\rceil}\right)$, which is according to Proposition 4.4 in $\mathcal{O}\left(\frac{m}{kh^2}\ln\left(\frac{m}{k}\right)\left(\ln\ln h^{-1} + \ln\gamma^{-1}\right)\right)$. $\square$

The following algorithm is a solution to $\mathcal{P}_k^{m,\gamma}(\exists h\mathrm{GCW} \wedge \Delta^0)$.

**Theorem E.2.** *Let $\mathcal{A}$ be Algorithm 5 called with parameters $m, k \in \mathbb{N}$ with $k \le m$ and $\gamma, h \in (0, 1)$. Then, $\mathcal{A}$ solves $\mathcal{P}_k^{m,\gamma}(\exists h\mathrm{GCW} \wedge \Delta^0)$ and terminates a.s. for any $\mathbf{P} \in PM_k^m(\exists h\mathrm{GCW} \wedge \Delta^0)$ before some time $t'(m, k, h, \gamma) \in \mathcal{O}\left(\frac{m}{kh^2}\ln\left(\frac{m}{k\gamma}\right)\right)$.*

**Algorithm 5** Solution to $\mathcal{P}_k^{m,\gamma}(\exists h\mathrm{GCW} \wedge \Delta^0)$

---

**Input:** $k, m \in \mathbb{N}$, $\gamma \in (0,1)$, $h \in (0,1)$, sample access to $\mathbf{P} = \{\mathbf{P}(\cdot|S)\}_{S \in [m]_k}$,

**Initialization:** $\widetilde{\mathcal{A}} := $ Alg. 1, $i_0 \leftarrow \mathrm{UNSURE}$, $h' \leftarrow \frac{h}{3}$, $\gamma' \leftarrow \frac{\gamma}{\lceil m/(k-1)\rceil}$, let $S_1 \in [m]_k$ arbitrary,
$F_1 \leftarrow [m]$, $s \leftarrow 1$

$\triangleright S_s$ : candidate set in round $s$, $\quad F_s$ : remaining elements in round $s$
$\triangleright i_s \in S_s \cup \{\mathrm{UNSURE}\}$ : output of $\widetilde{\mathcal{A}}$ in round $s$

1: **while** $|F_s| > 0$ **do**
2: $\quad i_s \leftarrow \widetilde{\mathcal{A}}(h', \gamma', $ sample access to $\mathbf{P}(\cdot|S_s))$
3: $\quad F_{s+1} \leftarrow F_s \setminus S_s$
4: $\quad$ Write $F_{s+1} = \{j_1, \ldots, j_{|F_{s+1}|}\}$.
5: $\quad$ **if** $|F_{s+1}| < k$ **then**
6: $\quad\quad$ Fix distinct $j_{|F_{s+1}|+1}, \ldots, j_k \in [m] \setminus (F_{s+1} \cup \{i_s\})$.
7: $\quad$ **if** $i_s \in [m]$ **then** $S_{s+1} \leftarrow \{i_s, j_1, \ldots, j_{k-1}\}$
8: $\quad$ **else** $S_{s+1} \leftarrow \{j_1, \ldots, j_k\}$
9: $\quad s \leftarrow s + 1$
10: $i_s \leftarrow \widetilde{\mathcal{A}}(h', \gamma', $ sample access to $\mathbf{P}(\cdot|S_s))$
11: **if** $i_s \in [m]$ **then return** $i_s$
12: **else return** 1

---

*Proof of Theorem E.2.* Let us define the random variable $s^{\mathcal{A}} := \min\{s \in \mathbb{N} \,|\, F_s = \emptyset\} \in \mathbb{N} \cup \{\infty\}$ and suppose $\mathbf{P} \in PM_k^m$ to be arbitrary but fixed for the moment.

**Claim 1:** We have $s^{\mathcal{A}} \le s' := \lceil \frac{m}{k-1}\rceil$ a.s. w.r.t. $\mathbb{P}_\mathbf{P}$.

**Proof of Claim 1:** Assume on the contrary that $s^{\mathcal{A}} > s'$. Note that $|F_s|$ is monotonically decreasing in $s$. Whenever $|F_s| \ge k$, then $|S_s \cap F_s| \ge k - 1$ and thus $|F_{s+1}| \le |F_s| - (k-1)$ are fulfilled. Hence, $|F_s| \le m - s(k-1)$ holds for any $s \le s' - 1$. In particular, we have $|F_{s'-1}| \le k - 1$, which implies $F_{s'} = \emptyset$, contradicting the assumption $s^{\mathcal{A}} > s'$. This proves that $s^{\mathcal{A}} \le s'$ is fulfilled a.s. $\blacksquare$

Using that $\mathcal{A}$ makes exactly $s^{\mathcal{A}}$ calls of $\tilde{\mathcal{A}}$ (i.e., Algorithm 1) with parameters $h', \gamma'$ and each such call is executed with a sample complexity of exactly $\lceil 8\ln(4/\gamma')/h'^2\rceil$, the total sample complexity of $\mathcal{A}$ is at most

$$s'\lceil 8\ln(4/\gamma')/h'^2\rceil = \left\lceil \frac{m}{k-1}\right\rceil \left\lceil \frac{72}{h^2}\ln\left(\frac{4\lceil m/(k-1)\rceil}{\gamma}\right)\right\rceil,$$

which is in $\mathcal{O}\left(\frac{m}{kh^2}\ln\left(\frac{m}{k\gamma}\right)\right)$ as $\max\{m, k, h^{-1}, \gamma^{-1}\} \to \infty$. It remains to prove correctness of $\mathcal{A}$. Write $\mathcal{A}'$ for Algorithm 6 called with the same parameters as $\mathcal{A}$.

**Claim 2:** For any $\mathbf{P} \in PM_k^m$, we have
$$\mathbb{P}_\mathbf{P}\left(\mathbf{D}(\mathcal{A}) \ne \mathrm{GCW}(\mathbf{P})\right) = \mathbb{P}_\mathbf{P}\left(\mathbf{D}(\mathcal{A}') \ne \mathrm{GCW}(\mathbf{P})\right).$$

**Proof of Claim 2:** This follows directly from the fact that for any $S \in [m]_k$, different calls of $\tilde{\mathcal{A}}$ on $\mathbf{P}(\cdot|S)$ are by assumption executed on different samples of $\mathbf{P}(\cdot|S)$ and thus independent of each other. $\blacksquare$

This result shows that it is sufficient to prove correctness of $\mathcal{A}'$. In the following, we denote by $s$, $i_s$ $F_s$ and $S_s$ the internal statistics of $\mathcal{A}'$ and write $\widetilde{\mathcal{A}}_s$ for that instance of $\widetilde{\mathcal{A}}$, which is executed in $\mathcal{A}'$ to determine $i_s$. Let $\mathbf{P} \in PM_k^m(\exists h\mathrm{GCW} \wedge \Delta^0)$ be fixed and define $i := \mathrm{GCW}(\mathbf{P})$.

**Claim 3:** For all $s \le s'$ we have
$$\mathbb{P}_\mathbf{P}\left(i \in S_s \text{ and } i_s \ne i\right) \le \gamma'.$$

**Proof of Claim 3:** Suppose $B \in [m]_k$ with $i \in [m]$ and $\mathbb{P}_\mathbf{P}(S_s = B) > 0$ to be arbitrary but fixed for the moment. By assumption on $\mathbf{P}$ we have $\mathbf{P}(\cdot|B) \in \Delta_k^{3h'}$ and since $\widetilde{\mathcal{A}}_s$ is Algorithm 1 executed with parameters $h', \gamma'$ and sample access to $\mathbf{P}(\cdot|S_s)$ only, Lemma 4.3 assures

$\quad \mathbb{P}_\mathbf{P}\left(i \in S_s \text{ and } i_s \ne i | S_s = B\right)$

$\quad = \mathbb{P}_{\mathbf{P}(\cdot|B)}(\text{Alg. 1 started with } h', \gamma' \text{ does not output } \mathrm{mode}(\mathbf{P}(\cdot|B))) \le \gamma'.$

Claim 3 thus follows via summation over all such $B$. ■

On the event $\{T^{\mathcal{A}'} < \infty\}$, we infer from $[m] = F_0 \supseteq F_1 \supseteq \cdots \supseteq F_{s^{\mathcal{A}}} = \cdots = F_{s'} = \emptyset$ and $\forall s \leq s' : F_{s+1} = F_s \setminus S_s$ similarly as in the proof of Theorem E.1

$$\{\mathbf{D}(\mathcal{A}') \neq i\} \subseteq \bigcup_{s \leq s'} \{i \in S_s \text{ and } i_s \neq i\}.$$

As $T^{\mathcal{A}'} < \infty$ holds a.s. w.r.t. $\mathbb{P}_{\mathbf{P}}$, combining this with Claim 3 directly yields

$$\mathbb{P}_{\mathbf{P}}(\mathbf{D}(\mathcal{A}') \neq i) \leq \sum_{s \leq s'} \gamma' = \gamma,$$

which completes the proof. □

---

**Algorithm 6** Modification of Algorithm 5 for the proof of Theorem E.2

---

**Input:** $k, m \in \mathbb{N}$, $\gamma \in (0,1)$, $h \in (0,1)$, sample access to $\mathbf{P} = \{\mathbf{P}(\cdot|S)\}_{S \in [m]_k}$,

**Initialization:** $\widetilde{\mathcal{A}} := $ Algorithm 1, $i_0 \leftarrow$ UNSURE, $h' \leftarrow \frac{h}{3}$, $\gamma' \leftarrow \frac{\gamma}{\lceil m/(k-1) \rceil}$

$S_1 \leftarrow [k]$, $F_1 \leftarrow [m]$, $s \leftarrow 1$

1: Execute steps 1–8 of Algorithm 5.
2: let $s' \leftarrow \lceil \frac{m}{k-1} \rceil$
3: **while** $s < s'$ **do**
4:      $i_s \leftarrow \widetilde{\mathcal{A}}(h', \gamma', \text{sample access to } \mathbf{P}(\cdot|S_s))$
5:      $F_{s+1} \leftarrow F_s$, $S_{s+1} \leftarrow S_s$
6:      $s \leftarrow s + 1$
7:      $i_s \leftarrow \widetilde{\mathcal{A}}(h', \gamma', \text{sample access to } \mathbf{P}(\cdot|S_s))$
8: **return** $i_s$

---

## F    Proof of Theorem 5.2

Before proving Theorem 5.2, we require some preparation. For $S \in [m]_k$ and $\mathbf{p}, \mathbf{q} \in \Delta_S$ let us write $\mathrm{KL}(\mathbf{p}, \mathbf{q})$ for the *Kullback-Leibler divergence* of random variables $X \sim \mathrm{Cat}(\mathbf{p})$ and $Y \sim \mathrm{Cat}(\mathbf{q})$, i.e.,

$$\mathrm{KL}(\mathbf{p}, \mathbf{q}) = \begin{cases} \sum_{x \in S : p_x > 0} p_x \ln\left(\frac{p_x}{q_x}\right), & \text{if } \forall y \in S : q_y = 0 \Rightarrow p_y = 0, \\ \infty, & \text{otherwise.} \end{cases}$$

For the sake of convenience, we write in the binary case $k = 2$ simply $\mathrm{kl}(x,y) := \mathrm{KL}((x, 1-x), (y, 1-y))$ for any $x, y \in [0,1]$.

**Lemma F.1.**      *(i) For any $S \in [m]_k$ and $\mathbf{p}, \mathbf{q} \in \Delta_S$ we have*

$$\mathrm{KL}(\mathbf{p}, \mathbf{q}) \leq \sum_{x \in S} \frac{(p_x - q_x)^2}{q_x}.$$

*(ii) The inequality $\mathrm{kl}(\gamma, 1 - \gamma) \geq \ln((2.4\gamma)^{-1})$ holds for any $\gamma \in (0,1)$.*

*Proof.* The statement from (i) is Lemma 3 in [11] and for (ii) cf. Equation (3) in [21]. □

Given an algorithm $\mathcal{A}$, which tackles the problem $\mathcal{P}_k^{m,\gamma}(\Delta^h)$, let us write $S_t^{\mathcal{A}}$ for the query (element of $[m]_k$) made at time step $t$. Moreover, define $T_S^{\mathcal{A}}$ to be the number of times $\mathcal{A}$ makes the query $S \in [m]_k$ before termination, i.e., $T_S^{\mathcal{A}} = \sum_{t=1}^{T^{\mathcal{A}}} \mathbf{1}_{\{S_s^{\mathcal{A}} = S\}}$ and $T^{\mathcal{A}} = \sum_{S \in [m]_k} T_S^{\mathcal{A}}$ are fulfilled. Let $i_t^{\mathcal{A}} \in S_t^{\mathcal{A}}$ be the feedback observed by $\mathcal{A}$ at time step $t$, after having queried $S_t^{\mathcal{A}}$, and write $\mathcal{F}_t^{\mathcal{A}} := \sigma(S_1^{\mathcal{A}}, i_t^{\mathcal{A}}, \ldots, S_t^{\mathcal{A}}, i_t^{\mathcal{A}})$ for the sigma algebra generated by the behaviour and observed feedback of $\mathcal{A}$ until time $t$, and as usual $\mathcal{F}_{T^{\mathcal{A}}} := \mathcal{F}_{T^{\mathcal{A}}}^{\mathcal{A}} = \sigma\left(\bigcup_{t \leq T^{\mathcal{A}}} \mathcal{F}_t^{\mathcal{A}}\right)$.

Since $\mathcal{A}$ may be thought of as a multi-armed bandit with $\binom{m}{k}$ arms (one for each $S \in [m]_k$) and "rewards" $i_t^{\mathcal{A}} \in S_t^{\mathcal{A}}$, we may translate Lemma 1 from [21] to our setting in the following way:

**Lemma F.2.** *Let* $\mathbf{P}, \mathbf{P}' \in PM_k^m(\Delta^h \wedge \exists \mathrm{GCW})$ *with*[7] $\mathbf{P}(j|S), \mathbf{P}'(j|S) > 0$ *for any* $S \in [m]_k$ *and* $j \in S$. *If an algorithm* $\mathcal{A}$ *tackles* $\mathcal{P}_k^{m,\gamma}(\Delta^h \wedge \exists \mathrm{GCW})$ *and fulfills* $\mathbb{E}_{\mathbf{P}}[T^{\mathcal{A}}], \mathbb{E}_{\mathbf{P}'}[T^{\mathcal{A}}] < \infty$, *then*

$$\sum\nolimits_{S \in [m]_k} \mathbb{E}_{\mathbf{P}}\left[T_S^{\mathcal{A}}\right] \mathrm{KL}(\mathbf{P}(\cdot|S), \mathbf{P}'(\cdot|S)) \geq \sup\nolimits_{\mathcal{E} \in \mathcal{F}_{T^{\mathcal{A}}}} \mathrm{kl}\left(\mathbb{P}_{\mathbf{P}}(\mathcal{E}), \mathbb{P}_{\mathbf{P}'}(\mathcal{E})\right)$$

We are now ready to prove Theorem 5.2. The proof idea is similar to the one followed in the proof of Theorem 7 in [35].

*Proof of Theorem 5.2.* We prove the instance-wise and asymptotic lower bound separately.

**Part 1: Proof of the instance-wise bound**
After relabeling the items in $[m]$, we may suppose w.l.o.g. $\mathrm{GCW}(\mathbf{P}) = 1$ throughout the proof. Write for convenience $\mathbf{P}^{[1]} := \mathbf{P}$, recall that $m_S = \mathrm{mode}(\mathbf{P}^{[1]}(\cdot|S))$ for any $S \in [m]_k$ and define $\mathbf{P}^{[l]} \in PM_k^m(\Delta^h)$ for each $l \in \{2, \ldots, m\}$ via

$$\mathbf{P}^{[l]}(l|S) := \mathbf{P}^{[1]}(m_S|S), \quad \mathbf{P}^{[l]}(m_S|S) := \mathbf{P}^{[1]}(l|S),$$
$$\mathbf{P}^{[l]}(j|S) := \mathbf{P}^{[1]}(j|S) \text{ for all } j \in S \setminus \{l, m_S\} \tag{17}$$

for any $S \in [m]_k$ with $l \in S$ and

$$\mathbf{P}^{[l]}(j|S) := \mathbf{P}^{[1]}(j|S) \text{ for all } j \in S$$

for any $S \in [m]_k$ with $l \notin S$. Abbreviating $\mathbf{P}_S^{[r]} := \mathbf{P}^{[r]}(\cdot|S)$ we directly obtain $\mathrm{KL}\left(\mathbf{P}_S^{[1]}, \mathbf{P}_S^{[l]}\right) = 0$ whenever $S \notin [m]_k^{(l)} := \{S \in [m]_k \mid l \in S \text{ and } l \neq m_S\}$. Define

$$\Sigma(l) := \sum\nolimits_{S \in [m]_k^{(l)}} \mathbb{E}_{\mathbf{P}^{[1]}}\left[T_S^{\mathcal{A}}\right]$$

for each $l \in \{2, \ldots, m\}$. Now, suppose $l$ to be fixed for the moment and note that $\mathrm{GCW}(\mathbf{P}^{[l]}) = l$ holds by construction of $\mathbf{P}^{[l]}$. As $\mathcal{A}$ solves $\mathcal{P}_k^{m,\gamma}(\Delta^h)$, the event $\mathcal{E} := \{\mathbf{D}(\mathcal{A}) = 1\} \in \mathcal{F}_{T^{\mathcal{A}}}$ fulfills $\mathbb{P}_{\mathbf{P}^{[1]}}(\mathcal{E}) \geq 1 - \gamma$ and $\mathbb{P}_{\mathbf{P}^{[l]}}(\mathcal{E}) \leq \gamma$. Consequently, by applying part (ii) of Lemma F.1 and Lemma F.2, we obtain

$$\ln\left((2.4\gamma)^{-1}\right) \leq \mathrm{kl}\left(\mathbb{P}_{\mathbf{P}^{[1]}}(\mathcal{E}), \mathbb{P}_{\mathbf{P}^{[l]}}(\mathcal{E})\right)$$
$$\leq \sum\nolimits_{S \in [m]_k} \mathbb{E}_{\mathbf{P}^{[1]}}\left[T_S^{\mathcal{A}}\right] \mathrm{KL}\left(\mathbf{P}_S^{[1]}, \mathbf{P}_S^{[l]}\right)$$
$$= \sum\nolimits_{S \in [m]_k^{(l)}} \mathbb{E}_{\mathbf{P}^{[1]}}\left[T_S^{\mathcal{A}}\right] \mathrm{KL}\left(\mathbf{P}_S^{[1]}, \mathbf{P}_S^{[l]}\right),$$

that is,

$$\Sigma(l) \geq \ln\left((2.4\gamma)^{-1}\right) \min\nolimits_{S \in [m]_k^{(l)}} \frac{1}{\mathrm{KL}\left(\mathbf{P}_S^{[1]}, \mathbf{P}_S^{[l]}\right)}. \tag{18}$$

For any $S = \{i_1, \ldots, i_k\} \in [m]_k$ with $i_1 := m_S$ the term $\mathbb{E}_{\mathbf{P}^{[1]}}\left[T_S^{\mathcal{A}}\right]$ appears exactly $k - 1$ times as summand in

$$\Sigma(2) + \cdots + \Sigma(m) = \sum\nolimits_{l=2}^m \sum\nolimits_{S \in [m]_k : m_S \neq l \in S} \mathbb{E}_{\mathbf{P}^{[1]}}\left[T_S^{\mathcal{A}}\right],$$

namely as one summand in $\Sigma(i_2), \ldots, \Sigma(i_k)$ each. Hence, (18) lets us infer

$$(k-1)\mathbb{E}_{\mathbf{P}^{[1]}}\left[T^{\mathcal{A}}\right] = \sum\nolimits_{S \in [m]_k} (k-1)\mathbb{E}_{\mathbf{P}^{[1]}}\left[T_S^{\mathcal{A}}\right]$$
$$\geq \Sigma(2) + \cdots + \Sigma(m)$$
$$\geq \ln\left((2.4\gamma)^{-1}\right) \sum\nolimits_{l=2}^m \min\nolimits_{S \in [m]_k^{(l)}} \frac{1}{\mathrm{KL}\left(\mathbf{P}_S^{[1]}, \mathbf{P}_S^{[l]}\right)}.$$

---

[7]We put these conditions on $\mathbf{P}$ and $\mathbf{P}'$ in order to guarantee mutually absolutely continuity of the "rewards" $i_S \sim \mathrm{Cat}(\mathbf{P}(\cdot|S))$ resp. $i_S' \sim \mathrm{Cat}(\mathbf{P}'(\cdot|S))$, $S \in [m]_k$, which is formally required in Lemma 1 in [21].

This completes our proof of the instance-wise bound. ∎

**Part 2: Proof of the worst-case bound**

Since the statement is trivial for $h = 1$, we may assume w.l.o.g. $h \in (0, 1)$ in the following. Let us abbreviate $\Delta_{[m]_k} := \{\mathbf{w} = (w_S)_{S \in [m]_k} \in [0, 1]^{[m]_k} \mid \sum_{S \in [m]_k} w_S = 1\}$. For $S \in [m]_k$, write $S = \{S_{(1)}, \ldots, S_{(k)}\}$ with $S_{(1)} < \cdots < S_{(k)}$. Suppose $\varepsilon \in (0, 1/2)$ to be arbitrary but fixed for the moment and define $\mathbf{P}^{[1,\varepsilon]} \in PM_k^m(\exists \mathrm{GCW} \wedge \Delta^h)$ via

$$\mathbf{P}^{[1,\varepsilon]}(S_{(1)}|S) := \frac{1 + h + 2\varepsilon}{2}, \quad \mathbf{P}^{[1]}(S_{(2)}|S) := \frac{1 - h}{2}$$

and

$$\forall j \in \{3, \ldots, k\} : \mathbf{P}^{[1]}(S_{(j)}|S) := \frac{\varepsilon}{k - 2}.$$

for any $S \in [m]_k$. For $l \in \{2, \ldots, m\}$ let $\mathbf{P}^{[l,\varepsilon]}$ be as $\mathbf{P}^{[1,\varepsilon]}$ with $[m]$ being relabeled via the $l$-shift $\nu_l : [m] \to [m]$ given by

$$1 \mapsto l, \quad 2 \mapsto l + 1, \quad \ldots \quad m - l - 1 \mapsto m, \quad m - l \mapsto 1, \quad \ldots \quad m \mapsto l - 1,$$

i.e., $\mathbf{P}^{[l,\varepsilon]}(\nu_l(i_r)|\{\nu_l(i_1), \ldots, \nu_l(i_k)\}) = \mathbf{P}^{[1,\varepsilon]}(i_r|\{i_1, \ldots, i_k\})$ for any $\{i_1, \ldots, i_k\} \in [m]_k$ and $r \in [k]$. Then, $\mathbf{P}^{[l]} \in PM_k^m(\exists \mathrm{GCW} \wedge \Delta^h)$ and $\mathrm{GCW}(\mathbf{P}^{[l]}) = l$ hold for any $l \in [m]$. Write

$$\mathfrak{P}^*(\varepsilon) := \{\mathbf{P}^{[1,\varepsilon]}, \mathbf{P}^{[2,\varepsilon]}, \ldots \mathbf{P}^{[m,\varepsilon]}\}$$

and define

$$\mathfrak{P}_*(\neg l) := \{\mathbf{P} \in PM_k^m(\exists \mathrm{GCW} \wedge \Delta^h) \mid \mathrm{GCW}(\mathbf{P}) \neq l \text{ and } \forall S \in [m]_k : \min_{j \in S} \mathbf{P}(j|S) > 0\}.$$

For any $\mathbf{P}, \mathbf{P}' \in PM_k^m(\exists \mathrm{GCW} \wedge \Delta^h)$ fulfilling $\min_{S \in [m]_k} \min_{j \in S} \mathbf{P}(j|S) > 0$ as well as $\min_{S \in [m]_k} \min_{j \in S} \mathbf{P}'(j|S) > 0$ and $\mathrm{GCW}(\mathbf{P}) \neq \mathrm{GCW}(\mathbf{P}')$ Lemma F.2 guarantees similarly as above

$$\ln((2.4\gamma)^{-1}) \leq \sum_{S \in [m]_k} \mathbb{E}_{\mathbf{P}} \left[ T_S^{\mathcal{A}} \right] \mathrm{KL}(\mathbf{P}_S, \mathbf{P}'_S),$$

where we have written $\mathbf{P}_S$ resp. $\mathbf{P}'_S$ for $\mathbf{P}(\cdot|S)$ resp. $\mathbf{P}'(\cdot|S)$. Regarding arbitrariness of $\mathbf{P}$ and $\mathbf{P}'$ therein and using that $\mathbb{E}_{\mathbf{P}}[T^{\mathcal{A}}] > 0$ and $\left( \mathbb{E}_{\mathbf{P}}[T_S^{\mathcal{A}}]/\mathbb{E}_{\mathbf{P}}[T^{\mathcal{A}}] \right)_{S \in [m]_k} \in \Delta_{[m]_k}$ hold trivially for any $\mathbf{P} \in PM_k^m$, we may follow an idea from [18] (cf. the proof of Theorem 1 therein) and estimate

$$\ln((2.4\gamma)^{-1}) \leq \min_{\mathbf{P} \in \mathfrak{P}^*(\varepsilon)} \inf_{\mathbf{P}' \in \mathfrak{P}_*(\neg \mathrm{GCW}(\mathbf{P}))} \sum_{S \in [m]_k} \mathbb{E}_{\mathbf{P}} \left[ T_S^{\mathcal{A}} \right] \mathrm{KL}(\mathbf{P}_S, \mathbf{P}'_S)$$

$$\leq \min_{\mathbf{P} \in \mathfrak{P}^*(\varepsilon)} \mathbb{E}_{\mathbf{P}}[T^{\mathcal{A}}] \inf_{\mathbf{P}' \in \mathfrak{P}_*(\neg \mathrm{GCW}(\mathbf{P}))} \sum_{S \in [m]_k} \frac{\mathbb{E}_{\mathbf{P}} \left[ T_S^{\mathcal{A}} \right]}{\mathbb{E}_{\mathbf{P}}[T^{\mathcal{A}}]} \mathrm{KL}(\mathbf{P}_S, \mathbf{P}'_S)$$

$$\leq \sup_{\mathbf{w} \in \Delta_{[m]_k}} \min_{\mathbf{P} \in \mathfrak{P}^*(\varepsilon)} \mathbb{E}_{\mathbf{P}}[T^{\mathcal{A}}] \inf_{\mathbf{P}' \in \mathfrak{P}_*(\neg \mathrm{GCW}(\mathbf{P}))} \sum_{S \in [m]_k} w_S \mathrm{KL}(\mathbf{P}_S, \mathbf{P}'_S). \quad (19)$$

Suppose $\mathbf{w} \in \Delta_{[m]_k}$ to be arbitrary but fixed for the moment. The identity

$$k = k \sum_{S \in [m]_k} w_S = \sum_{l \in [m]} \sum_{S \in [m]_k : l \in S} w_S$$

assures the existence of some $l = l(\mathbf{w}) \in [m]$ with $\sum_{S \in [m]_k : l \in S} w_S \leq \frac{k}{m}$. Abbreviate $\mathbf{P} := \mathbf{P}^{[l,\varepsilon]}$. After relabeling $[m]$ via $\nu_l^{-1}$, we may assume w.l.o.g. $l = 1$ in the following, i.e. $\mathbf{P} = \mathbf{P}^{[1,\varepsilon]} \in \mathfrak{P}^*(\varepsilon)$. Define $\mathbf{P}' \in PM_k^m$ via

$$\mathbf{P}'(2|S) := \frac{1 + h + 2\varepsilon}{2}, \quad \mathbf{P}'(\min S \setminus \{2\} \mid S) := \frac{1 - h}{2} \quad \text{and} \quad \mathbf{P}'(j|S) := \frac{\varepsilon}{k - 2}$$

for any $j \in S \setminus \{2, \min(S \setminus \{2\})\}$, if $2 \in S$, and

$$\mathbf{P}'(j|S) := \mathbf{P}(j|S)$$

for any $j \in S$, if $2 \notin S$. From $\mathbf{P}' \in PM_k^m(\exists \text{GCW} \wedge \Delta^h)$ and $\text{GCW}(\mathbf{P}') = 2 \neq 1 = \text{GCW}(\mathbf{P})$ we infer $\mathbf{P}' \in \mathfrak{P}_*(\neg \text{GCW}(\mathbf{P}))$. In case $\{1,2\} \not\subseteq S$, we have $\mathbf{P}(j|S) = \mathbf{P}'(j|S)$ for any $j \in S$ and thus $\text{KL}(\mathbf{P}_S, \mathbf{P}'_S) = 0$. In the remaining case $\{1,2\} \subseteq S$ Lemma F.1 allows us to estimate

$$\text{KL}(\mathbf{P}_S, \mathbf{P}'_S)$$
$$= \text{KL}\left(\left(\frac{1+h+2\varepsilon}{2}, \frac{1-h}{2}, \frac{\varepsilon}{k-2} \cdots, \frac{\varepsilon}{k-2}\right), \left(\frac{1-h}{2}, \frac{1+h+2\varepsilon}{2}, \frac{\varepsilon}{k-2} \cdots, \frac{\varepsilon}{k-2}\right)\right)$$
$$\leq (h+\varepsilon)^2 \left(\frac{2}{1-h} + \frac{2}{1+h+\varepsilon}\right) = \frac{(4+2\varepsilon)(h+\varepsilon)^2}{(1-h)(1+h+\varepsilon)}.$$

Regarding the choice of $l = 1$ we infer

$$\sum_{S \in [m]_k} w_S \text{KL}(\mathbf{P}_S, \mathbf{P}'_S) = \sum_{S \in [m]_k : \{1,2\} \subseteq S} w_S \text{KL}(\mathbf{P}_S, \mathbf{P}'_S)$$
$$\leq \frac{(4+2\varepsilon)(h+\varepsilon)^2}{(1-h)(1+h+\varepsilon)} \sum_{S \in [m]_k : 1 \in S} w_S \leq \frac{k(4+2\varepsilon)(h+\varepsilon)^2}{m(1-h)(1+h+\varepsilon)}$$

and thus clearly

$$\mathbb{E}_{\mathbf{P}}[T^\mathcal{A}] \sum_{S \in [m]_k} w_S \text{KL}(\mathbf{P}_S, \mathbf{P}'_S) \leq \frac{k(4+2\varepsilon)(h+\varepsilon)^2}{m(1-h)(1+h+\varepsilon)} \mathbb{E}_{\mathbf{P}}[T^\mathcal{A}].$$

Since $\mathbf{w}$ was arbitrary and $\mathbf{P} = \mathbf{P}^{[l(\mathbf{w}),\varepsilon]}$, combining this with (19) yields

$$\ln((2.4\gamma)^{-1})$$
$$\leq \sup_{\mathbf{w} \in \Delta_{[m]_k}} \min_{\mathbf{P} \in \mathfrak{P}^*(\varepsilon)} \mathbb{E}_{\mathbf{P}}[T^\mathcal{A}] \inf_{\mathbf{P}' \in \mathfrak{P}_*(\neg \text{GCW}(\mathbf{P}))} \sum_{S \in [m]_k} w_S \text{KL}(\mathbf{P}_S, \mathbf{P}'_S)$$
$$\leq \sup_{\mathbf{w} \in \Delta_{[m]_k}} \mathbb{E}_{\mathbf{P}^{[l(\mathbf{w}),\varepsilon]}}[T^\mathcal{A}] \inf_{\mathbf{P}' \in \mathfrak{P}_*(\neg \text{GCW}(\mathbf{P}^{[l(\mathbf{w}),\varepsilon]}))} \sum_{S \in [m]_k} w_S \text{KL}\left(\mathbf{P}_S^{[l(\mathbf{w}),\varepsilon]}, \mathbf{P}'_S\right)$$
$$\leq \frac{k(4+2\varepsilon)(h+\varepsilon)^2}{m(1-h)(1+h+\varepsilon)} \sup_{\mathbf{w} \in \Delta_{[m]_k}} \mathbb{E}_{\mathbf{P}^{[l(\mathbf{w}),\varepsilon]}}[T^\mathcal{A}]$$
$$\leq \frac{k(4+2\varepsilon)(h+\varepsilon)^2}{m(1-h)(1+h+\varepsilon)} \max_{l \in [m]} \mathbb{E}_{\mathbf{P}^{[l,\varepsilon]}}[T^\mathcal{A}].$$

As $\varepsilon \in (0, 1/2)$ was arbitrary, we finally conclude

$$\sup_{\mathbf{P} \in PM_k^m(\exists \text{GCW} \wedge \Delta^h)} \mathbb{E}_{\mathbf{P}}\left[T^\mathcal{A}\right] \geq \sup_{\varepsilon \in (0,1/2)} \max_{l \in [m]} \mathbb{E}_{\mathbf{P}^{[l,\varepsilon]}}\left[T^\mathcal{A}\right]$$
$$\geq \sup_{\varepsilon \in (0,1/2)} \frac{m(1-h)(1+h+\varepsilon)}{k(4+2\varepsilon)(h+\varepsilon)^2} \ln((2.4\gamma)^{-1})$$
$$\geq \frac{m(1-h^2)}{4kh^2} \ln((2.4\gamma)^{-1}).$$

$\square$

**Remark F.3.** *The instance-wise bound in Theorem 5.2 appears to be maximal on an instance* $\mathbf{P} \in PM_k^m$ *defined via*

$$\mathbf{P}(m_S|S) := \frac{1-h+hk}{k} \quad \text{and} \quad \mathbf{P}(j|S) := \frac{1-h}{k} \text{ for each } j \in S \setminus \{m_S\}.$$

*with* $m_S := \min S$ *for each* $S \in [m]_k$. *Note that* $\mathbf{P}(m_S|S) = \mathbf{P}(j|S) + h$ *is fulfilled for each* $S \in [m]_k$, $j \in S \setminus \{m_S\}$. *Regarding the definition of* $m_S$ *we thus have* $\mathbf{P} \in PM_k^m(\Delta^h)$ *with* $\text{GCW}(\mathbf{P}) = 1$. *With* $\mathbf{P}^{[l]}(\cdot|S)$ *defined as in Theorem 5.2 we can estimate for each* $l \in \{2, \ldots, m\}$

*and $S \in [m]_k$ with $l \in S \setminus \{m_S\}$ via Lemma F.1*

$$\mathrm{KL}\left(\mathbf{P}(\cdot|S), \mathbf{P}^{[l]}(\cdot|S)\right) \leq \sum\nolimits_{j \in S} \frac{(\mathbf{P}(j|S) - \mathbf{P}^{[l]}(j|S))^2}{\mathbf{P}(j|S)}$$

$$= \frac{(\mathbf{P}(m_S|S) - \mathbf{P}^{[l]}(m_S|S))^2}{\mathbf{P}^{[l]}(m_S|S)} + \frac{(\mathbf{P}(l|S) - \mathbf{P}^{[l]}(l|S))^2}{\mathbf{P}^{[l]}(l|S)}$$

$$= \frac{(\mathbf{P}(m_S|S) - \mathbf{P}(l|S))^2}{\mathbf{P}(l|S)} + \frac{(\mathbf{P}(l|S) - \mathbf{P}(m_S|S))^2}{\mathbf{P}(m_S|S)}$$

$$= \frac{(\mathbf{P}(m_S|S) - \mathbf{P}(l|S))^2(\mathbf{P}(m_S|S) + \mathbf{P}(l|S))}{\mathbf{P}(m_S|S)\mathbf{P}(l|S)}$$

$$= \frac{h^2 k(1 - h + hk + 1 - h)}{(1 - h + hk)(1 - h)} \leq \frac{2kh^2}{1 - h},$$

*where we have used $hk \geq 0$ in the last step. Consequently, the instance-wise bound from Theorem 5.2 yields*

$$\mathbb{E}_{\mathbf{P}}\left[T^{\mathcal{A}}\right] \geq \frac{(m - 1)(1 - h)\ln((2.4\gamma)^{-1})}{2h^2 k(k - 1)} \in \Omega\left(\frac{m}{k^2 h^2} \ln \frac{1}{\gamma}\right),$$

*which is by a factor $1/k$ asymptotically smaller than the worst-case bound stated in Theorem 5.2.*

In the case of dueling bandits ($k = 2$), the instance-dependent bound from Theorem 5.2 reduces to

$$\mathbb{E}_{\mathbf{P}}\left[T^{\mathcal{A}}\right] \geq \ln((2.4\gamma)^{-1}) \sum\nolimits_{l \in [m] \setminus \{i\}} \frac{1}{\mathrm{KL}(\mathbf{P}(\cdot|\{i, l\}), \mathbf{P}^{[l]}(\cdot|\{i, l\}))}$$

$$= \ln((2.4\gamma)^{-1}) \sum\nolimits_{l \in [m] \setminus \{i\}} \frac{1}{\mathrm{kl}(\mathbf{P}(i|\{i, l\}), \mathbf{P}(l|\{i, l\}))}$$

for any $\mathbf{P} \in PM_2^m(\exists \mathrm{GCW} \wedge \Delta^h)$ with $\mathrm{GCW}(\mathbf{P}) = i$ and any solution $\mathcal{A}$ to $\mathcal{P}_2^{m,\gamma}(\exists \mathrm{GCW} \wedge \Delta^h)$. By means of this, we obtain the following worst-case sample lower bound, which is by a factor $\frac{2(m-1)}{m}$ larger than the one stated in Theorem 5.2.

**Corollary F.4.** *If $\mathcal{A}$ solves $\mathcal{P}_2^{m,\gamma}(\exists \mathrm{GCW} \wedge \Delta^h)$, then*

$$\sup\nolimits_{\mathbf{P} \in PM_2^m(\exists \mathrm{GCW} \wedge \Delta^h)} \mathbb{E}_{\mathbf{P}}\left[T^{\mathcal{A}}\right] \geq \frac{(m - 1)(1 - h^2)}{4h^2} \ln((2.4\gamma)^{-1}).$$

*Proof.* Define $\mathbf{P} \in PM_2^m(\exists \mathrm{GCW} \wedge \Delta^h)$ via $\mathbf{P}(i|\{i, j\}) := \frac{1+h}{2}$ for any $1 \leq i < j \leq m$. Theorem 5.2 and Lemma F.1 allow us to infer

$$\mathbb{E}_{\mathbf{P}}\left[T^{\mathcal{A}}\right] \geq \frac{(m - 1)\ln((2.4\gamma)^{-1})}{\mathrm{kl}((1 + h)/2, (1 - h)/2)}$$

$$\geq (m - 1)\ln((2.4\gamma)^{-1})\left(\frac{2h^2}{1 - h} + \frac{2h^2}{1 + h}\right)^{-1}$$

$$= \frac{(m - 1)(1 - h^2)\ln((2.4\gamma)^{-1})}{4h^2}.$$

$\square$

**Remark F.5.** *Suppose $\mathcal{A}$ solves $\mathcal{P}_k^{k,\gamma}(\Delta^h)$, let $\mathbf{p} \in \Delta_k^h$ and write $i := \mathrm{mode}(\mathbf{p})$. According to Prop. C.1 we have*

$$\mathbb{E}_{\mathbf{p}}\left[T^{\mathcal{A}}\right] \geq \max\nolimits_{l \in [m] \setminus \{i\}} \frac{1 - 2\gamma}{2\phi_{l,i}(\mathbf{p})(p_l + p_i)} \left\lceil \frac{\ln((1 - \gamma)/\gamma)}{\ln\left((1/2 + \phi_{l,i}(\mathbf{p}))/(1/2 - \phi_{l,i}(\mathbf{p}))\right)} \right\rceil =: \mathrm{LB}_1(\mathbf{p}, \gamma)$$

*with $\phi_{l,i}(\mathbf{p}) := \frac{p_i - p_l}{2(p_l + p_i)}$, and Thm. 5.2 guarantees*

$$\mathbb{E}_{\mathbf{p}}\left[T^{\mathcal{A}}\right] \geq \frac{\ln((2.4\gamma)^{-1})}{k - 1} \sum\nolimits_{l \in [k] \setminus \{i\}} \left(p_l \ln\left(\frac{p_l}{p_i}\right) + p_i \ln\left(\frac{p_i}{p_l}\right)\right)^{-1} =: \mathrm{LB}_2(\mathbf{p}, \gamma).$$

*In an empirical study we observed $\mathrm{LB}_1(\mathbf{p}, \gamma) > \mathrm{LB}_2(\mathbf{p}, \gamma)$ for all of 1000 parameters $\mathbf{p}$ sampled iid and uniformly at random from $\Delta_k^0$, for any $(k, \gamma) \in \{5, 10, 15\} \times \{0.01, 0.05, 0.1\}$. For example, we have $\mathrm{LB}_1((0.2, 0.2, 0.15, 0.2, 0.25), 0.05) \approx 252 > 152.9 \approx \mathrm{LB}_2((0.2, 0.2, 0.15, 0.2, 0.25), 0.05)$. This indicates that the instance-wise lower bound of Prop. C.1 is larger than that from Thm. 5.2.*

## G    Additional Experiments

### G.1    Comparison of DKWT with PAC-WRAPPER

In this section, we provide further experimental results. First, we repeat the experiment regarding the comparison of DKWT and PW from Section 7 for $\boldsymbol{\theta} = (1, 2^{-1}, 2^{-2}, \ldots, 2^{-9})$, with $\gamma = 0.1$ and for different values of $k$. Table 5 shows the results obtained with 10 repetitions. Similar to the results in the main paper, both algorithms apparently keep the desired confidence of 90%, but PW requires far more samples for this. The fact that the observed sample complexities are not throughout decreasing in $k$ is supposedly due to the large standard errors and the little number of repetitions. However, they strongly indicate that DKWT outperforms PW in terms of sample complexity.

Table 5: Comparison of DKWT with PAC-WRAPPER (PW) on $\boldsymbol{\theta} = (1, 2^{-1}, 2^{-2}, \ldots, 2^{-9})$

| | $T^{\mathcal{A}}$ | | Accuracy | |
| $k$ | DKWT | PW | DKWT | PW |
|---|---|---|---|---|
| 2 | **8310** (0.0) | 2509460 (226634.0) | 1.00 | 1.00 |
| 3 | **4078** (348.9) | 46277676 (30635546.4) | 1.00 | 1.00 |
| 4 | **3925** (1014.3) | 775101 (108535.7) | 1.00 | 1.00 |
| 5 | **3397** (529.2) | 6450264 (1363336.3) | 1.00 | 1.00 |
| 6 | **2213** (465.0) | 130069344 (77405795.5) | 1.00 | 1.00 |
| 7 | **2856** (507.4) | 253206333 (125199242.0) | 1.00 | 1.00 |
| 8 | **3817** (608.9) | 27159632 (12458792.0) | 1.00 | 1.00 |
| 9 | **2855** (680.7) | 146229360 (79427860.6) | 1.00 | 1.00 |

Next, we compare DKWT with PW on synthetic data considered in [35], where PW has first been introduced. We restrict ourselves to $\boldsymbol{\theta}^{\mathrm{arith}}, \boldsymbol{\theta}^{\mathrm{geo}} \in [0, 1]^{16}$ defined via

$$\theta_1^{\mathrm{arith}} := 1, \quad \forall i \in [15] : \theta_{i+1}^{\mathrm{arith}} := \theta_i^{\mathrm{arith}} - 0.06,$$

$$\theta_1^{\mathrm{geo}} := 1, \quad \forall i \in [15] : \theta_{i+1}^{\mathrm{geo}} := \frac{4}{5} \cdot \theta_i^{\mathrm{geo}},$$

because the other synthetic datasets considered in Fig. 2 of [35] (i.e., **g1** and **b1**) are not in $PM_k^m(\exists \mathrm{GCW} \wedge \Delta^0)$, which is formally required for DKWT. For $\boldsymbol{\theta} \in \{\boldsymbol{\theta}^{\mathrm{arith}}, \boldsymbol{\theta}^{\mathrm{geo}}\}$ we execute DKWT with $\gamma = 0.01$ for 1000 repetitions on feedback generated by $\mathbf{P}(\boldsymbol{\theta})$ and report the mean termination time (and standard error in brackets) as well as the observed accuracy in Table 6. A look at Fig. 2 of [35] reveals that DKWT indeed outperforms PW on both datasets while still keeping its theoretical guarantees.

Table 6: Results of DKWT on $\boldsymbol{\theta}^{\mathrm{arith}}$ and $\boldsymbol{\theta}^{\mathrm{geo}}$

| | $T^{\mathcal{A}}$ of $\mathcal{A} =$DKWT | Accuracy |
|---|---|---|
| $\boldsymbol{\theta}^{\mathrm{arith}}$ | 1277781 (22284.0) | 1.00 |
| $\boldsymbol{\theta}^{\mathrm{geo}}$ | 55132 (910.5) | 1.00 |

### G.2    Comparison of DKWT with SELECT, SEEBS and EXPLORE-THEN-VERIFY

The authors of [29] restrict themselves in the analysis of their algorithm SEEBS to probability models $\mathbf{P} \in PM_2^m(\exists \mathrm{GCW} \wedge \Delta^0)$, which fulfill both of the following conditions:

- *Strong stochastic transitivity (SST):* For all distinct distinct $i, j, k \in [m]$ with $\mathbf{P}(i|\{i, j\}) \geq 1/2$ and $\mathbf{P}(j|\{j, k\}) \geq 1/2$ we have

$$\mathbf{P}(i|\{i, k\}) \geq \max\{\mathbf{P}(i|\{i, j\}), \mathbf{P}(j|\{j, k\})\}$$

- *Stochastic triangle inequality (STI):* For all distinct $i, j, k \in [m]$ we have

$$|\mathbf{P}(i|\{i, k\}) - 1/2| \leq |\mathbf{P}(i|\{i, j\}) - 1/2| + |\mathbf{P}(j|\{j, k\}) - 1/2|.$$

In particular, SEEBS is only proven to identify the correct GCW with confidence $\geq 1 - \gamma$ for any $\mathbf{P}$ in a set $PM_2^m(\exists \text{GCW} \wedge \Delta^0 \wedge \text{SST} \wedge \text{STI}) \subsetneq PM_2^m(\exists \text{GCW} \wedge \Delta^0)$.

Table 7 shows the observed termination times (and standard errors thereof in brackets) of DKWT, SELECT, SEEBS and EtV compared on $PM_2^m(\exists \text{GCW} \wedge \Delta^h)$ in the same manner as done in Section 7, where the true value of $h$ is revealed to SELECT but not to DKWT, SEEBS or EtV. The results for $m \in \{5, 10\}$ are averaged over 100 repetitions and are partly the same as shown in Table 4, the results for $m \in \{15, 20\}$ are averaged over 10 repetitions. Table 8 shows the corresponding accuracies observed during the experiment underlying Table 7. Almost any algorithm in any case achieves an accuracy of $\geq 95\%$, the only exception is SELECT for $m = 20$ and $h = 0.20$ and this is supposedly due to the little number of repetitions considered. The results indicate again that DKWT outperforms SEEBS but not SELECT. Since SELECT obtains as further information the true value of $h$, this is not at all surprising.

Table 7: Comparison of DKWT, SELECT, SEEBS and EXPLORE-THEN-VERIFY (EtV)

| | | $T^{\mathcal{A}}$ | | | |
|---|---|---|---|---|---|
| $m$ | $h$ | DKWT | SELECT | SEEBS | EtV |
| 5 | 0.20 | 6010 (293.2) | **252** (4.2) | 7305 (432.1) | 8601 (589.2) |
| 5 | 0.15 | 8874 (460.0) | **460** (7.3) | 13393 (904.5) | 11899 (986.9) |
| 5 | 0.10 | 15769 (1457.1) | **989** (17.0) | 19802 (1543.2) | 260171 (210678.1) |
| 5 | 0.05 | 31454 (4127.4) | **3924** (68.6) | 36855 (3533.2) | 156534 (115903.1) |
| 10 | 0.20 | 14334 (492.8) | **565** (2.5) | 16956 (617.9) | 26115 (969.2) |
| 10 | 0.15 | 18563 (734.5) | **1009** (4.2) | 27527 (1126.7) | 32548 (2514.6) |
| 10 | 0.10 | 33040 (1625.1) | **2245** (9.7) | 47330 (2138.2) | 68858 (11304.5) |
| 10 | 0.05 | 78660 (6517.2) | **8971** (39.2) | 83877 (5842.6) | 220098 (92484.9) |
| 15 | 0.20 | 21932 (1618.1) | **803** (13.9) | 28605 (2161.5) | 54197 (5307.3) |
| 15 | 0.15 | 27446 (2500.0) | **1436** (12.3) | 38084 (4985.3) | 78753 (27741.4) |
| 15 | 0.10 | 45737 (6709.6) | **3248** (20.7) | 67383 (8117.1) | 116014 (24282.2) |
| 15 | 0.05 | 114152 (18704.0) | **12993** (82.7) | 108738 (19780.4) | 2804238 (2560594.1) |
| 20 | 0.20 | 32038 (1209.2) | **1154** (8.7) | 40910 (2893.1) | 78286 (3451.5) |
| 20 | 0.15 | 39792 (3923.6) | **2080** (12.6) | 58793 (4828.0) | 122582 (24065.7) |
| 20 | 0.10 | 87667 (13380.8) | **4616** (32.3) | 105249 (13231.8) | 631195 (281883.6) |
| 20 | 0.05 | 134628 (21743.3) | **18375** (138.2) | 164439 (30175.4) | 2094505 (1694236.4) |

Table 8: Accuracies of DKWT, SELECT, SEEBS and EXPLORE-THEN-VERIFY (EtV) corresponding to the experiment of Table 7

| | | Accuracy | | | |
|---|---|---|---|---|---|
| $m$ | $h$ | DKWT | SELECT | SEEBS | EtV |
| 5 | 0.20 | 1.00 | 0.97 | 1.00 | 1.00 |
| 5 | 0.15 | 1.00 | 1.00 | 1.00 | 1.00 |
| 5 | 0.10 | 1.00 | 0.99 | 1.00 | 1.00 |
| 5 | 0.05 | 1.00 | 1.00 | 1.00 | 1.00 |
| 10 | 0.20 | 1.00 | 0.95 | 1.00 | 1.00 |
| 10 | 0.15 | 1.00 | 0.98 | 1.00 | 1.00 |
| 10 | 0.10 | 1.00 | 0.99 | 1.00 | 1.00 |
| 10 | 0.05 | 1.00 | 1.00 | 1.00 | 1.00 |
| 15 | 0.20 | 1.00 | 1.00 | 1.00 | 1.00 |
| 15 | 0.15 | 1.00 | 1.00 | 1.00 | 1.00 |
| 15 | 0.10 | 1.00 | 1.00 | 1.00 | 1.00 |
| 15 | 0.05 | 1.00 | 1.00 | 1.00 | 1.00 |
| 20 | 0.20 | 1.00 | 0.90 | 1.00 | 1.00 |
| 20 | 0.15 | 1.00 | 1.00 | 1.00 | 1.00 |
| 20 | 0.10 | 1.00 | 1.00 | 1.00 | 1.00 |
| 20 | 0.05 | 1.00 | 1.00 | 1.00 | 1.00 |

### G.3 Comparison of DKWT with Alg. 5

Finally, we compare DKWT and Alg. 5 by means of their average sample complexity and accuracy when executed on $1000$ instances $\mathbf{P}$, which were drawn independently and uniformly at random from (a) $PM_k^5(\exists \mathrm{GCW} \wedge \Delta^h)$ and (b) $PM_k^5(\exists h\mathrm{GCW} \wedge \Delta^{0.01})$. We choose $\gamma = 0.05$ and restrict ourselves due to $PM_5^5(\exists \mathrm{GCW} \wedge \Delta^h) = PM_5^5(\exists h\mathrm{GCW} \wedge \Delta^{0.01})$ to $k \in \{2, 3, 4\}$. Similarly as in our comparison to SELECT, Alg. 5 is revealed the true value of $h$ and started with this as parameter. The results are collected in (a) Table 9 and (b) Table 10. In any of the cases (a) and (b), DKWT apparently outperforms Alg. 5 if $h$ is smaller than some threshold $h_0$, and the value of $h_0$ appears to be significantly larger for (a) than for (b). This indicates that Alg. 5 may be preferable over DKWT if $h(\mathbf{P})$ is small and $\mathbf{P} \in PM_k^m(\exists h'\mathrm{GCW} \wedge \Delta^0)$ holds for some a priori known $h' \in (0, 1/2)$.

Table 9: Comparison of DKWT with Alg. 5 on $PM_k^5(\exists \mathrm{GCW} \wedge \Delta^h)$

| | | $T^{\mathcal{A}}$ | | Accuracy | |
|---|---|---|---|---|---|
| $k$ | $h$ | DKWT | Alg. 5 | DKWT | Alg. 5 |
| 2 | 0.9 | 4155 (0.0) | **2664** (0.0) | 1.00 | 1.00 |
| 2 | 0.7 | **4155** (0.0) | 4405 (0.0) | 1.00 | 1.00 |
| 2 | 0.5 | **4155** (0.0) | 8630 (0.0) | 1.00 | 1.00 |
| 2 | 0.3 | **4195** (12.7) | 23970 (0.0) | 1.00 | 1.00 |
| 2 | 0.1 | **14729** (423.4) | 215695 (0.0) | 1.00 | 1.00 |
| 3 | 0.9 | 2298 (0.0) | **1464** (0.0) | 1.00 | 1.00 |
| 3 | 0.7 | **2298** (0.0) | 2418 (0.0) | 1.00 | 1.00 |
| 3 | 0.5 | **2298** (0.0) | 4737 (0.0) | 1.00 | 1.00 |
| 3 | 0.3 | **2381** (17.5) | 13155 (0.0) | 1.00 | 1.00 |
| 3 | 0.1 | **14933** (436.1) | 118383 (0.0) | 1.00 | 1.00 |
| 4 | 0.9 | 1428 (0.0) | **1356** (0.0) | 1.00 | 1.00 |
| 4 | 0.7 | **1428** (0.0) | 2238 (0.0) | 1.00 | 1.00 |
| 4 | 0.5 | **1428** (0.0) | 4386 (0.0) | 1.00 | 1.00 |
| 4 | 0.3 | **1492** (15.0) | 12183 (0.0) | 1.00 | 1.00 |
| 4 | 0.1 | **13449** (306.4) | 109626 (0.0) | 1.00 | 1.00 |

Table 10: Comparison of DKWT with Alg. 5 on $PM_k^5(\exists h\mathrm{GCW} \wedge \Delta^{0.01})$

| | | $T^{\mathcal{A}}$ | | Accuracy | |
|---|---|---|---|---|---|
| $k$ | $h$ | DKWT | Alg. 5 | DKWT | Alg. 5 |
| 2 | 0.9 | 53913 (7092.4) | **2477** (8.0) | 1.00 | 1.00 |
| 2 | 0.7 | 63647 (8322.8) | **4124** (13.0) | 1.00 | 1.00 |
| 2 | 0.5 | 54370 (6753.8) | **8167** (24.2) | 1.00 | 1.00 |
| 2 | 0.3 | 59488 (7738.0) | **23275** (53.4) | 1.00 | 1.00 |
| 2 | 0.1 | **60682** (7256.5) | 214358 (236.4) | 1.00 | 1.00 |
| 3 | 0.9 | 40359 (6188.7) | **1464** (0.0) | 1.00 | 1.00 |
| 3 | 0.7 | 27069 (3621.2) | **2418** (0.0) | 1.00 | 1.00 |
| 3 | 0.5 | 37362 (5774.2) | **4737** (0.0) | 1.00 | 1.00 |
| 3 | 0.3 | 31553 (4551.6) | **13155** (0.0) | 1.00 | 1.00 |
| 3 | 0.1 | **45929** (5277.3) | 118383 (0.0) | 1.00 | 1.00 |
| 4 | 0.9 | 24164 (4446.0) | **1356** (0.0) | 1.00 | 1.00 |
| 4 | 0.7 | 39088 (6293.2) | **2238** (0.0) | 1.00 | 1.00 |
| 4 | 0.5 | 31835 (5462.0) | **4386** (0.0) | 1.00 | 1.00 |
| 4 | 0.3 | 31796 (5131.8) | **12183** (0.0) | 1.00 | 1.00 |
| 4 | 0.1 | **48202** (5765.3) | 109626 (0.0) | 1.00 | 1.00 |