# OpenReview forum: "Identification of the Generalized Condorcet Winner in Multi-dueling Bandits"
_NeurIPS.cc/2021/Conference — NeurIPS 2021 Poster_

### Official Review · Reviewer_RVxf · 2021-07-13

**Rating:** 6
**Confidence:** 4

**Summary:**

This work studies the BAI problem in the multi-dueling bandits.
It proposes algorithms with theoretical guarantee and provides related lower bounds.
There are detailed comparisons to existing works.
Moreover, it compare the proposed algorithm with other existing algorithms numerically.

**Limitations And Societal Impact:**

There is one suggestion for future work: if the query size can be different at different time step $t$, does the algorithm still work? Can there be any theoretical guarantee?

**Main Review:**

In general, the paper is well organized.
It involves rich amount of results and the amount of notations is a bit heavy. It would be easier for the readers to understand if there can be a table to compare all the upper and lower bounds.

However, I have one major concern regarding the single bandit case $m=k$. What is the difference from this case to the standard multi-arm bandits --- the problem where the agent aims to find $1$ best arm among $m$ arms?



=========

Thanks for the detailed response from the author(s). I have read them and all other discussions. Though I tend to remain my score, I am more favor to accept the paper for the time being.

**Time Spent Reviewing:**

2

---

> ### Author Response · Authors · 2021-08-09
> **Answer to Reviewer RVxf**
>
> Dear reviewer, thank you for your helpful feedback.
>
> #  Overview of Results
>
> Please note that we already give an overview of the lower and upper bounds in asymptotic form in Table 1. There, for the ease of readability, we have decided to leave out the technical instance-wise versions and to highlight their differences of the asymptotics w.r.t. $m,k,h$ and $\gamma$. Moreover, please note that we provide a list of frequently used symbols at the beginning of the appendix to increase readability of the theoretical parts.
>
> # Difference "Single-bandit Case" and Standard Multi-armed Bandits
>
> In the standard multi-armed bandit setting, there are $m$ different arms, and at each time step, the learner may choose one of them (pull one arm), whereupon a noisy numerical reward is observed, drawn from the unknown reward distribution associated to that arm. The best-arm-identification problem is usually understood as "finding the arm with the highest expected reward." In the setting of our $m=k$-case, the learner can only "pull all arms at once" and obtain as feedback only the identity of the arm that has generated the highest noisy numerical reward, but not the rewards themselves (i.e., the rewards are latent). In particular, the "single-bandit case" differs from the standard multi-armed bandit setting as follows:
>
> 1. The feedback: It is of a **qualitative nature** (which arm generated the highest noisy reward) instead of **numerical nature** (what is the generated noisy reward).
>
> 2. The actions: Only one possible action (pull all arms at once) instead of one for each arm.
>
> # Different Query Sizes
>
> Even though our solution works only for the case of equal-size query sets, we think that the general case (i.e., with arbitrary query-set size) would also be solvable, albeit coming with more complex solutions and sample complexity bounds. We think that this could indeed be an interesting direction for future work, so thanks a lot for this suggestion. Such a variant of the problem setting has in fact been investigated in [2], but again only for the Plackett-Luce model and in an $(\epsilon,\delta)$-PAC scenario.
>
> # References
>
> [2]: A. Saha and A. Gopalan. From PAC to Instance-Optimal Sample Complexity in the Plackett-Luce Model. IMCL 2020.

---

> > ### Comment · Reviewer_RVxf · 2021-09-01
> > **Reply to author**
> >
> > Dear authors(s),
> >
> > Thanks for solving my concerns. The contribution of the work seems to be more significant.
> > Though I tend to remain my score, I am now more favor to accept the paper.

---

### Official Review · Reviewer_1aHC · 2021-07-16

**Rating:** 7
**Confidence:** 5

**Summary:**

This paper studies the problem of identifying the generalized condorcet winner (GCW) of the multi-dueling bandits. In multi-dueling bandits, each time the learner chooses a set with size $k$ to sample, and the sample returns an arm as the winner according to a distribution. The GCW is an arm that has the largest probability to win for any set that contains it. The focus on this problem is to find the sample complexity (aka number of samples needed) bounds for finding the GCW of a set of arms with confidence $1-\gamma$. This problem can be viewed as a generalization of the "top item selection from noisy comparisons" problems or the "best arm identification from dueling bandits" problems. The model considered in this paper is very general and covers the popular  Plackett-Luce model as well as most of the cases that have been studied in the literature.

Under this general model, the authors developed the sample complexity upper bounds and lower bounds and the bounds are optimal up to at most a log factor. Numerical results also show the improvements of the proposed algorithm.

**Limitations And Societal Impact:**

There are two main limitations in this work.

First, the algorithm and the lower bounds only work for worst cases, while in practice, we may expect instance-wise bounds that depend on the actual gaps of differences of the winning probabilities. There are instances where some of the gaps are very small but most of the gaps are very large. For these instances, the bounds in this paper may be much larger than those utilizing the unequal gaps.

Second, for the PL model, it is interesting to discuss whether and to what degree the value of $k$ can influence the sample complexity if the parameters remain the same.


**Main Review:**

This paper studies a novel and interesting abstraction of the long existing problems of dueling bandits or ranking from comparisons, and this abstraction is general and reasonable enough to cover most of the cases that would be met in the literature and in practice. The algorithm proposed in this work is non-trivial, and its sample complexity upper bound is close enough to the lower bound with only a log factor.

The results of this work are good enough for an acceptance in my opinion.

The paper is well written, with good mathematical definitions as well as verbal explanations.

The results are significant for understanding the theoretical limits for best arm/item identification from multi-wise comparisons or multi-dueling bandits.

Therefore, I tend to give a score 7.

**Time Spent Reviewing:**

2

---

> ### Author Response · Authors · 2021-08-09
> **Answer to Reviewer 1aHC**
>
> Dear reviewer, thanks for your positive feedback.
>
> # Worst-case vs. Instance-wise Bounds
>
> Being aware that our paper is already somewhat technical, we have tried to formulate the results in the main part in a more intuitive way, deferring the rigorous details to the appendix. However, please note that both our lower and upper bounds for GCW identification are instance-wise:
> 1.	In Theorem 5.2. the math-display in Line 224 is an instance-wise lower bound, while the math-display in 226 is the worst-case lower bound inferred from the instance-wise lower bound.
> 2.	The sample complexity in Theorem 6.2 is stated in a worst-case manner, but we write before the theorem, that the instance-wise sample complexity can be found in Theorem E.1 in the appendix.
> 3.	Our bounds for the case $m=k$ stated in Proposition 4.1 and Proposition 4.4 (with the rigorous technical definition of $t_{0}(\gamma,h)$ deferred to the proof) are instance-wise as well.
>
> # Addressing the Questions
>
> ## Role of $k$ in PL
>
> The question to which extent the value of $k$ influences the complexity of the problem if restricting to PL-instances with the same parameters is theoretically already answered by the instance-wise sample complexity bounds in [2], which we restated as Theorems B.1 and B.2 in the appendix. Note that the dependency of the lower bound on $k$ is simply $1/k$. For the upper bound, there is an additional $k$ in the $\log$-term as well as in the instance-dependent constant $\Theta_{[k]}$ appearing as an additional factor.
>
> ## Sample Complexity Sensitivity
>
> Since the focus of our paper is actually not on the PL-case but rather on non-parametric assumptions of the environment (via the very general probabilistic model) and the comparison of these, we would refrain from analyzing this question further in this work. The main reason is that, for the dueling bandit case, it was recently shown that parametric assumptions such as the PL model do not lead to qualitatively stricter lower bounds on sample complexity compared to non-parametric assumptions in similar identification tasks [14]. Therefore, we believe that the seemingly more difficult non-parametric setting is of greater importance for the research community.
>
> # References
>
> [2]: A. Saha and A. Gopalan. From PAC to Instance-Optimal Sample Complexity in the Plackett-Luce Model. IMCL 2020.
>
> [14] R. Heckel, N. Shah, K. Ramchandran, and M.  Wainwright. Active ranking from pairwise comparisons and when parametric assumptions do not help. The Annals of Statistics 2019.

---

### Official Review · Reviewer_NPR2 · 2021-07-17

**Rating:** 6
**Confidence:** 3

**Summary:**

This paper proposes the notion of a "generalised Condorcet winner"  (GCW) among contestants in a tournament (or candidates in an election). Every subset of contestants is associated with a probability distribution over the contestants in the subset; a GCW is the most probable choice in every subset. The authors propose a sampling algorithm for determining a GCW of a tournament with sufficiently high probability, and provide both lower and upper bounds on the sample complexity. They consider special cases based on the number of contestants (m) and the size of the queried subset (k).

**Ethical Concerns:**

I do not see any ethical issues.

**Limitations And Societal Impact:**

I do not see any potential negative societal impact.

**Main Review:**

In my opinion, the paper has two major weaknesses.

1. The paper is premised on the assumption that the input tournament will indeed have a GCW--which, as the authors themselves acknowledge, is not guaranteed. It is not clear how the proposed algorithms will fare when run on a tournament that is without a GCW. Moreover, I do not find sufficient motivation for the query model (in which subsets of size k, k being arbitrary, are queried). The experiments provided are on a synthetic data set: they do not bring out the practical relevance of the query model and the algorithms--how they can benefit real-world applications. In short, I am unable to attribute any practical significance to this paper. It must be assessed solely for its theoretical merit.

2. The paper is very hard to read. To be fair to the authors, the contents are technically dense: multiple problems, multiple parameters, and so on. And I do believe that their presentation is CORRECT. However, substantive portions of the paper contain dense notation interspersed with the commentary (starting with Section 3, wherein the problem is introduced), hindering CLARITY. I do not expect readers who are not already familiar with this problem and associated notation to be able to follow the paper.

I have two specific questions for the authors.

- The case of m = k, which amounts to mode estimation, has recently been studied in the following paper, which provides an algorithm, as well as upper and lower bounds that match up to a logarithmic factor.

Sequential Mode Estimation with Oracle Queries. Dhruti Shah, Tuhinangshu Choudhury, Nikhil Karamchandani, and Aditya Gopalan. In Proc. AAAI 2020, pp. 5644—5651. AAAI Press, 2020.

Could the authors please compare/distinguish their own problem and bounds from the one in this paper? Also, to the best of my understanding, the authors have not shown experimental results related to mode estimation, and comparisons with other mode estimation algorithms. Why not?

- Could the authors please explain the qualitative interpretation of the parameter h? I was under the impression while reading the paper that it was similar in spirit to the usual tolerance parameter epsilon in PAC formulations, but the authors seem to indicate in the conclusion that there is room for a "margin" epsilon over and above h.

Post-rebuttal: Based on the authors' response and the discussion among the reviewers, I am now more positive about both the theoretical merit of the paper and its readability. I update my score accordingly.

**Time Spent Reviewing:**

2

---

> ### Author Response · Authors · 2021-08-09
> **Answer to Reviewer NPR2**
>
> Dear reviewer, thanks a lot for your helpful feedback and the reference to [1]. We have not been aware of this paper, though it is definitely related to our work. Let us first address the two main issues you see in our work.
>
> # Addressing Weaknesses
>
> 1. Yes, we assume the existence of a GCW throughout the entire work, even if it may be violated in practice. However, such assumptions are common in theoretical papers in this field [4], and in comparisons to others (such as the PL-assumption in [2] or the assumption of strong stochastic transitivity in [3]) this assumption is rather mild. In fact, the GCW-assumption is the analogue of the existence of the "best arm" for best-arm-identification in Dueling bandits -- an assumption commonly made in the related literature [4].
>
>  Note that our considered query model, which restricts to query sets of equal size $k$, has also been considered in other papers -- mostly for the goal of regret minimization [6, 7, 11], but also for GCW identification under assumptions stricter than the mere existence of a GCW [5]. It generalizes the Dueling Bandits setting, which itself is an increasingly broad field with numerous real-life applications, see for instance [8] or [9, 10] (the latter two make even stronger assumptions than merely the existence of a Condorcet Winner).
> However, our considered problem setting allows modeling more general sequential decision processes in which qualitative comparisons of more than two available choice alternatives can be carried out at once. There are multiple practically relevant applications of this more general variant, such as algorithm configuration [12] or online retrieval evaluation [13]. If accepted, we would add such applications to our related work section.
>
> 2. We agree that the paper is quite technical. However, we think that the notation used is necessary and convenient for a formally correct presentation of our results. We do not see any redundant or superfluous notation. The reader does not have to be familiar with our notation beforehand, because every notation is formally introduced before its usage. Please note that in order to increase the readability, we have (a) deferred detailed versions of some results to the appendix and restricted ourselves to simplified versions of these in the main paper, and (b) provided a list of frequently used symbols at the beginning of the appendix.
>
> You say that the paper "must be assessed solely for its theoretical merit," an appraisal that we essentially agree with. Our work is definitely focused on the theoretical analysis of the GCW identification and NOT on applications. The other reviewers think that the "rich amount of results" are "significant for understanding the theoretical limits for best arm/item identification from multi-wise comparisons or multi-dueling bandits“ and "good enough for an acceptance".
>
> # Mode estimation
>
> Thanks a lot for the pointer to [1], we did not know this paper before. It contains solutions to our problem in the special case $m=k$, and the results presented therein are comparable but not identical to ours:
>
> Their lower bound (Theo. 3) depends on a measure changing argument whereas our first one (Prop. 4.1) is inferred from the optimality of the Sequential Probability Ratio Test. Asymptotically, these two are similar, but ours also provides as additional information the asymptotic behavior for $k\rightarrow \infty$. However, our second lower bound (Prop. 4.2) is not covered by [1].
>
> Their upper bound (Theo. 2) is asymptotically slightly larger than ours (Prop. 4.4): It (a) is slightly increasing in $k$ whereas ours does not depend at all on $k$, and (b) its dependence on $h(p) = p_{1}-p_{2}$ (assuming $p_{1} > p_{2} > ... > p_{k}$) is of the order $\ln(1/h(p))$, whereas ours is of order $\ln \ln (1/h(p))$. If accepted, we will of course relate our results to [1].
>
> Even though the mode estimation plays an important role in our algorithmic solution for GCW identification, the main focus of this work is on the GCW identification problem itself, which goes far beyond the former. Nevertheless, we already solve the mode estimation problem theoretically (almost) optimal and the theoretical results of [1] are not (asymptotically) better, whence comparing these (or even further) mode estimation algorithms is of no benefit for our sample complexity bounds for GCW identification, at least not from a theoretical point of view. That said, the question which mode estimation component works best in practice sounds like an interesting question that could be addressed in future work.
>
> # Interpretation of parameter $h$
>
> The parameter $h(P)$ measures the difficulty of the considered underlying environment described by $P$. Restricting e.g. to $\Delta^{h}$-instances means that in any allowed query-set $S$, there is one element that is "at least $h$ better" than any other one, i.e., $P(i|S) \geq P(j|S) +h$ for any $j\not= i$. Thus, it is a distributional assumption, which guarantees that the GCW is -- if existent -- unique and (to some extent) "easy to identify".
>
> The parameter $\epsilon$ in the PAC setting instead relaxes the considered **goal** of the problem: Instead of finding an optimal element (the GCW), one only aims for an $\epsilon$-optimal element, i.e., one that is at most "by $\epsilon$ outperformed" by any other element. Here, the $\epsilon$-optimal element does not have to be unique and there might be multiple "correct" decisions. Of course, if a GCW exists, it is $\epsilon$-optimal, but it is not necessarily the only $\epsilon$-optimal element.
>
> # References
>
> [1]: D. Shah, T. Choudhury, N. Karamchandani, and A. Gopalan. Sequential Mode Estimation with Oracle Queries. AAAI 2020.
>
> [2]: A. Saha and A. Gopalan. From PAC to Instance-Optimal Sample Complexity in the Plackett-Luce Model. IMCL 2020.
>
> [3]: M. Falahatgar, A. Orlitsky, V. Pichapati, and A. T. Suresh. Maximum selection and ranking under noisy comparisons. ICML 2017.
>
> [4]: V. Bengs, R. Busa-Fekete, A. El Mesaoudi-Paul, and E. Hüllermeier. Preference-based online learning with dueling bandits: A survey. JMLR 2021.
>
> [5]: A. Saha and A. Gopalan. Best-item learning in random utility models with subset choices. AISTATS 2020.
>
> [6]: A. Saha and A. Gopalan. Battle of bandits. UAI 2018.
>
> [7]: V. Bengs and E. Hüllermeier. Preselection bandits. ICML 2020.
>
> [8]: S. Guo, S. Sanner, T. Graepel, and W. Buntine. Score-based Bayesian skill learning. ECML/PKDD 2012.
>
> [9]: Y. Sui and J. Burdick. Clinical online recommendation with subgroup rank feedback. RecSys 2014.
>
> [10]: Y. Sui, Y. Yue, and J. Burdick. Correlational dueling bandits with application to clinical treatment in large decision spaces. IJCAI 2017.
>
> [11]: A. Agarwal, N. Johnson, and S. Agarwal. Choice bandits. NeurIPS 2020.
>
> [12]: A. El Mesaoudi-Paul, D. Weiß, V. Bengs, E. Hüllermeier, and K. Tierney. Pool-based realtime algorithm configuration: A preselection bandit approach. LION 2020.
>
> [13]: A. Schuth, H. Oosterhuis, S. Whiteson, and M. de Rijke. Multileave gradient descent for fast online learning to rank. WSDM 2016.

---

### Decision · Program_Chairs · 2021-09-27

**Decision:**

Accept (Poster)

**Comment:**

The reviewers came to consensus that the theoretical strength overtakes the concerns such as the assumption on the existence of GCW and the heaviness of the technical materials. I agree with these opinions and please sincerely address the concerns raised by the reviewers in the final version such as the relation with the previous work pointed out by NPR2. In particular, though I agree that the dense notation of this paper might be somewhat unavoidable, I expect that the authors make the best effort to improve the readability with more intuitions for them so that the paper becomes a good starting point for multi-dueling setting.